# Fading to Grow: Growing Preference Ratios via Preference Fading Discrete Diffusion for Recommendation

**Guoqing Hu**[‡]    **An Zhang**[‡*]  **Shuchang Liu**[§]   **Wenyu Mao**[‡]   **Jiancan Wu**[‡]

**Xun Yang**[‡]   **Xiang Li**[§]   **Lantao Hu**[§]   **Han Li**[§]   **Kun Gai**[§]   **Xiang Wang**[‡]

[‡]University of Science and Technology of China
[§]Independent Researcher
{HugoChinn}@mail.ustc.edu.cn
{an_zhang,xyang21}@ustc.edu.cn
{wenyumao2, wujcan, xiangwang1223}@gmail.com
{xiangli.th11, hulantao, lee.han06}@gmail.com, gai.kun@qq.com

## Abstract

Recommenders aim to rank items from a discrete item corpus in line with user interests, yet suffer from extremely sparse user preference data. Recent advances in diffusion models have inspired diffusion-based recommenders, which alleviate sparsity by injecting noise during a forward process to prevent the collapse of perturbed preference distributions. However, current diffusion-based recommenders predominantly rely on continuous Gaussian noise, which is intrinsically mismatched with the discrete nature of user preference data in recommendation. In this paper, building upon recent advances in discrete diffusion, we propose **PreferGrow**, a discrete diffusion-based recommender system that models preference ratios by fading and growing user preferences over the discrete item corpus. PreferGrow differs from existing diffusion-based recommenders in three core aspects: (1) Discrete modeling of preference ratios: PreferGrow models relative preference ratios between item pairs, rather than operating in the item representation or raw score simplex. This formulation aligns naturally with the discrete and ranking-oriented nature of recommendation tasks. (2) Perturbing via preference fading: Instead of injecting continuous noise, PreferGrow fades user preferences by replacing the preferred item with alternatives—physically akin to negative sampling—thereby eliminating the need for any prior noise assumption. (3) Preference reconstruction via growing: PreferGrow reconstructs user preferences by iteratively growing the preference signals from the estimated ratios. PreferGrow offers a well-defined matrix-based formulation with theoretical guarantees on Markovianity and reversibility, and it demonstrates consistent performance gains over state-of-the-art diffusion-based recommenders across five benchmark datasets, highlighting both its theoretical soundness and empirical effectiveness. Our codes are available at https://github.com/Hugo-Chinn/PreferGrow.

## 1 Introduction

Recommender systems aim to rank items from a discrete item set that align with user interests, where ones in the user-item interaction matrix denote observed interactions, and zeros indicate unobserved

---

[*]Corresponding authors.

39th Conference on Neural Information Processing Systems (NeurIPS 2025).

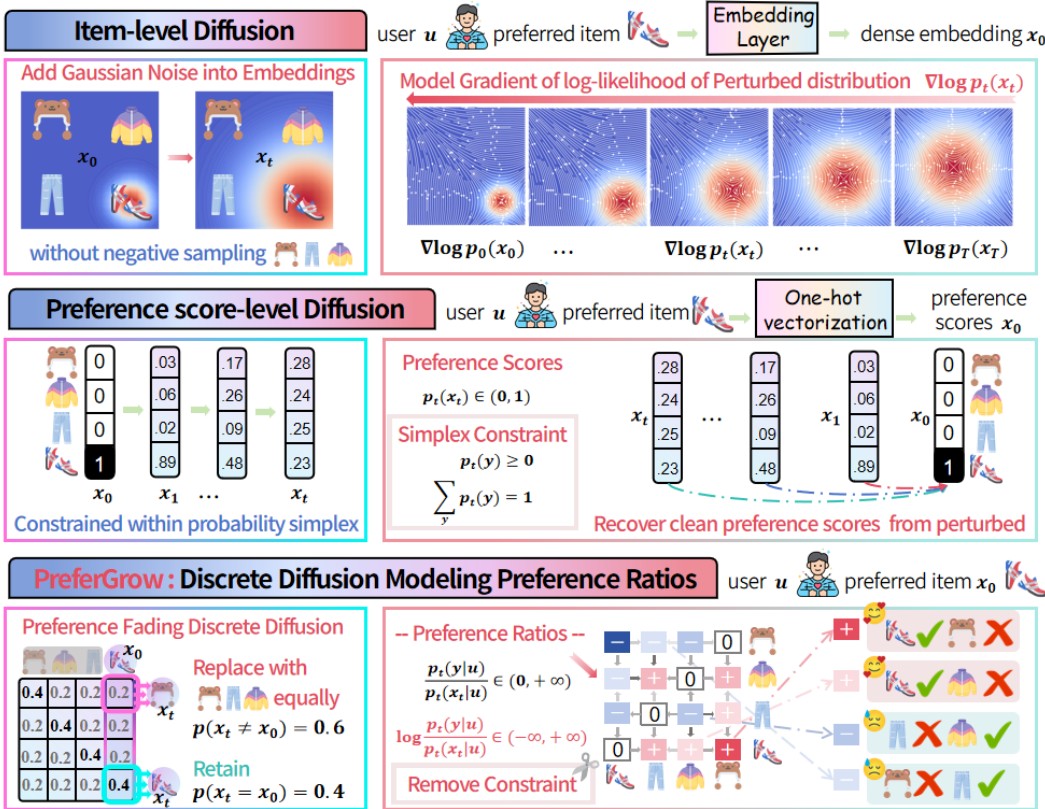

Figure 1: Comparison of diffusion-based recommenders in terms of modeling targets and perturbation strategies. Item-level diffusion recommenders (top) add Gaussian noise into dense item embeddings but overlook negative signals. Preference score-level diffusion recommenders (middle) perturb one-hot preference vectors under the constraints of the probability simplex. PreferGrow (bottom) directly models discrete preference ratios via preference fading.

or missing entries. In real-world scenarios, this interaction matrix is often extremely sparse [1, 2], which poses a significant challenge for recommenders in accurately modeling user preferences [3–5]. Recent advances in diffusion models (DMs) offer a promising solution: by injecting noise during the forward process, DMs transform sparse data into smoother, denser distributions while preventing collapse into isotropic zero values [6–8]. Motivated by these properties, diffusion-based recommenders [9–37] have proliferated in recent studies, showing strong potential in addressing data sparsity and improving preference modeling.

Current diffusion-based recommenders predominantly adopt continuous noise as the perturbation mechanism by adding noise into either dense item embeddings or one-hot preference score vectors. As shown in Figure 1, item-level diffusion recommenders [9–27] apply Gaussian noise to user-interacted item embeddings during the forward process, and learn a score function—the gradient of the perturbed distribution's log-likelihood—to recover item embeddings during the reverse process. However, these methods overlook the negative signals in recommendations, lacking mechanisms such as negative sampling to differentiate user preferences [9, 27]. In contrast, preference score-level diffusion recommenders [28–37] model users' preference scores among the full item corpus, *i.e.,* user interacted one-hot vectors, by perturbing them within the probability simplex in the forward process. While the reverse process attempts to reconstruct the preference scores, the simplex constraints—non-negativity and normalization—introduce additional optimization difficulties [38]. Worse still, both diffusion-based recommenders assume a prior noise in the sampling process, *e.g.,* Gaussian [9] or Bernoulli [29], which may poorly reflect the inherently discrete and sparse nature of the user preference data in recommendation scenarios.

Building on recent advances in discrete diffusion models [39, 38, 40], we propose **PreferGrow**, a discrete diffusion-based recommender that models the *preference ratios* by fading and growing user preferences through forward and backward processes. Unlike existing continuous diffusion-based

recommenders that model preference scores (*e.g.,* probabilities that users interacted with items), PreferGrow directly models relative preference ratios over a discrete item set. This formulation aligns more naturally with the discrete and ranking-oriented nature of recommendation tasks and avoids the strong constraints imposed by the probability simplex. In the forward perturbation, PreferGrow fades user preference by replacing the preferred item with alternatives, enabling explicit negative sampling and alleviating data sparsity without relying on predefined noise distribution. In the backward generation, PreferGrow reconstructs user preferences by iteratively growing the preference signal from the estimated ratios. We further provide a theoretical analysis showing that these forward and backward processes preserve key properties of discrete diffusion models.

PreferGrow offers a unified and theoretically grounded framework for discrete diffusion-based recommendation, characterized by a well-defined matrix-based formulation, flexible and interpretable preference ratio modeling aligned with diverse negative sampling strategies, and consistently superior empirical performance.

- **Theoretical foundation:** At the core of the formulation for PreferGrow is an idempotent fading matrix used to replace preferred items during the forward perturbation process. We provide a closed-form solution of this preference fading matrix and theoretically prove that its idempotent property is critical for ensuring both the Markov property and reversibility of the diffusion process.
- **Flexible modeling:** By parameterizing the preference fading matrix, PreferGrow flexibly supports point-wise, pair-wise, and hybrid preference ratios modeling. These variants correspond to distinct and physically interpretable negative sampling strategies, enabling the framework to adapt to diverse modeling targets in recommendation.
- **Empirical validation:** Extensive experiments on five benchmark datasets demonstrate that Prefer-Grow consistently outperforms existing diffusion-based recommenders, validating the practical effectiveness of its theoretical foundations.

## 2 Preliminaries

**User preference data** is represented as a pair $(u, i)$, where $u$ denotes the user and $i$ is the preferred item. The preference score $p(i|u)$ indicates the probability that the user $u$ interacts with item $i$. In sequential recommendation, $u$ denotes the item sequence that the user has interacted with.

**Diffusion-based Recommenders** generally consist of three major components: a forward noise-adding process for perturbation, a generative modeling target, and a backward denoising process for reconstruction. Specifically, **item-level diffusion-based recommenders** encode a preferred item $i$ into its dense embedding $\mathbf{x}_0$, and instantiate three components as follows [41]:

- *forward perturbation*: $\mathbf{x}_t = \sqrt{\alpha_t}\mathbf{x}_0 + \sqrt{1 - \alpha_t}\epsilon_t$, where $\alpha_t \in [0, 1], \epsilon_t \sim \mathcal{N}(0, I)$.
- *modeling target*: network $\mathbf{s}_\Theta(\mathbf{x}_t, t, u) \approx \nabla_{\mathbf{x}_t} \log p_t(\mathbf{x}_t|u) = \frac{\sqrt{\alpha_t}\mathbf{x}_0 - \mathbf{x}_t}{1 - \alpha_t} = -\frac{\epsilon_t}{\sqrt{1 - \alpha_t}}$.
- *backward generation*: $\mathbf{x}_s = \sqrt{\alpha_s}\frac{\mathbf{x}_t + (1 - \alpha_t)\mathbf{s}_\Theta(\mathbf{x}_t, t, u)}{\sqrt{\alpha_t}} - \sqrt{1 - \alpha_s}\sqrt{1 - \alpha_t}\mathbf{s}_\Theta(\mathbf{x}_t, t, u), s < t$.

**Preference score–level diffusion-based recommenders** represent the preferred item $i$ as a one-hot preference score $\mathbf{x}_0$, modeling user preference over the entire item space. Early works [28, 30, 34] adopt Gaussian noise priors $\mathcal{N}(\cdot, \cdot)$; however, Gaussian perturbations are incompatible with the probability simplex constraints inherent to preference scores. To address this mismatch, subsequent studies [29, 35] introduce discrete priors to replace the Gaussian assumption that preserves the simplex structure throughout the diffusion process. For instance, RecFusion [29] adopts a Bernoulli prior $\mathcal{B}(\cdot; \cdot)$, resulting in a binomial diffusion formulation:

- *forward perturbation*: $\mathbf{x}_t \sim \mathcal{B}\left(\mathbf{x}_t; \alpha_t\mathbf{x}_0 + \frac{\alpha_t(1 - \alpha_t)}{2}\right), \alpha_t \in [0, 1]$.
- *modeling target*: $\mathcal{B}(\mathbf{x}_{t-1}; \mathbf{s}_\Theta(\mathbf{x}_t, t, u)) \approx p(\mathbf{x}_{t-1} \mid \mathbf{x}_t)$.
- *backward generation*: $\mathbf{x}_{t-1} \sim \mathcal{B}(\mathbf{x}_{t-1}; \mathbf{s}_\Theta(\mathbf{x}_t, t, u))$.

The other approach employs a Categorical prior $\mathrm{Cat}(\cdot; \cdot)$:

- *forward perturbation:* $p_t(\mathbf{x}_t \mid \mathbf{x}_{t-1}) = \mathrm{Cat}(\mathbf{x}_t; \overline{\mathbf{Q}}_t\mathbf{x}_0)$ where $\overline{\mathbf{Q}}_t = \prod_{i=1}^{t} \mathbf{Q}_i$.

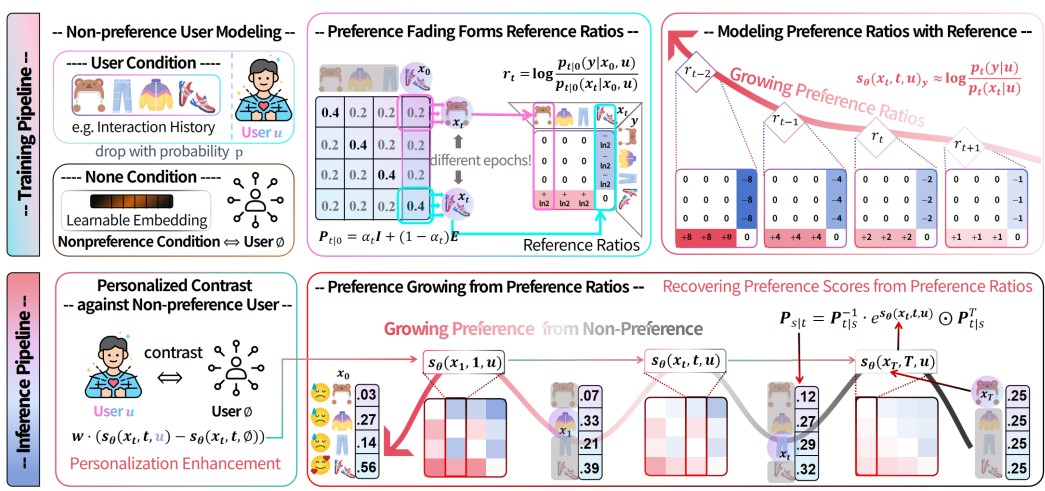

Figure 2: The overall training and inference pipeline of PreferGrow under the pair-wise setting.

- *modeling target:* $\mathbf{s}_\Theta(\mathbf{x}_t, t, u) \approx \mathbf{x}_0$.

- *backward generation:* $p_t\left(\mathbf{x}_{t-1} \mid \mathbf{x}_t, \mathbf{x}_0 = \mathbf{s}_\Theta(\mathbf{x}_t, t, u)\right) = \mathrm{Cat}\left(\mathbf{x}_{t-1}; \frac{\mathbf{Q}_t^\top \mathbf{x}_t \odot \overline{\mathbf{Q}}_{t-1} \mathbf{s}_\Theta(\mathbf{x}_t, t, u)}{\mathbf{x}_t^\top \overline{\mathbf{Q}}_t \mathbf{s}_\Theta(\mathbf{x}_t, t, u)}\right)$.

**Preference Ratios** $\log \frac{p(i_p|u)}{p(i_d|u)}$ characterize the relative preference between items, and a positive value indicates a more preferred item $i_p$ and a less preferred item $i_d$. Notably, preference modeling in RLHF [42–44] and DPO [45–48] is grounded in the Bradley–Terry model [49], which explicitly models preference ratios as in Equation (1). This underscores the effectiveness of preference ratios in more faithfully capturing user preferences.

$$p(i_p \succ i_d \mid u) = \frac{p(i_p \mid u)}{p(i_p \mid u) + p(i_d \mid u)} = \frac{1}{1 + \exp\left(-\log \frac{p(i_p|u)}{p(i_d|u)}\right)} = \sigma(\log \frac{p(i_p \mid u)}{p(i_d \mid u)}). \quad (1)$$

**Discrete Diffusion Models** previously are formulated based on the following Kolmogorov forward and backward equations [50, 51, 38]:

$$\frac{\partial \mathbf{P}_{t|s}}{\partial t} = \mathbf{Q}_t \mathbf{P}_{t|s}, \qquad \frac{\partial \mathbf{P}_{s|t}}{\partial t} = \mathbf{R}_s \mathbf{P}_{s|t}, \quad (2)$$

where $\mathbf{P}_{t|s}$ denotes the transition probability matrix from time $s$ to time $t$, $\mathbf{Q}_t$ is the forward transition rate matrix at time $t$, and $\mathbf{R}_s$ is the reverse-time transition rate matrix at time $s$. Discrete diffusion models further model the ratios of data distributions through the following score entropy loss [38]:

$$\mathcal{L}_{SE} = \mathbb{E}_{x_0 \sim p_{data}} \mathbb{E}_{t \in U[0,T]} \mathbb{E}_{x_t \sim p_{t|0}(\cdot|x_0)} \left[\sum_{y \in \mathcal{X}} \mathbf{Q}_t(x_t, y) \cdot l_{SE}(x_0, x_t, y)\right], \quad (3)$$

$$l_{SE}(x_0, x_t, y) = e^{\mathbf{s}_\Theta(y, x_t)} - \frac{p_{t|0}(y|x_0)}{p_{t|0}(x_t|x_0)} \mathbf{s}_\Theta(y, x_t) + \frac{p_{t|0}(y|x_0)}{p_{t|0}(x_t|x_0)} [\log \frac{p_{t|0}(y|x_0)}{p_{t|0}(x_t|x_0)} - 1], \quad (4)$$

where $\mathcal{X}$ denotes the discrete state space, and $\mathbf{s}_\Theta(y, x_t)$ is the output of a neural network that estimates the preference ratio between $y$ and $x_t$ at timestep $t$.

## 3 Method

As illustrated in Figure 2, the pipeline of PreferGrow consists of three stages: the preference fading discrete diffusion process (Section 3.1), the modeling of preference ratios with reference via score entropy loss (Section 3.2), and the preference growing reverse generation process (Section 3.3). Additionally, PreferGrow models a non-preference user and achieves personalized enhancement (Section 3.4). Theoretical proofs are provided in Appendix C, while the computation of score entropy loss under different fading matrix settings is detailed in Appendix D.

## 3.1 Forward Perturbation: Preference Fading Discrete Diffusion Process

PreferGrow operates directly on the discrete item set $\mathcal{X}$, wherein $x_0$ represents the preferred item $i$ of user $u$. PreferGrow first fades user preference based on whether to retain or not to retain the preferred item $x_0$. The whole fading process is further achieved by progressively decreasing the probability of retention $\alpha_t$ from timestep $0$ to $T$, fading user preferences towards non-preference.

### 3.1.1 Preference Fading Forms Reference Ratios

Building upon recent advances in discrete DM [39, 38, 40], we introduce a preference fading discrete diffusion model tailored for recommendation. For a user preference data $(u, x_0)$, we fade user preference by retaining the target item $x_t = x_0$ with probability $\alpha_t$ and replacing $x_0$ according to discrete distribution $\mathbf{E}(\mathcal{X}, x_0)$ as the following equation:

$$p_{t|0}(x_t|x_0) = \alpha_t \delta_{x_0}(x_t) + (1 - \alpha_t)\mathbf{E}(x_t, x_0), x_t \in \mathcal{X}. \tag{5}$$

where Dirac delta function $\delta_{x_0}(x_t)$ equals 1 when $x_t = x_0$, and 0 otherwise. Also, in matrix form:

$$\mathbf{P}_{t|0} = \alpha_t \mathbf{I} + (1 - \alpha_t)\mathbf{E}, \tag{6}$$

where $\mathbf{I}$ is the identity matrix, representing retention, and $\mathbf{E}$ is a matrix whose column sums equal 1, defining the replacing mode for perturbation. For simplicity, we refer to $\mathbf{E}$ as fading matrix. We further demonstrate that the idempotence of the fading matrix $\mathbf{E}$ ensures the Markov property and reversibility of preference fading discrete diffusion process, which is crucial for the reverse generation.

**Theorem 1.** *Suppose $\alpha_t : [0, T] \to [0, 1]$ is a strictly decreasing function with $\alpha_0 = 1$ and $\alpha_T = 0$. If $\mathbf{E} \in \mathbb{R}^{N \times N}$ is idempotent, i.e., $\mathbf{E}^2 = \mathbf{E}$, the following properties hold:*

- **Markov property:** *$\{\mathbf{P}_{t|0}\}_{t=0}^T$ is a Markov process and satisfies the Chapman-Kolmogorov equation:*

$$\mathbf{P}_{t|s} := \frac{\alpha_t}{\alpha_s}\mathbf{I} + \left(1 - \frac{\alpha_t}{\alpha_s}\right)\mathbf{E}, \tag{7}$$

$$\mathbf{P}_{t|r} = \mathbf{P}_{t|s}\mathbf{P}_{s|r}, \quad \text{for all } 0 \le r \le s \le t \le 1. \tag{8}$$

- **Invertibility:** *Each $\mathbf{P}_{t|s}$ is invertible, and its inverse is given by:*

$$\mathbf{P}_{t|s}^{-1} = \frac{\alpha_s}{\alpha_t}\mathbf{I} + \left(1 - \frac{\alpha_s}{\alpha_t}\right)\mathbf{E}. \tag{9}$$

Upon obtaining $x_t$ through preference fading, as illustrated in Figure 2, we can compute the reference ratios for all items $\mathcal{X}$ at timestep $t$ with fading awareness $1 - \alpha_t$ as follows:

$$r_t(x_0, x_t, y) = \log \frac{p_{t|0}(y \mid x_0, u)}{p_{t|0}(x_t \mid x_0, u)} = \log \frac{\alpha_t \delta_{x_0}(y) + (1 - \alpha_t)\mathbf{E}(y, x_0)}{\alpha_t \delta_{x_0}(x_t) + (1 - \alpha_t)\mathbf{E}(x_t, x_0)}, \forall y \in \mathcal{X}. \tag{10}$$

As shown in Figure 2, when the preferred item $x_0$ is retained, *i.e.,* $x_t = x_0$, the reference ratio $r_t(x_0, x_t = x_0, y \neq x_0)$ is negative, reflecting a tendency to stay with the current item. Conversely, when $x_0$ is replaced, *i.e.,* $x_t \neq x_0$, the reference ratio $r_t(x_0, x_t \neq x_0, y = x_0)$ tends to be positive, highlighting the preference for the positive item $x_0$. In this case, the faded item $x_t$ can be interpreted as a negative item, establishing a conceptual connection to the mechanism of negative sampling.

### 3.1.2 Design Paradigms of the Idempotent Fading Matrix

Given that designing such an idempotent fading matrix is crucial, we derive that, if the preference fading process converges to a unified state, the fading matrix $\mathbf{E}$ can be expressed in closed form.

**Proposition 1.** *Suppose the preference fading Markov process $\{\mathbf{P}_{t|0}\}_{t=0}^T$ converges to a unified non-preference state $\vec{p}_T$. Then, the fading matrix $\mathbf{E}$ can be expressed in closed form as:*

$$\mathbf{E} = \frac{\vec{p}_T \vec{1}^\top}{\vec{1}^\top \vec{p}_T}. \tag{11}$$

Equation (11) gives a rank-1 solution for the fading matrix $\mathbf{E}$. Most existing discrete diffusion models [38–40, 52–55] implicitly adopt such rank-1 instances (i.e., the absorbing and uniform cases). In Appendix D, examine several representative rank-1 configurations of $\mathbf{E}$ that, in a physical sense, correspond to distinct negative sampling strategies and thus induce different forms of preference ratios. Specifically, by parameterizing the rank-1 preference-fading matrix, PreferGrow flexibly supports point-wise, pair-wise, and hybrid preference-ratio modeling. These variants align with interpretable negative-sampling schemes, enabling the framework to adapt to diverse recommendation objectives. For each setting, we provide analytical solutions for the fading matrix, closed-form expressions for the reference ratios, and simplified training losses. We believe prior works on negative sampling [56, 57] offer valuable guidance for designing more effective rank-1 fading matrices; a systematic exploration is left for future work. Beyond the rank-1 case, Appendix E derives a general rank-$r$ solution for $\mathbf{E}$. We present a closed-form characterization together with a discussion of its physical interpretation. Notably, a rank-$r$ fading matrix induces a quantization of the item space — i.e., a partition into $r$ components. A comprehensive study of this quantization mechanism and its algorithmic implications is deferred to our future work.

### 3.2 Modeling Target: Preference Ratios with Reference via Score Entropy

We begin by computing the reference ratios $r_t(x_0, x_t, y)$ induced by preference fading, as defined in Equation (10). These reference ratios are then employed to guide the modeling of the user-conditioned preference ratio $\log \frac{p_t(y|u)}{p_t(x_t|u)}$, where $u$ denotes the user context. Concretely, given a training instance $(u, x_0)$ in sequential recommendation, we encode the interaction history $u$ with a sequential recommender to obtain a user embedding $\mathbf{u} = \mathrm{SeqRec}(u)$ (instantiated as SASRec [58] in our experiments). We then parameterize a learnable function $s_\Theta$ to estimate the preference ratio for each $y \in \mathcal{X}$ at time $t$:

$$s_\Theta(x_t, t, u)_y = \mathbf{y}^\top \mathbf{MLP}\big(\mathrm{concat}(\mathbf{x}_t, \mathbf{t}, \mathbf{u})\big), \quad y \in \mathcal{X}, \tag{12}$$

where $\mathbf{y}$ and $\mathbf{x}_t$ are the embeddings of item $y$ and the faded item $x_t$, respectively, and $\mathbf{t}$ is the embedding of timestep $t$. For the training objective, we adopt the well-known score entropy loss for discrete diffusion modeling [38], as defined in Equations (3) and (4). To legitimately employ it, we first verify that PreferGrow satisfies the applicability conditions of score entropy loss.

**Proposition 2.** *The preference-fading discrete diffusion* $\{\mathbf{P}_{t|s}\}_{0 \leq s \leq t \leq T}$ *with an idempotent fading matrix* $\mathbf{E}$ *satisfies the Kolmogorov forward equation:*

$$\frac{\partial \mathbf{P}_{t|s}}{\partial t} \;=\; \mathbf{Q}_t \, \mathbf{P}_{t|s}. \tag{13}$$

*Here the rate matrix* $\mathbf{Q}_t$ *and* $\alpha_t$ *are defined succinctly by*

$$\mathbf{Q}_t := \lim_{s \to t} \frac{\partial \mathbf{P}_{t|s}}{\partial t} = \beta(t)\,(\mathbf{E} - \mathbf{I}), \quad \alpha_t := \exp\Big(-\int_0^t \beta(\tau)\,\mathrm{d}\tau\Big), \tag{14}$$

*with* $\beta(\tau) > 0$ *and* $\mathbf{I}$ *the identity matrix.*

To better explain the effectiveness of the Score Entropy (SE) loss in recommendation, we theoretically analyze its connection to the Binary Cross-Entropy (BCE) loss [56].

**Proposition 3.** *Define the soft label* $\pi_{y \succ x_t | x_0} = p(y \succ x_t \mid x_0) = \sigma(r_t(x_0, x_t, y))$ *as in Equation* (1), *where* $\sigma$ *denotes the sigmoid function, and use it as the label for the BCE loss. Treating* $\sigma(s_\Theta(x_t, t, u)_y)$ *as the prediction for the BCE loss yields the soft-label BCE objective:*

$$\mathcal{L}_{sBCE} = -\pi_{y \succ x_t | x_0} \log \sigma(s_\Theta(x_t, t, u)_y) - (1 - \pi_{y \succ x_t | x_0}) \log(1 - \sigma(s_\Theta(x_t, t, u)_y)).$$

*From a gradient perspective, the SE loss and the soft-label BCE loss are related by:*

$$\nabla_{s_\Theta} \mathcal{L}_{SE} = (1 + e^{s_\Theta(x_t, t, u)_y})(1 + e^{r_t(x_0, x_t, y)}) \nabla_{s_\Theta} \mathcal{L}_{sBCE}. \tag{15}$$

Consequently, the SE loss and the soft-label BCE loss share the same descent direction (up to a positive scalar) and attain the same optima. Hence, although originating from discrete diffusion modeling, SE loss is equally well suited to ranking-oriented recommendation tasks as the BCE loss.

## 3.3 Backward Generation: Preference Growing from Preference Ratios

After estimating the preference ratios via score entropy, we reverse the preference fading process to grow user preferences from the non-preference state. We first demonstrate that this reverse process of preference fading, referred to as the preference growing process, is Markovian and satisfies the Kolmogorov backward equation. Together with Proposition 2, this implies that PreferGrow with an idempotent fading matrix inherits the key properties of previous discrete diffusion models [51, 38].

**Theorem 2.** *If the preference fading process $\{\mathbf{P}_{t|0}\}_{t=0}^{T}$ with an idempotent fading matrix $\mathbf{E}$ converges to a unified non-preference state $\vec{p}_T$, the reverse preference growing process $\{\mathbf{P}_{s|T}\}_{s=T}^{0}$ holds:*

- *Markov property:* $\{\mathbf{P}_{s|T}\}_{s=T}^{0}$ *is also a Markov process:*

$$\mathbf{P}_{s|t} = \mathbf{P}_{t|s}^{-1} \cdot \left[ \vec{p}_t \cdot \left( \frac{1}{\vec{p}_t} \right)^{\top} \right] \odot \mathbf{P}_{t|s}^{\top}, \quad \forall 0 \leq s \leq t \leq T. \tag{16}$$

- *Kolmogorov backward equation:* *Preference growing* $\{\mathbf{P}_{s|T}\}_{s=T}^{0}$ *satisfies:*

$$\frac{\partial \mathbf{P}_{s|t}}{\partial t} = \mathbf{R}_s \mathbf{P}_{s|t}, \quad \forall 0 \leq s \leq t \leq T. \tag{17}$$

$$\mathbf{R}_s := \lim_{t \to s} \frac{\partial \mathbf{P}_{s|t}}{\partial s} = \mathbf{Q}_s^{\top} \odot \left[ \vec{p}_s \cdot \left( \frac{1}{\vec{p}_s} \right)^{\top} \right] - \mathbf{Q}_s \cdot \left[ \vec{p}_s \cdot \left( \frac{1}{\vec{p}_s} \right)^{\top} \right] \odot \mathbf{I}. \tag{18}$$

*The matrix $\vec{p}_t \cdot \left( \frac{1}{\vec{p}_t} \right)^{\top}$ denotes exponential preference ratios with entries given by $\frac{p_t(y|u)}{p_t(x_t|u)} \forall x_t, y \in \mathcal{X}$.*

At this point, we are equipped with the approximated preference ratios $\mathbf{s}_{\Theta}$ to grow user preferences. Given a user condition $u$, we first sample an item $x_T$ from the non-preference state $\vec{p}_T$, and then compute the reverse transition matrix $\mathbf{P}_{s|t}$ to progressively grow user preference over time as:

$$p_{s|t}(x_s = y|x_t) = p_{t|s}(x_t|x_s = y) \cdot \sum_{z \in \mathcal{X}} p_{t|s}^{-1}(x_t = y|x_s = z) \cdot e^{\mathbf{s}_{\Theta}(x_t,t,u)_z}, \forall 0 \leq s \leq t \leq T. \tag{19}$$

As illustrated in Figure 2, we iteratively grow user preferences backward until reaching $x_0$. Finally, based on the preference scores at timestep 0, we recommend the top-$K$ items with the highest scores.

## 3.4 Modeling Non-preference User for Personalization

### 3.4.1 Non-preference User Modeling

To enhance the personalization of preference ratios, we introduce the modeling of non-preference users—*i.e.,* cold-start users with no interaction history—during training. This allows the model to perform personalized contrast against the non-preference user during inference, thereby reinforcing user-specific signals for personalized recommendations. As illustrated in Figure 2, we randomly drop the user condition $u$ with a fixed probability $p$ during training, and replace it with a learnable embedding that represents the non-preference user $\phi$.

### 3.4.2 Personalized Contrast against Non-preference User

The non-preference user $\phi$ is defined such that $p_{t|0}(x_t|x_0, \phi) = p_{t|0}(x_t|x_0)$. According to Bayes' theorem $p_{t|0}(x_t|x_0, u) = \frac{p_{t|0}(x_t|x_0, \phi)}{p(u|x_0)} \cdot p(u|x_0, x_t)$, there holds the following ratio condition:

$$\frac{p_{t|0}(x_t = y|x_0, u)}{p_{t|0}(x_t = x|x_0, u)} = \frac{p_{t|0}(x_t = y|x_0)}{p_{t|0}(x_t = x|x_0)} \cdot \frac{p(u|x_0, x_t = y)}{p(u|x_0, x_t = x)}. \tag{20}$$

We then estimate the ratio of the likelihood with a personalization strength parameter $w$ as [59]:

$$\frac{p(u|x_0, x_t = y)}{p(u|x_0, x_t = x)} \approx \left\{ \frac{p_{t|0}(x_t = y|x_0, \phi)}{p_{t|0}(x_t = x|x_0, \phi)} \cdot \left[ \frac{p_{t|0}(x_t = y|x_0, u)}{p_{t|0}(x_t = x|x_0, u)} \right]^{-1} \right\}^{w}. \tag{21}$$

Therefore, the personalization-enhanced preference ratios with strength $w$ are computed as follows:

$$\hat{s}_{\Theta}(x_t, t, u)_y = w \cdot s_{\Theta}(x_t, t, u)_y + (1 - w) \cdot s_{\Theta}(x_t, t, \phi)_y. \tag{22}$$

Table 1: Performance comparison across different datasets and baselines.

| Dataset | | | MoviesLens | | Steam | | Beauty | | Toys | | Sports | |
|---|---|---|---|---|---|---|---|---|---|---|---|---|
| Method | | | HR | NDCG | HR | NDCG | HR | NDCG | HR | NDCG | HR | NDCG |
| SASRec | | @5 | 0.0905 | 0.0502 | 0.0321 | 0.0192 | 0.0326 | 0.0221 | 0.0340 | 0.0258 | 0.0154 | 0.0101 |
| | | @10 | 0.1709 | 0.0760 | 0.0572 | 0.0272 | 0.0438 | 0.0257 | 0.0438 | 0.0289 | 0.0230 | 0.0126 |
| | | @20 | 0.2899 | 0.1059 | 0.0957 | 0.0368 | 0.0595 | 0.0297 | 0.0499 | 0.0305 | 0.0298 | 0.0143 |
| Caser | | @5 | 0.0925 | 0.0576 | 0.0258 | 0.0163 | 0.0170 | 0.0106 | 0.0093 | 0.0072 | 0.0051 | 0.0030 |
| | | @10 | 0.1605 | 0.0794 | 0.0446 | 0.0223 | 0.0219 | 0.0122 | 0.0129 | 0.0083 | 0.0093 | 0.0043 |
| | | @20 | 0.2592 | 0.1042 | 0.0736 | 0.0295 | 0.0295 | 0.0141 | 0.0185 | 0.0098 | 0.0149 | 0.0057 |
| GRURec | | @5 | 0.0892 | 0.0551 | 0.0255 | 0.0156 | 0.0130 | 0.0082 | 0.0124 | 0.0078 | 0.0070 | 0.0048 |
| | | @10 | 0.1534 | 0.0757 | 0.0441 | 0.0216 | 0.0188 | 0.0101 | 0.0180 | 0.0092 | 0.0107 | 0.0059 |
| | | @20 | 0.2501 | 0.1000 | 0.0769 | 0.0298 | 0.0313 | 0.0132 | 0.0304 | 0.0117 | 0.0171 | 0.0076 |
| DreamRec | | @5 | 0.0676 | 0.0437 | 0.0109 | 0.0073 | 0.0300 | 0.0247 | 0.0381 | 0.0304 | 0.0095 | 0.0084 |
| | | @10 | 0.1083 | 0.0568 | 0.0157 | 0.0088 | 0.0353 | 0.0265 | 0.0412 | 0.0314 | 0.0112 | 0.0089 |
| | | @20 | 0.1610 | 0.0701 | 0.0218 | 0.0104 | 0.0402 | 0.0277 | 0.0463 | 0.0327 | 0.0149 | 0.0098 |
| PreferDiff | | @5 | 0.0538 | 0.0349 | 0.0167 | 0.0105 | 0.0335 | 0.0272 | 0.0386 | 0.0308 | 0.0168 | 0.0130 |
| | | @10 | 0.0852 | 0.0450 | 0.0297 | 0.0145 | 0.0434 | 0.0304 | 0.0494 | 0.0343 | 0.0211 | 0.0144 |
| | | @20 | 0.1270 | 0.0555 | 0.0526 | 0.0203 | 0.0577 | 0.0340 | **0.0644** | 0.0380 | 0.0256 | 0.0155 |
| DiffRec | | @5 | 0.0266 | 0.0121 | 0.0268 | 0.0143 | 0.0095 | 0.0055 | 0.0053 | 0.0028 | 0.0063 | 0.0035 |
| | | @10 | 0.0889 | 0.0320 | 0.0657 | 0.0268 | 0.0218 | 0.0094 | 0.0153 | 0.0059 | 0.0117 | 0.0052 |
| | | @20 | 0.1768 | 0.0541 | 0.1159 | 0.0394 | 0.0355 | 0.0128 | 0.0226 | 0.0078 | 0.0194 | 0.0072 |
| DDSR | | @5 | 0.0871 | 0.0533 | 0.0252 | 0.0157 | 0.0291 | 0.0217 | 0.0386 | 0.0302 | 0.0146 | 0.0109 |
| | | @10 | 0.1523 | 0.0742 | 0.0437 | 0.0216 | 0.0434 | 0.0262 | 0.0479 | 0.0332 | 0.0213 | 0.0130 |
| | | @20 | 0.2450 | 0.0975 | 0.0736 | 0.0291 | 0.0608 | 0.0306 | 0.0618 | 0.0367 | 0.0298 | 0.0151 |
| PreferGrow | Impr. | | ✓ | ✓ | ✓ | ✓ | ✓ | ✓ | ✓ | ✓ | ✓ | ✓ |
| Hybrid | | @5 | 0.1409 | 0.0911 | **0.0615** | **0.0406** | **0.0420** | **0.0322** | **0.0441** | **0.0331** | **0.0207** | **0.0142** |
| | | @10 | 0.2229 | 0.1174 | **0.0935** | **0.0508** | **0.0532** | **0.0358** | **0.0519** | **0.0356** | **0.0267** | **0.0162** |
| | | @20 | 0.3355 | 0.1458 | **0.1413** | **0.0628** | **0.0708** | **0.0402** | 0.0625 | **0.0383** | **0.0343** | **0.0181** |
| Adaptive | | @5 | **0.1413** | 0.0912 | 0.0583 | 0.0375 | 0.0396 | 0.0310 | 0.0413 | 0.0304 | 0.0168 | 0.0127 |
| | | @10 | **0.2240** | **0.1177** | 0.0914 | 0.0481 | 0.0508 | 0.0347 | 0.0513 | 0.0337 | 0.0207 | 0.0140 |
| | | @20 | **0.3362** | **0.1460** | 0.1387 | 0.0600 | 0.0610 | 0.0373 | 0.0642 | 0.0370 | 0.0267 | 0.0155 |

Legend (left):
- PreferGrow-Adaptive
- w/o PointWise
- w/o Nonpreference
- w/o Nonpreference + PointWise

Legend (right):
- PreferGrow-Hybrid
- w/o PointWise
- w/o Nonpreference
- w/o PairWise
- w/o Nonpreference + PointWise
- w/o Nonpreference + PairWise

(a) Abalation of Adaptive on Steam.

(b) Ablation of Hybrid on Steam.

Figure 3: Ablation Study on Steam, showing the percentage of the relative effectiveness.

# 4 Experiment

We compare PreferGrow with a variety of baselines under the all-ranking evaluation protocol, including classical recommenders (SASRec [58], Caser [60], GRURec [61]), item-level diffusion-based recommenders (DreamRec [9], PreferDiff [27]), and preference score-level diffusion-based recommenders (DiffRec [28], DDSR [35]) across five benchmark datasets. Details of the datasets are provided in Appendix G.1, discussions of the baselines are presented in Appendix G.2, and the training and evaluation settings are described in Appendix G.3.

## 4.1 Overall Comparsion

From the results in Table 1, we observe that PreferGrow consistently outperforms all baselines. We attribute the effectiveness of PreferGrow to three factors: 1) its discrete diffusion process aligns with the discrete nature of the recommendation scenario; 2) the preference fading noise perturbation mechanism, akin to negative sampling; and 3) preference ratios modeling without the simplex

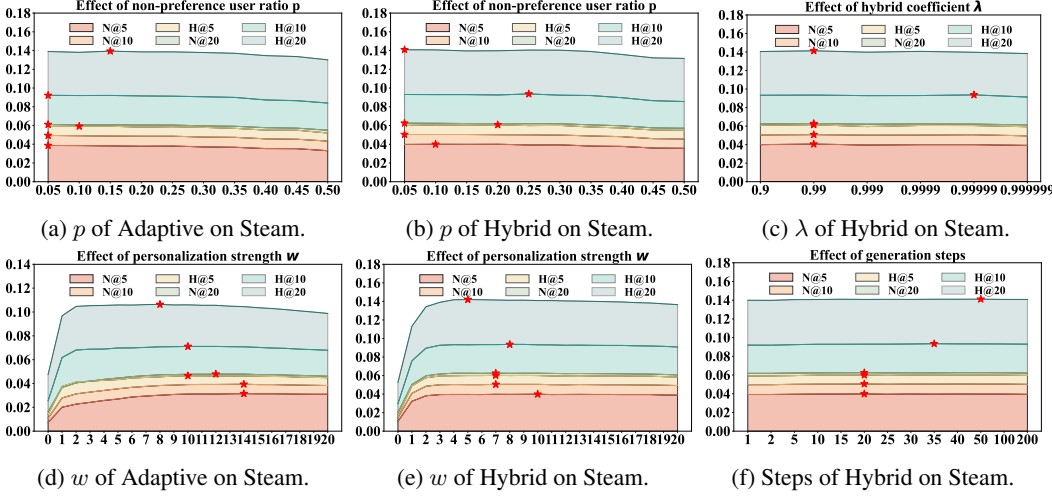

Figure 4: Hyperparameter analysis of PreferGrow on Steam. Red stars denote the best results.

constraints, which better captures user preferences. Additionally, the hybrid and adaptive fading matrix settings, inspired by the negative sampling strategy, also demonstrate the flexibility.

### 4.2 Ablation Study

As presented in Figure 3, we perform a thorough analysis and evaluation of each key component within PreferGrow to assess their individual significance. The ablation study is conducted using the following three variations: (1) *w/o-PointWise*, PreferGrow without the general hard negative item $x_{-1}$; (2) *w/o-PairWise*, Point-Wise PreferGrow with $\vec{p}_T = \vec{e}_{-1}$; (3) *w/o-Nonpreference*, PreferGrow without modeling the non-preference user $\phi$, resulting in no personalized enhancement during the backward generation, as well as the combination of the aforementioned ablations. Overall, PairWise preference ratios modeling and non-preference user modeling are crucial for the effectiveness of PreferGrow, while PairWise preference ratios modeling adds an additional refinement.

### 4.3 Hyper-parameter Analysis

We further investigate the effect of key hyperparameters on the performance of PreferGrow (Figure 4). Overall, PreferGrow is fairly robust to the non-preference user proportion $p$, the hybrid coefficient $\lambda$, and the number of sampling steps, while exhibiting relatively higher sensitivity to the personalization strength $w$. Importantly, $w$ is chosen at inference time, so no retraining is required. On the one hand, $w$ is locally stable within a reasonable range (see Figure 4); on the other hand, we show that the optimal $w$ is highly consistent across data splits (Appendix G.4). Taken together, these properties ensure that tuning $w$ is practical and does not introduce significant overhead.

## 5 Limitations

While effective, PreferGrow still has several aspects that warrant further optimization.

- **Higher modeling complexity.** As shown in Table 8, targeting *preference ratios* incurs substantially higher complexity than prior diffusion-based recommenders. On the one hand, preference ratios are more expressive and thus raise the potential ceiling of the model; on the other hand, they also increase the learning difficulty under finite model capacity—for example, leading to longer training time than previous diffusion-based methods (Appendix G.5). A principled remedy is to extend PreferGrow to a rank-$r$ fading matrix. By inducing a quantization structure, this formulation can markedly reduce the complexity of preference ratios while preserving their expressive power, thus achieving a better expressiveness–redundancy trade-off. We identify this as a primary avenue for future work. In addition, we observe that the final 50% of training yields only about a 5% gain in NDCG@5, suggesting substantial redundancy in the training process. A likely cause is uniform timestep sampling: different timesteps reflect different degrees of user preference fading and thus vary in the difficulty of modeling preference ratios. Incorporating this difficulty — by

beginning with easier timesteps (mild fading) and gradually advancing to harder ones — may further accelerate convergence.

- **High computational complexity.** As shown in Table 8, PreferGrow has $\mathcal{O}(N)$ complexity for both loss computation and inference, where $N$ represents the size of the item corpus. While this is comparable to prior diffusion-based recommenders, it becomes impractical when scaling to extremely large item sets (e.g., billions of items), where even $\mathcal{O}(N)$ is prohibitive. A promising next step is to incorporate a quantization structure such as *Semantic IDs (SIDs)* to reduce the cost from $\mathcal{O}(N)$ to $\mathcal{O}(mc)$. SIDs represent each item with $m$ codebooks of size $c$, enabling up to $c^m$ items while maintaining only $\mathcal{O}(mc)$ computation. Extending PREFERGROW with rank-$r$ fading matrix to operate directly on SIDs is our future work.

## 6 Conclusion

Building upon the previous diffusion-based recommenders and discrete diffusion models discussed in Appendix A, this paper introduce a new discrete diffusion-based recommender tailored for discrete and sparse recommendation scenarios, named PreferGrow. In summary, PreferGrow distinguishes itself from existing diffusion-based recommenders in three key aspects: (1) **Discrete Diffusion:** It operates directly on the discrete item set, fully aligning with the discrete nature of recommendation. (2) **Preference Fading:** It fades user preferences by replacing the preferred item with others, akin to negative sampling, thus removing the need for prior noise assumptions. (3) **Preference Ratios:** It estimates preference ratios by modeling the logarithmic ratios of user–item interaction probabilities, circumventing the constraints of the probability simplex. PreferGrow features a well-defined theoretical formulation and demonstrates superior performance in experiments. We further discuss the broader impacts of PreferGrow in Appendix B.

## Acknowledgments

This research is supported by the National Natural Science Foundation of China (62572449).

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

# A Related Work

Table 2: Comparison of Diffusion-based Recommenders

| | *Modeling Target* | *Forward Perturbation* | *Negative Sampling* | *Representative Work* |
|---|---|---|---|---|
| **Continuous** | Score Function | Gaussian on Embedding | ✗ | DreamRec [9] |
| | Score Function | Gaussian on Embedding | ✓ | PreferDiff [27] |
| | Preference Scores | Gaussian on One-hot | ✓ | DiffRec [28] |
| **Discrete** | Preference Scores | Bernoulli on One-hot | ✓ | RecFusion [62] |
| | Preference Scores | Categorical on One-hot | ✓ | DDSR [35]) |
| | Preference Ratios | Fading on Items | ✓ | **PreferGrow** (Ours) |

**Diffusion-based recommenders** [9–37] utilize forward perturbation in diffusion models to address data sparsity [6], thereby better adapting to sparse recommendation scenarios. They typically consist of three core components: the modeling objective, the forward noise addition, and the corresponding backward generation process, which are summarized in Table 2. Current research on diffusion-based recommenders can be classified into two primary approaches: one involves adding noise to dense item embeddings at the item level, and the other focuses on perturbing the one-hot interaction probability vector for all items, which is referred to as preference scores. The first research line, pioneered by DreamRec [9] and DiffuRec [10], encodes user-preferred items as dense embeddings and adds Gaussian noise to these item embeddings. DreamRec [9] models the score functions (the gradient of the log-likelihood of the perturbed distribution) without negative sampling, whereas DiffuRec [10] incorporates a recommendation loss that includes negative sampling. Building on them, PreferDiff [27] introduced an optimization objective derived from BPR loss [63], which integrates multiple negative samples into the generative modeling framework. However, the application of continuous Gaussian noise to positive preferred items, in contrast to the discrete nature of negative samples, creates an inherent mismatch, making it difficult to optimize both simultaneously during training, leading to a trade-off [10, 64, 27]. [25] Additionally, other works have introduced more sophisticated module designs [11, 12, 14, 16–18, 20, 23, 25, 26] or applied them to different recommendation tasks [13–22, 24]. For instance, DimeRec [64] incorporates multi-interest models, DiQDiff [23] introduces semantic vector quantization, and DiffuASR [13] applies item-level diffusion to sequential recommendation data augmentation. The second research line [28–37] involves converting user preference data into one-hot vectors, which are then mapped to preference scores within the probability simplex. DiffRec [28] add continuous Gaussian noise to the preference scores, and then learn to recover the clean preference scores from the perturbed ones. Consecutively, LD4MRec [30] refines the design for efficient multimedia recommendation, and D3Rec [34] introduces targeted category preferences to control diversity during inference. However, the constraints of the probability simplex—non-negativity and normalization—pose significant challenges in accurately estimating the preference scores [38]. To address these challenges while considering the probability simplex constraints, RecFusion [62] assumes a Bernoulli noise prior, completing the binomial diffusion process and subsequently modeling the parameters of the reverse binomial distribution to facilitate the reverse generation of preference scores. On the other hand, DDSR [35] adopts a categorical noise prior, as proposed in [65], and directly recovers clean preference scores from the perturbed ones. Nonetheless, the constraints of the probability simplex may limit the effectiveness of preference score modeling. Moreover, both diffusion-based recommenders rely on prior noise assumptions, such as Gaussian [9] or Bernoulli noise [29], which may not be optimal for recommendation scenarios where user preference data is inherently discrete.

**Discrete Diffusion Models** [65, 54, 39, 55, 38, 40, 66] have made substantial advances recently. Initially, D3PM [65] proposed a discrete diffusion framework based on an arbitrary probability transition matrix, trained with the evidence lower bound of the log-likelihood. Subsequently, LDR [54] extended this framework to a continuous-time setting using the Kolmogorov forward and backward equations. However, modeling score functions in such models presents challenges, as the gradient of the data distribution is undefined. To address this, CSM [39] introduced Concrete Score—discretization of score functions and the ratios of data distributions—as modeling objectives

for discrete diffusion models. Building upon these advances, SEDD [38] further bridges discrete diffusion models and ConcreteScore by introducing the score entropy loss. Expanding on these developments, we present PreferGrow, a matrix-based discrete diffusion framework which perturbing data by retaining or replacing items within a discrete corpus. The idempotent property of the replacement matrix (or fading matrix) is central to PreferGrow, and we demonstrate that it satisfies the Kolmogorov forward and backward equations as LDR [54], aligning it with prior works. Additionally, we introduce a design paradigm for the idempotent replacement matrix, which unifies previous approaches, including absorbing and uniform settings.

## B  Broader Impacts

Theoretically, PreferGrow introduces a well-defined discrete diffusion model building upon prior work. While designed for recommendation, PreferGrow is also applicable to other discrete domains, such as molecular design in chemistry and protein structure prediction. There are many potential societal consequences of our work, none of which we believe warrant specific attention at this time.

## C  Proofs of Main Results

*Proof of Theorem 1:* $\alpha_t : [0, T] \to [0, 1]$ is a strictly decreasing function with $\alpha_0 = 1$ and $\alpha_T = 0$ and the preference fading discrete diffusion process is denoted as $\mathbf{P}_{t|0} = \alpha_t \mathbf{I} + (1 - \alpha_t)\mathbf{E}, \forall t \in [0, T]$. Then we can rewrite preference fading discrete diffusion process as for $\alpha_0 = 1$:

$$\mathbf{P}_{t|0} = \frac{\alpha_t}{\alpha_0}\mathbf{I} + (1 - \frac{\alpha_t}{\alpha_0})\mathbf{E}, \forall t \in [0, T]. \tag{23}$$

For any well-defined $\mathbf{P}_{t|s,0}$ with $0 < s < t$, the following limiting distribution constraint must hold:

$$p_{t|0}(x_t|x_0) = \sum_{x_s \in \mathcal{X}} p_{t|s,0}(x_t|x_s, x_0)p_{s|0}(x_s|x_0), \forall x_t, x_0 \in \mathcal{X}. \tag{24}$$

In matrix form, this constraint reduces to $\mathbf{P}_{t|0} = \mathbf{P}_{t|s,0}\,\mathbf{P}_{s|0}$ for all $0 < s < t$. Note that one possible particular solution of this equation is:

$$\mathbf{P}_{t|s,0} := \mathbf{P}_{t|s} = \frac{\alpha_t}{\alpha_s}\mathbf{I} + (1 - \frac{\alpha_t}{\alpha_s})\mathbf{E}, \forall 0 \le s \le t \le T. \tag{25}$$

We will show that this indeed satisfies the constraint under the condition that $\mathbf{E}$ is idempotent.

$$\begin{aligned}
\mathbf{P}_{t|s}\mathbf{P}_{s|0} &= \left[\frac{\alpha_t}{\alpha_s}\mathbf{I} + (1 - \frac{\alpha_t}{\alpha_s})\mathbf{E}\right] \cdot \left[\frac{\alpha_s}{\alpha_0}\mathbf{I} + (1 - \frac{\alpha_s}{\alpha_0})\mathbf{E}\right] \\
&= \frac{\alpha_t}{\alpha_0}\mathbf{I} + \left(\frac{\alpha_t}{\alpha_s} - 2\frac{\alpha_t}{\alpha_0} + \frac{\alpha_s}{\alpha_0}\right)\mathbf{E} + \left(1 + \frac{\alpha_t}{\alpha_0} - \frac{\alpha_s}{\alpha_0} - \frac{\alpha_t}{\alpha_s}\right)\mathbf{E}^2 \\
&= \frac{\alpha_t}{\alpha_0}\mathbf{I} + \mathbf{E}^2 - \frac{\alpha_t}{\alpha_0}\mathbf{E} + \left(\frac{\alpha_t}{\alpha_s} - \frac{\alpha_t}{\alpha_0} + \frac{\alpha_s}{\alpha_0}\right)(\mathbf{E} - \mathbf{E}^2) \\
&\qquad \text{the fading matrix is idempotent} \Rightarrow \mathbf{E}^2 = \mathbf{E} \\
&= \frac{\alpha_t}{\alpha_0}\mathbf{I} + (1 - \frac{\alpha_t}{\alpha_0})\mathbf{E} \\
&= \mathbf{P}_{t|0}.
\end{aligned} \tag{26}$$

Similarly, for all $0 \le r \le s \le t \le T$, there holds the Chapman-Kolmogorov equation:

$$\begin{aligned}
\mathbf{P}_{t|s}\mathbf{P}_{s|r} &= \left[\frac{\alpha_t}{\alpha_s}\mathbf{I} + (1 - \frac{\alpha_t}{\alpha_s})\mathbf{E}\right] \cdot \left[\frac{\alpha_s}{\alpha_r}\mathbf{I} + (1 - \frac{\alpha_s}{\alpha_r})\mathbf{E}\right] \\
&= \frac{\alpha_t}{\alpha_r}\mathbf{I} + \mathbf{E}^2 - \frac{\alpha_t}{\alpha_r}\mathbf{E} + \left(\frac{\alpha_t}{\alpha_s} - \frac{\alpha_t}{\alpha_r} + \frac{\alpha_s}{\alpha_r}\right)(\mathbf{E} - \mathbf{E}^2) \\
&\qquad \text{the fading matrix is idempotent} \Rightarrow \mathbf{E}^2 = \mathbf{E} \\
&= \frac{\alpha_t}{\alpha_r}\mathbf{I} + (1 - \frac{\alpha_t}{\alpha_r})\mathbf{E} \\
&= \mathbf{P}_{t|r}.
\end{aligned} \tag{27}$$

We further note that $\mathbf{P}_{t|s}, \forall 0 \le s \le t \le T$ is invertible, with inverse given by:

$$\mathbf{P}_{t|s}^{-1} = \frac{\alpha_s}{\alpha_t}\mathbf{I} + (1 - \frac{\alpha_s}{\alpha_t})\mathbf{E}, \forall 0 \le s \le t \le T. \tag{28}$$

$$\mathbf{P}_{t|s}^{-1}\mathbf{P}_{t|s} = \left[\frac{\alpha_s}{\alpha_t}\mathbf{I} + (1 - \frac{\alpha_s}{\alpha_t})\mathbf{E}\right] \cdot \left[\frac{\alpha_t}{\alpha_s}\mathbf{I} + (1 - \frac{\alpha_t}{\alpha_s})\mathbf{E}\right] \tag{29}$$

$$= \mathbf{I} + \left(\frac{\alpha_t}{\alpha_s} + \frac{\alpha_s}{\alpha_t} - 2\right) \cdot (\mathbf{E} - \mathbf{E}^2)$$

the fading matrix is idempotent $\Rightarrow \mathbf{E}^2 = \mathbf{E}$

$$= \mathbf{I}.$$

$\forall 0 \le r \le s \le t \le T$, combined with limiting distribution constraint $\mathbf{P}_{t|r} = \mathbf{P}_{t|s,r}\mathbf{P}_{s|r}$, there are:

$$\mathbf{P}_{t|s,r} = \mathbf{P}_{t|r}\mathbf{P}_{s|r}^{-1}$$

$$= \left[\frac{\alpha_t}{\alpha_r}\mathbf{I} + (1 - \frac{\alpha_t}{\alpha_r})\mathbf{E}\right] \cdot \left[\frac{\alpha_r}{\alpha_s}\mathbf{I} + (1 - \frac{\alpha_r}{\alpha_s})\mathbf{E}\right] \tag{30}$$

$$= \frac{\alpha_t}{\alpha_s}\mathbf{I} + \mathbf{E}^2 - \frac{\alpha_t}{\alpha_s}\mathbf{E} + \left(\frac{\alpha_t}{\alpha_r} - \frac{\alpha_t}{\alpha_s} + \frac{\alpha_r}{\alpha_s}\right)(\mathbf{E} - \mathbf{E}^2)$$

the fading matrix is idempotent $\Rightarrow \mathbf{E}^2 = \mathbf{E}$

$$= \frac{\alpha_t}{\alpha_s}\mathbf{I} + (1 - \frac{\alpha_t}{\alpha_s})\mathbf{E}$$

$$= \mathbf{P}_{t|s}.$$

$\mathbf{P}_{t|s,r} = \mathbf{P}_{t|s}$ indicates $p_{t|s,r}(x_t|x_s, x_r) = p_{t|s}(x_t|x_s), \forall 0 \le r \le s \le t \le T$. In summary, the preference fading discrete diffusion process is Markovian but not time-homogeneous, satisfies the Chapman–Kolmogorov equation, and is reversible. $\qquad\square$

*Proof of Proposition 1:* $\alpha_t : [0, T] \to [0, 1]$ is a strictly decreasing function with $\alpha_0 = 1$ and $\alpha_T = 0$. $\alpha_t$ is further defined as $e^{-\int_0^t \beta(\tau)\mathrm{d}\tau}$ with $\beta(\tau) > 0$. The preference fading discrete Markov diffusion process is denoted as $\mathbf{P}_{t|s} = \frac{\alpha_t}{\alpha_s}\mathbf{I} + (1 - \frac{\alpha_t}{\alpha_s})\mathbf{E}, \forall 0 \le s \le t \le T$. We first show that the preference fading discrete Markov diffusion process converges to a unified non-preference state $\vec{p}_T$ is well-defined. The transition rate matrix $\mathbf{Q}_t$ is computed as follows:

$$\mathbf{Q}_t = \lim_{s \to t} \frac{\partial \mathbf{P}_{t|s}}{\partial t} \tag{31}$$

$$= \lim_{s \to t} \frac{\partial}{\partial t}\left(\frac{\alpha_t}{\alpha_s}\mathbf{I} + (1 - \frac{\alpha_t}{\alpha_s})\mathbf{E}\right)$$

$$= \lim_{s \to t} \frac{\partial}{\partial \alpha_t}\left(\frac{\alpha_t}{\alpha_s}\mathbf{I} + (1 - \frac{\alpha_t}{\alpha_s})\mathbf{E}\right) \cdot \frac{\partial \alpha_t}{\partial t}$$

$$= \lim_{s \to t} \left(\frac{1}{\alpha_s}\mathbf{I} - \frac{1}{\alpha_s}\mathbf{E}\right) \cdot (-\beta(t)) \cdot \alpha_t$$

$$= \beta(t) \cdot (\mathbf{E} - \mathbf{I}).$$

The rate matrix $\mathbf{Q}_t$ characterizes the velocity of probability transitions at time $t$, encompassing both the direction and rate of transition. As shown in Equation (32), $\mathbf{Q}_t$ at any time $t \in [0, T]$ shares a consistent transition direction $\mathbf{I} - \mathbf{E}$, while the transition rate $\beta(t)$ varies over time. This time-dependent rate results in a non-homogeneous preference fading process. However, the shared transition direction ensures that all diffusion paths converge to the same non-preference state, making the process well-defined. Specifically, the stationary distribution $\vec{\pi}_t$ at time $t$ satisfies the equilibrium condition $\mathbf{Q}_t\vec{\pi}_t = \vec{0}$. Since different $\mathbf{Q}_t$ matrices differ only by a scalar factor $\beta(t)$, they yield the same stationary solution $\vec{\pi}$. Consequently, the Markov process converges to a common steady-state distribution $\vec{\pi}$, *i.e.,* the non-preference state $\vec{p}_T$:

$$\mathbf{Q}_t\vec{\pi} = \vec{0} \Rightarrow (\mathbf{I} - \mathbf{E})\,\vec{p}_T = \vec{0}. \tag{32}$$

Given that $(\mathbf{I} - \mathbf{E})\mathbf{E} = \mathbf{0}$, each column of $\mathbf{E}$ satisfies the non-preference state equation $(\mathbf{I} - \mathbf{E})\,\vec{p}_T = \vec{0}$. To ensure a unique solution $\vec{p}_T$, we therefore assume that all columns of $\mathbf{E}$ are identical, i.e. $\mathbf{E} \propto \vec{p}_T \cdot \vec{1}^\top$. Considering the idempotence constraint $\mathbf{E}^2 = \mathbf{E}$, we then have:

$$\mathbf{E} = \frac{\vec{p}_T \cdot \vec{1}^\top}{\vec{1}^\top \vec{p}_T}. \tag{33}$$

$$\mathbf{E}^2 = \frac{\vec{p}_T \cdot \vec{1}^\top \cdot \vec{p}_T \cdot \vec{1}^\top}{(\vec{1}^\top \vec{p}_T)^2} \tag{34}$$

$$= \frac{\vec{p}_T \cdot (\vec{1}^\top \vec{p}_T) \cdot \vec{1}^\top}{(\vec{1}^\top \vec{p}_T)^2}$$

$$= \frac{\vec{p}_T \cdot \vec{1}^\top}{\vec{1}^\top \vec{p}_T}$$

$$= \mathbf{E}.$$

$\square$

*Proof of Proposition 2:* We have $\mathbf{P}_{t|s} = \frac{\alpha_t}{\alpha_s}\mathbf{I} + (1 - \frac{\alpha_t}{\alpha_s})\mathbf{E}$ and $\mathbf{Q}_t = \beta(t) \cdot (\mathbf{E} - \mathbf{I})$. Then we compute $\frac{\partial \mathbf{P}_{t|s}}{\partial t}$ as follows:

$$\frac{\partial \mathbf{P}_{t|s}}{\partial t} = \frac{\partial}{\partial t}\left(\frac{\alpha_t}{\alpha_s}\mathbf{I} + (1 - \frac{\alpha_t}{\alpha_s})\mathbf{E}\right) \tag{35}$$

$$= \frac{\partial}{\partial \alpha_t}\left(\frac{\alpha_t}{\alpha_s}\mathbf{I} + (1 - \frac{\alpha_t}{\alpha_s})\mathbf{E}\right) \cdot \frac{\partial \alpha_t}{\partial t}$$

$$= \left(\frac{1}{\alpha_s}\mathbf{I} - \frac{1}{\alpha_s}\mathbf{E}\right) \cdot (-\beta(t)) \cdot \alpha_t$$

$$= \beta(t) \cdot \frac{\alpha_t}{\alpha_s} \cdot (\mathbf{E} - \mathbf{I}).$$

There holds the Kolmogorov forward equation:

$$\mathbf{Q}_t \mathbf{P}_{t|s} = \beta(t) \cdot (\mathbf{E} - \mathbf{I}) \cdot \left[\frac{\alpha_t}{\alpha_s}\mathbf{I} + (1 - \frac{\alpha_t}{\alpha_s})\mathbf{E}\right] \tag{36}$$

$$= \beta(t) \cdot \left[-\frac{\alpha_t}{\alpha_s}\mathbf{I} + (2\frac{\alpha_t}{\alpha_s} - 1)\mathbf{E} + (1 - \frac{\alpha_t}{\alpha_s})\mathbf{E}^2\right]$$

$$\text{the fading matrix is idempotent} \Rightarrow \mathbf{E}^2 = \mathbf{E}$$

$$= \beta(t) \cdot \frac{\alpha_t}{\alpha_s} \cdot (\mathbf{E} - \mathbf{I})$$

$$= \frac{\partial \mathbf{P}_{t|s}}{\partial t}.$$

$\square$

*Proof of Proposition 3:* We begin by considering the expression for the loss function $\mathcal{L}_{SE}$, which is given as:

$$\mathcal{L}_{SE}(x_0, x_t, y) = e^{s_\Theta(x_t, t, u)_y} - s_\Theta(x_t, t, u)_y \cdot e^{r_t(x_0, x_t, y)} + e^{r_t(x_0, x_t, y)}(r_t(x_0, x_t, y) - 1). \tag{37}$$

Then, we compute the gradient of this loss with respect to $s_\Theta$, which will help establish a link between this and the binary cross-entropy loss.

$$\nabla_{s_\Theta}\mathcal{L}_{SE} = e^{s_\Theta(x_t, t, u)_y} - e^{r_t(x_0, x_t, y)} \tag{38}$$

$$= e^{s_\Theta(x_t, t, u)_y}(1 + e^{r_t(x_0, x_t, y)}) - (1 + e^{s_\Theta(x_t, t, u)_y})e^{r_t(x_0, x_t, y)}$$

$$= (1 + e^{s_\Theta(x_t, t, u)_y})(1 + e^{r_t(x_0, x_t, y)})[\sigma(s_\Theta(x_t, t, u)_y) - \sigma(r_t(x_0, x_t, y))].$$

Next, we consider the soft binary cross-entropy loss $\mathcal{L}_{sBCE}$, which is defined as:

$$\mathcal{L}_{sBCE}(x_0, x_t, y) = -\pi_{y \succ x_t | x_0} \log \sigma(s_\Theta(x_t, t, u)_y) - (1 - \pi_{y \succ x_t | x_0}) \log(1 - \sigma(s_\Theta(x_t, t, u)_y)). \quad (39)$$

where the soft label $\pi_{y \succ x_t | x_0} = p(y \succ x_t \mid x_0) = \sigma(r_t(x_0, x_t, y))$ is the probability of the preference of $y$ over $x_t$, and $\sigma(\cdot)$ is the sigmoid function.

Now, we compute the gradient of $\mathcal{L}_{sBCE}$ with respect to $s_\Theta$:

$$\begin{aligned}
\nabla_{s_\Theta} \mathcal{L}_{sBCE} &= -\pi_{y \succ x_t | x_0}(1 - \sigma(s_\Theta(x_t, t, u)_y)) + (1 - \pi_{y \succ x_t | x_0})\sigma(s_\Theta(x_t, t, u)_y) \quad (40) \\
&= \sigma(s_\Theta(x_t, t, u)_y) - \pi_{y \succ x_t | x_0} \\
&= \sigma(s_\Theta(x_t, t, u)_y) - \sigma(r_t(x_0, x_t, y)).
\end{aligned}$$

We can now relate the gradients of both loss functions:

$$\nabla_{s_\Theta} \mathcal{L}_{SE} = (1 + e^{s_\Theta(x_t, t, u)_y})(1 + e^{r_t(x_0, x_t, y)})\nabla_{s_\Theta} \mathcal{L}_{sBCE}. \quad (41)$$

$\square$

*Proof of Theorem 2:* The reverse preference growing process is denoted as $\Omega = \{\mathbf{P}_{s|T}\}_{s=T}^0$. A collection $\mathcal{F}$ of subsets of $\Omega$ is called a $\sigma$-algebra on $\Omega$ if it satisfies: 1) $\Omega \in \mathcal{F}$. 2) If $A \in \mathcal{F}$ then its complement $A^c = \Omega \setminus A$ also belongs to $\mathcal{F}$. 3) If $\{A_n\}_{n=1}^\infty \subseteq \mathcal{F}$, then the countable union $\bigcup_{n=1}^\infty A_n \in \mathcal{F}$. We first show that the preference growing process satisfies the Markov property. $\mathcal{F}_t$ is a $\sigma$-algebra of the preference growing process $\Omega_t = \{\mathbf{P}_{s|T}\}_{s=T}^t$. $\forall A \in \mathcal{F}_t$ and $0 \leq s \leq t \leq T$:

$$\begin{aligned}
p_{s|\geq t}(x_s | x_t, A) &= \frac{p_{s,t|>t}(x_s, x_t | A)}{p_{t|>t}(x_t | A)} \cdot \frac{p(A)}{p(A)} \quad (42) \\
&= \frac{p_{s,\geq t}(x_s, x_t, A)}{p_{\geq t}(x_t, A)} \\
&= \frac{p_{>t|t,s}(A | x_t, x_s)}{p_{>t|t}(A | x_t)} \cdot \frac{p_{t|s}(x_t | x_s) p_s(x_s)}{p_t(x_t)}
\end{aligned}$$

the preference fading process is Markovian $\Rightarrow \dfrac{p_{>t|t,s}(A | x_t, x_s)}{p_{>t|t}(A | x_t)} = 1$

$$= \frac{p_s(x_s)}{p_t(x_t)} p_{t|s}(x_t | x_s)$$

the Bayes' theorem $\Rightarrow \dfrac{p_s(x_s)}{p_t(x_t)} p_{t|s}(x_t | x_s) = p_{s|t}(x_s | x_t)$

$$= p_{s|t}(x_s | x_t).$$

We rewrite the equation $p_{s|t}(x_s | x_t) = \frac{p_s(x_s)}{p_t(x_t)} \cdot p_{t|s}(x_t | x_s)$ in matrix form $\forall 0 \leq s \leq t \leq T$:

$$p_{s|t}(x_s | x_t) = \frac{p_s(x_s)}{p_t(x_t)} \cdot p_{t|s}(x_t | x_s). \quad (43)$$

$\vec{p}_s \cdot \left(\dfrac{1}{\vec{p}_t}\right)^\top$ denotes the matrix with entries $\dfrac{p_s(x_s = y)}{p_t(x_t = x)}, \forall x, y \in \mathcal{X}$

$$\mathbf{P}_{s|t} = \left[\vec{p}_s \cdot \left(\frac{1}{\vec{p}_t}\right)^\top\right] \odot \mathbf{P}_{t|s}^\top$$

$$\vec{p}_t = \mathbf{P}_{t|s} \cdot \vec{p}_s \Rightarrow \vec{p}_s = \mathbf{P}_{s|t}^{-1} \cdot \vec{p}_t$$

$$= \mathbf{P}_{t|s}^{-1} \cdot \left[\vec{p}_t \cdot \left(\frac{1}{\vec{p}_t}\right)^\top\right] \odot \mathbf{P}_{t|s}^\top.$$

The reverse-time preference growing process progresses from $s = T$ to $s = 0$, and thus $\alpha_s$ is an increasing function, transitioning from $\alpha_T = 0$ to $\alpha_0 = 1$. We then compute the reverse transition rate matrix $\mathbf{R}_s$ as follows:

$$
\begin{aligned}
\mathbf{R}_s &= \lim_{t \to s} \frac{\partial \mathbf{P}_{s|t}}{\partial s} \\
&= \lim_{t \to s} \frac{\partial \mathbf{P}_{s|t}}{\partial \alpha_s} \cdot \frac{\partial \alpha_s}{\partial s} \quad\quad\quad (44) \\
&= \lim_{t \to s} \frac{\partial}{\partial \alpha_s} \left( \mathbf{P}_{t|s}^{-1} \cdot \left[ \vec{p}_t \cdot \left( \frac{1}{\vec{p}_t} \right)^\top \right] \odot \mathbf{P}_{t|s}^\top \right) \cdot \frac{\partial \alpha_s}{\partial s} \\
&= \lim_{t \to s} \left( \frac{\partial \mathbf{P}_{t|s}^{-1}}{\partial \alpha_s} \cdot \left[ \vec{p}_t \cdot \left( \frac{1}{\vec{p}_t} \right)^\top \right] \odot \mathbf{P}_{t|s}^\top + \mathbf{P}_{t|s}^{-1} \cdot \left[ \vec{p}_t \cdot \left( \frac{1}{\vec{p}_t} \right)^\top \right] \odot \frac{\partial \mathbf{P}_{t|s}^\top}{\partial \alpha_s} \right) \cdot \frac{\partial \alpha_s}{\partial s}
\end{aligned}
$$

$$
\frac{\partial \mathbf{P}_{t|s}^{-1}}{\partial \alpha_s} = \frac{\partial}{\partial \alpha_s} \left( \frac{\alpha_s}{\alpha_t} \mathbf{I} + (1 - \frac{\alpha_s}{\alpha_t}) \mathbf{E} \right) = \frac{1}{\alpha_t} (\mathbf{I} - \mathbf{E})
$$

$$
\lim_{t \to s} \mathbf{P}_{t|s}^{-1} = \lim_{t \to s} \left( \frac{\alpha_s}{\alpha_t} \mathbf{I} + (1 - \frac{\alpha_s}{\alpha_t}) \mathbf{E} \right) = \mathbf{I}
$$

$$
\frac{\partial \mathbf{P}_{t|s}^\top}{\partial \alpha_s} = \frac{\partial}{\partial \alpha_s} \left( \frac{\alpha_t}{\alpha_s} \mathbf{I} + (1 - \frac{\alpha_t}{\alpha_s}) \mathbf{E}^\top \right) = -\frac{\alpha_t}{\alpha_s^2} (\mathbf{I} - \mathbf{E}^\top)
$$

$$
\lim_{t \to s} \mathbf{P}_{t|s}^\top = \lim_{t \to s} \left( \frac{\alpha_t}{\alpha_s} \mathbf{I} + (1 - \frac{\alpha_t}{\alpha_s}) \mathbf{E}^\top \right) = \mathbf{I}
$$

$$
= \lim_{t \to s} \left( \frac{1}{\alpha_t} (\mathbf{I} - \mathbf{E}) \cdot \left[ \vec{p}_t \cdot \left( \frac{1}{\vec{p}_t} \right)^\top \right] \odot \mathbf{I} - \left[ \vec{p}_t \cdot \left( \frac{1}{\vec{p}_t} \right)^\top \right] \odot \frac{\alpha_t}{\alpha_s^2} (\mathbf{I} - \mathbf{E}^\top) \right) \cdot \frac{\partial \alpha_s}{\partial s}
$$

$$
\mathbf{Q}_t = \beta(t) \cdot (\mathbf{E} - \mathbf{I}), \, \mathbf{Q}_t^\top = \beta(t) \cdot (\mathbf{E}^\top - \mathbf{I})
$$

Note that from time $t$ to time $s < t$, $\alpha_s$ is increasing. $\Rightarrow \dfrac{\partial \alpha_s}{\partial s} = \alpha_s \beta(s) > 0$

$$
= \beta(s) \cdot (\mathbf{E}^\top - \mathbf{I}) \odot \left[ \vec{p}_s \cdot \left( \frac{1}{\vec{p}_s} \right)^\top \right] - \beta(s) \cdot (\mathbf{E} - \mathbf{I}) \cdot \left[ \vec{p}_s \cdot \left( \frac{1}{\vec{p}_s} \right)^\top \right] \odot \mathbf{I}
$$

$$
= \mathbf{Q}_s^\top \odot \left[ \vec{p}_s \cdot \left( \frac{1}{\vec{p}_s} \right)^\top \right] - \mathbf{Q}_s \cdot \left[ \vec{p}_t \cdot \left( \frac{1}{\vec{p}_t} \right)^\top \right] \odot \mathbf{I}.
$$

Moreover, we compute $\frac{\partial \mathbf{P}_{t|s}}{\partial s}$ and $\frac{\partial \mathbf{P}_{s|t}}{\partial s}$ as follows:

$$
\frac{\partial \mathbf{P}_{t|s}}{\partial s} = \frac{\partial}{\partial \alpha_s} \left( \frac{\alpha_t}{\alpha_s} \mathbf{I} + (1 - \frac{\alpha_t}{\alpha_s}) \mathbf{E} \right) \cdot \frac{\partial \alpha_s}{\partial s} \quad\quad\quad (45)
$$

Note that from time $t$ to time $s < t$, $\alpha_s$ is increasing. $\Rightarrow \dfrac{\partial \alpha_s}{\partial s} = \alpha_s \beta(s) > 0$

$$
\begin{aligned}
&= \left( -\frac{\alpha_t}{\alpha_s^2} \mathbf{I} + \frac{\alpha_t}{\alpha_s^2} \mathbf{E} \right) \cdot \alpha_s \beta(s) \\
&= \beta(s) \cdot \frac{\alpha_t}{\alpha_s} \cdot (\mathbf{E} - \mathbf{I}) \\
&\quad (\mathbf{E} - \mathbf{I})^2 = \mathbf{E}^2 - 2\mathbf{E} + \mathbf{I} = -(\mathbf{E} - \mathbf{I}), (\mathbf{E} - \mathbf{I})\mathbf{E} = 0 \\
&= \beta(s) \cdot \left[ -\frac{\alpha_t}{\alpha_s} \cdot (\mathbf{E} - \mathbf{I})^2 \right] + \beta(s) \cdot (\mathbf{E} - \mathbf{I})\mathbf{E} \\
&= \left[ \frac{\alpha_t}{\alpha_s} \mathbf{I} + (1 - \frac{\alpha_t}{\alpha_s}) \mathbf{E} \right] \cdot [\beta(s) \cdot (\mathbf{E} - \mathbf{I})] \\
&= \mathbf{P}_{t|s} \mathbf{Q}_s.
\end{aligned}
$$

$$\frac{\partial \mathbf{P}_{s|t}}{\partial s} = \frac{\partial}{\partial s}\left(\mathbf{P}_{t|s}^{-1} \cdot \left[\vec{p}_t \cdot \left(\frac{1}{\vec{p}_t}\right)^\top\right] \odot \mathbf{P}_{t|s}^\top\right) \tag{46}$$

$$= \frac{\partial}{\partial s}\left(\left[\vec{p}_s \cdot \left(\frac{1}{\vec{p}_t}\right)^\top\right] \odot \mathbf{P}_{t|s}^\top\right)$$

$$= \left[\frac{\partial \vec{p}_s}{\partial s} \cdot \left(\frac{1}{\vec{p}_t}\right)^\top\right] \odot \mathbf{P}_{t|s}^\top + \left[\vec{p}_s \cdot \left(\frac{1}{\vec{p}_t}\right)^\top\right] \odot \frac{\partial \mathbf{P}_{t|s}^\top}{\partial s}$$

The reverse time $s$ begins with $s = T$. $\Rightarrow \dfrac{\partial \mathbf{P}_{s|0}}{\partial s} = -\mathbf{Q}_s \cdot \mathbf{P}_{s|0}$

$$\frac{\partial \vec{p}_s}{\partial s} = \frac{\partial \mathbf{P}_{s|0} \cdot \vec{p}_0}{\partial s} = -\mathbf{Q}_s \cdot \mathbf{P}_{s|0} \cdot \vec{p}_0 = -\mathbf{Q}_s \cdot \vec{p}_s$$

$$\frac{\partial \mathbf{P}_{t|s}}{\partial s} = \mathbf{P}_{t|s} \cdot \mathbf{Q}_s \Rightarrow \frac{\partial \mathbf{P}_{t|s}^\top}{\partial s} = (\mathbf{P}_{t|s} \cdot \mathbf{Q}_s)^\top = \mathbf{Q}_s^\top \cdot \mathbf{P}_{t|s}^\top$$

$$= \left[\vec{p}_s \cdot \left(\frac{1}{\vec{p}_t}\right)^\top\right] \odot (\mathbf{Q}_s^\top \cdot \mathbf{P}_{t|s}^\top) - \left[\mathbf{Q}_s \cdot \vec{p}_s \cdot \left(\frac{1}{\vec{p}_t}\right)^\top\right] \odot \mathbf{P}_{t|s}^\top.$$

Finally, we prove that the preference growing process satisfies the Kolmogorov backward equation:

$$\mathbf{R}_s \mathbf{P}_{s|t} = \left\{\mathbf{Q}_s^\top \odot \left[\vec{p}_s \cdot \left(\frac{1}{\vec{p}_s}\right)^\top\right] - \mathbf{Q}_s \left[\vec{p}_t \cdot \left(\frac{1}{\vec{p}_t}\right)^\top\right] \odot \mathbf{I}\right\} \cdot \left\{\left[\vec{p}_s \cdot \left(\frac{1}{\vec{p}_t}\right)^\top\right] \odot \mathbf{P}_{t|s}^\top\right\} \tag{47}$$

The idea behind this step is to prove that the elements at each position of the matrices are identical, thereby establishing their equality.

The details are provided in Equation (48), (49) and (50).

$$= \left\{\left[\vec{p}_s \cdot \left(\frac{1}{\vec{p}_t}\right)^\top\right] \odot (\mathbf{Q}_s^\top \cdot \mathbf{P}_{t|s}^\top)\right\} - \left\{\left[\mathbf{Q}_s \cdot \vec{p}_s \cdot \left(\frac{1}{\vec{p}_t}\right)^\top\right] \odot \mathbf{P}_{t|s}^\top\right\}$$

$$= \frac{\partial \mathbf{P}_{s|t}}{\partial s}.$$

$$\left\{\mathbf{Q}_s^\top \odot \left[\vec{p}_s \cdot \left(\frac{1}{\vec{p}_s}\right)^\top\right]\right\} \cdot \left\{\left[\vec{p}_s \cdot \left(\frac{1}{\vec{p}_t}\right)^\top\right] \odot \mathbf{P}_{t|s}^\top\right\}(x, y) \tag{48}$$

$$= \sum_{z \in \mathcal{X}} q_s(z, x)\frac{p_s(x)}{p_s(z)} \cdot \frac{p_s(z)}{p_t(y)}p_{t|s}(y|z)$$

$$= \frac{p_s(x)}{p_t(y)} \cdot \sum_{z \in \mathcal{X}} q_s(z, x)p_{t|s}(y|z)$$

$$= \left\{\left[\vec{p}_s \cdot \left(\frac{1}{\vec{p}_t}\right)^\top\right] \odot (\mathbf{Q}_s^\top \cdot \mathbf{P}_{t|s}^\top)\right\}(x, y), \forall x, y \in \mathcal{X}.$$

$$\left\{\mathbf{Q}_s \cdot \left[\vec{p}_s \cdot \left(\frac{1}{\vec{p}_s}\right)^\top\right] \odot \mathbf{I}\right\}(x, y) \tag{49}$$

$$= \sum_{z \in \mathcal{X}} q_s(x, z)\frac{p_s(z)}{p_s(y)} \cdot \delta_x(y)$$

$$= \delta_x(y) \cdot \sum_{l \in \mathcal{X}} q_s(x, l)\frac{p_s(l)}{p_s(x)}.$$

$$\left\{ \mathbf{Q}_s \cdot \left[ \vec{p}_s \cdot \left( \frac{1}{\vec{p}_s} \right)^\top \right] \odot \mathbf{I} \right\} \cdot \left\{ \left[ \vec{p}_s \cdot \left( \frac{1}{\vec{p}_t} \right)^\top \right] \odot \mathbf{P}_{t|s}^\top \right\} (x, y) \qquad (50)$$

$$= \sum_{z \in \mathcal{X}} \delta_x(z) \cdot \sum_{l \in \mathcal{X}} q_s(x, l) \frac{p_s(l)}{p_s(x)} \cdot \frac{p_s(z)}{p_t(y)} p_{t|s}(y|z)$$

$$= \sum_{l \in \mathcal{X}} q_s(x, l) \frac{p_s(l)}{p_s(x)} \cdot \frac{p_s(x)}{p_t(y)} p_{t|s}(y|x)$$

$$= p_{t|s}(y|x) \cdot \sum_{l \in \mathcal{X}} q_s(x, l) \frac{p_s(l)}{p_t(y)}$$

$$= \left\{ \left[ \mathbf{Q}_s \cdot \vec{p}_s \cdot \left( \frac{1}{\vec{p}_t} \right)^\top \right] \odot \mathbf{P}_{t|s}^\top \right\} (x, y), \forall x, y \in \mathcal{X}.$$

$\square$

## D  Details of Different Fading Matrix Setting

### D.1  Point-Wise Setting

In the setting of masked discrete diffusion models [40, 52, 53], we can model point-wise preference ratios. Specifically, we introduce an auxiliary general hard negative item $x_{-1}$, which is represented as a learnable embedding. The unified non-preference state corresponds to the general hard negative item $x_{-1}$, that is, $\vec{p}_T = \vec{e}_{-1} \in \mathbb{R}^{N+1}$, where $\vec{e}_{-1}$ denotes the one-hot vector associated with $x_{-1}$. In this case, the reference ratios $r_t(x_0, x_t \in \{x_0, x_{-1}\}, y \in \{x_0, x_{-1}\})$ capture only the relative preference between the positive item $x_0$ and the general hard negative $x_{-1}$. Thus, by using $x_{-1}$ as a common reference, we derive the point-wise preference ratios.

Let $\vec{p}_T = \vec{e}_{-1} \in \mathbb{R}^{N+1}$. Then, the rank-1 fading matrix $\mathbf{E}$ is defined as follows:

$$\mathbf{E} = \begin{pmatrix} 0 & \cdots & 0 & 0 \\ \vdots & \ddots & \vdots & \vdots \\ 0 & \cdots & 0 & 0 \\ 1 & \cdots & 1 & 1 \end{pmatrix}$$

In this case, the reference ratios $r_t(x_0, x_t \in \{x_0, x_{-1}\}, y \in \{x_0, x_{-1}\})$ model only the ratios between the general hard negative $x_{-1}$ and real items $\mathcal{X}$.

$$r_t(x_0, x_t, y) = \begin{cases} r_t(x_0, x_t, x_t) = 0 & \text{if } x_t = y \\ r_t(x_0, x_0, x_{-1}) = \log \frac{1 - \alpha_t}{\alpha_t} & \text{if } y = x_{-1} \text{ and } x_t = x_0 \\ r_t(x_0, x_{-1}, x_0) = \log \frac{\alpha_t}{1 - \alpha_t} & \text{if } y = x_0 \text{ and } x_t = x_{-1} \end{cases}$$

$$\mathbf{Q}_t(x, y) = \beta(t) \cdot (\mathbf{E} - \mathbf{I}) = \begin{cases} -\beta(t) & \text{if } x = y \neq x_{-1} \\ \beta(t) & \text{if } x = x_{-1} \text{ and } x_t \neq x_{-1} \\ 0 & \text{otherwise} \end{cases}$$

We eastimates the preference ratios $\log \frac{p_t(y|u)}{p_t(x_t|u)}$ with $s_\Theta(x_t, t, u)_y$:

$$l_{SE}(x_0, x_t, y|u) = e^{s_\Theta(x_t, t, u)_y} - e^{r_t(x_0, x_t, y)} s_\Theta(x_t, t, u)_y + e^{r_t(x_0, x_t, y)} [r_t(x_0, x_t, y) - 1].$$

For one user preference data $(u, x_0)$, with faded item $x_t$, we compute $\mathcal{L}_{SE}(x_0, x_t, y|u)$:

$$\mathcal{L}_{SE} = \sum_{y \in \{x_0, x_{-1}\}} \mathbf{Q}_t(x_t, y) \cdot l_{SE}(x_0, x_t, y|u)$$

$$= \mathbf{Q}_t(x_t, x_t) \cdot l_{SE}(x_0, x_t, x_t|u) + \sum_{y \in \{x_0, x_{-1}\} \setminus \{x_t\}} \mathbf{Q}_t(x_t, y) \cdot l_{SE}(x_0, x_t, y|u)$$

$$s_\Theta(x_t, t, u)_{x_t} = r_t(x_0, x_t, x_t) = 0 \Rightarrow l_{SE}(x_0, x_t, x_t|u) = 0$$

$$= \sum_{y \in \{x_0, x_{-1}\} \setminus \{x_t\}} \mathbf{Q}_t(x_t, y) \cdot l_{SE}(x_0, x_t, y|u)$$

$$\mathbf{Q}_t(x_t \neq x_{-1}, y \neq x_t) = 0 \Rightarrow \mathcal{L}_{SE}(x_0, x_0, y|u) = 0 \Rightarrow x_t = x_{-1}$$

$$= \mathbf{Q}_t(x_{-1}, x_0) \cdot l_{SE}(x_0, x_{-1}, x_0|u)$$

$$= \beta(t) \cdot \left[ e^{s_\Theta(x_t, t, u)_y} - \frac{\alpha_t}{1 - \alpha_t} \cdot s_\Theta(x_t, t, u)_y + \frac{\alpha_t}{1 - \alpha_t} \left[ \log \frac{\alpha_t}{1 - \alpha_t} - 1 \right] \right].$$

## D.2 Pair-Wise Setting

In the field of natural language processing, uniform discrete diffusion models are generally considered inferior to masked discrete diffusion models [38, 40]. However, in the context of recommendation, uniform discrete diffusion models correspond to pair-wise preference ratios, which resemble random negative sampling, and are in fact superior to point-wise masked discrete diffusion models. Specifically, we define the unified non-preference state as having equal probability over all items, *i.e.,* $\vec{p}_T = \vec{1} \in \mathbb{R}^N$, where $\vec{1}$ denotes the all-ones vector. In this setting, the reference ratios $r_t(x_0, x_t \in \mathcal{X}, y \in \mathcal{X})$ capture the relative preferences among all item pairs.

Let $\vec{p}_T = \vec{1} \in \mathbb{R}^N$, we have fading matrix $\mathbf{E}$ as follows:

$$\mathbf{E} = \begin{pmatrix} \frac{1}{N} & \cdots & \frac{1}{N} \\ \vdots & \ddots & \vdots \\ \frac{1}{N} & \cdots & \frac{1}{N} \end{pmatrix}$$

In this case, the reference ratios $r_t(x_0, x_t \in \mathcal{X}, y \in \mathcal{X})$ model the ratios of all item pairs.

$$r_t(x_0, x_t, y) = \begin{cases} r_t(x_0, x_t, x_t) = 0 & \text{if } x_t = y \\ r_t(x_0, x_0, y \neq x_0) = -\log(1 + N \cdot \frac{\alpha_t}{1 - \alpha_t}) & \text{if } y \neq x_0 \text{ and } x_t = x_0 \\ r_t(x_0, x_t \neq x_0, x_0) = \log(1 + N \cdot \frac{\alpha_t}{1 - \alpha_t}) & \text{if } y = x_0 \text{ and } x_t \neq x_0 \\ r_t(x_0, x_t \neq x_0, y \neq x_0) = 0 & \text{otherwise} \end{cases}$$

$$\mathbf{Q}_t(x, y) = \beta(t) \cdot (\mathbf{E} - \mathbf{I}) = \begin{cases} \beta(t)(\frac{1}{N} - 1) & \text{if } x = y \\ \beta(t) \cdot \frac{1}{N} & \text{otherwise} \end{cases}$$

We eastimates the preference ratios $\log \frac{p_t(y|u)}{p_t(x_t|u)}$ with $s_\Theta(x_t, t, u)_y$:

$$l_{SE}(x_0, x_t, y|u) = e^{s_\Theta(x_t, t, u)_y} - e^{r_t(x_0, x_t, y)} s_\Theta(x_t, t, u)_y + e^{r_t(x_0, x_t, y)} [r_t(x_0, x_t, y) - 1].$$

For one user preference data $(u, x_0)$, with faded item $x_t$, we compute $\mathcal{L}_{SE}(x_0, x_t, y|u)$:

$$\mathcal{L}_{SE} = \sum_{y \in \mathcal{X}} \mathbf{Q}_t(x_t, y) \cdot l_{SE}(x_0, x_t, y|u)$$

$$= \mathbf{Q}_t(x_t, x_t) \cdot l_{SE}(x_0, x_t, x_t|u) + \sum_{y \in \mathcal{X} \setminus \{x_t\}} \mathbf{Q}_t(x_t, y) \cdot l_{SE}(x_0, x_t, y|u)$$

$$s_\Theta(x_t, t, u)_{x_t} = r_t(x_0, x_t, x_t) = 0 \Rightarrow l_{SE}(x_0, x_t, x_t|u) = 0$$

$$= \sum_{y \in \mathcal{X} \setminus \{x_t\}} \mathbf{Q}_t(x_t, y) \cdot l_{SE}(x_0, x_t, y|u)$$

$$\mathbf{Q}_t(x_t, y \neq x_t) = \beta(t) \cdot \frac{1}{N}$$

$$= \beta(t) \cdot \frac{1}{N} \cdot \sum_{y \in \mathcal{X} \setminus \{x_t\}} l_{SE}(x_0, x_{-1}, x_0|u)$$

$$e^{\bar{s}_\Theta(x_t,t,u)} = \frac{1}{N} \cdot \sum_{y \in \mathcal{X}\setminus\{x_t\}} e^{s_\Theta(x_t,t,u)_y} = \frac{1}{N} \cdot \left[ \sum_{y \in \mathcal{X}} e^{s_\Theta(x_t,t,u)_y} - 1 \right]$$

$$\bar{s}_\Theta(x_t,t,u) = \frac{1}{N} \cdot \sum_{y \in \mathcal{X}\setminus\{x_t\}} s_\Theta(x_t,t,u)_y = \frac{1}{N} \cdot \sum_{y \in \mathcal{X}} s_\Theta(x_t,t,u)_y$$

if $x_t = x_0$, denote$\Delta = N \cdot \dfrac{\alpha_t}{1-\alpha_t}$ :

$$= \beta(t) \cdot \left[ e^{\bar{s}_\Theta(x_t,t,u)} - \frac{1}{1+\Delta} \cdot \bar{s}_\Theta(x_t,t,u) - \frac{1}{1+\Delta}(\log(1+\Delta)+1) \right]$$

if $x_t \neq x_0$, denote$\Delta = N \cdot \dfrac{\alpha_t}{1-\alpha_t}$ :

$$= \beta(t) \cdot \left[ e^{\bar{s}_\Theta(x_t,t,u)} - \bar{s}_\Theta(x_t,t,u) - \Delta \cdot s_\Theta(x_t,t,u)_{x_0} + (1+\Delta)(\log(1+\Delta)-1) \right].$$

### D.3  Hybrid-Wise Setting

Note that the point-wise and pair-wise settings are independent of each other. Therefore, we can define a hybrid non-preference state as $\vec{p}_T = \lambda(\vec{1} \in \mathbb{R}^N, 0) + (1-\lambda)\vec{e}_{-1} \in \mathbb{R}^{n+1}$, simultaneously modeling point-wise and pair-wise preference ratios. Since $|\mathcal{X}|$ is typically large, the hybrid coefficient $\lambda$ should be designed as $1 - 10^{-n_\lambda}$, where $n_\lambda \in \mathbb{Z}^+$.

Let $\vec{p}_1 = \lambda(\vec{1}, 0) + (1-\lambda)\vec{e}_{-1} \in \mathbb{R}^{n+1}$, we have fading matrix $\mathbf{E}$ as follows:

$$\mathbf{E} = \begin{pmatrix} \frac{\lambda}{N} & \cdots & \frac{\lambda}{N} & \frac{\lambda}{N} \\ \vdots & \ddots & \vdots & \vdots \\ \frac{\lambda}{N} & \cdots & \frac{\lambda}{N} & \frac{\lambda}{N} \\ 1-\lambda & \cdots & 1-\lambda & 1-\lambda \end{pmatrix}$$

In this case, the reference ratios $r_t(x_0, x_t \in \mathcal{X}, y \in \mathcal{X})$ model the ratios of all item pairs.

$$r_t(x_0, x_t, y) = \begin{cases} 0 & \text{if } x_t = y \\ \log \frac{\alpha_t + (1-\alpha_t)\frac{\lambda}{N}}{(1-\alpha_t)(1-\lambda)} & \text{else if } y = x_0 \text{ and } x_t = x_{-1} \\ \log \frac{\frac{\lambda}{N}}{(1-\lambda)} & \text{else if } y \neq x_0 \text{ and } x_t = x_{-1} \\ \log \frac{(1-\alpha_t)(1-\lambda)}{\alpha_t + (1-\alpha_t)\frac{\lambda}{N}} & \text{else if } y = x_{-1} \text{ and } x_t = x_0 \\ \log \frac{(1-\alpha_t)\frac{\lambda}{N}}{\alpha_t + (1-\alpha_t)\frac{\lambda}{N}} & \text{else if } y \neq x_{-1} \text{ and } x_t = x_0 \\ \log \frac{(1-\alpha_t)(1-\lambda)}{(1-\alpha_t)\frac{\lambda}{N}} & \text{else if } y = x_{-1} \text{ and } x_t \neq \{x_{-1}, x_0\} \\ \log \frac{\alpha_t + (1-\alpha_t)\frac{\lambda}{N}}{(1-\alpha_t)\frac{\lambda}{N}} & \text{else if } y = x_0 \text{ and } x_t \neq \{x_{-1}, x_0\} \\ 0 & \text{else if } y \neq \{x_{-1}, x_0\} \text{ and } x_t \neq \{x_{-1}, x_0\} \end{cases}$$

$$\mathbf{Q}_t(x,y) = \beta(t) \cdot (\mathbf{E} - \mathbf{I}) = \begin{cases} \beta(t)(-\lambda) & \text{if } x = y = x_{-1} \\ \beta(t)(\frac{\lambda}{N} - 1) & \text{if } x = y \neq x_{-1} \\ \beta(t)(1-\lambda) & \text{if } x \neq y \text{ and } x = x_{-1} \\ \beta(t)(\frac{\lambda}{N}) & \text{if } x \neq y \text{ and } x \neq x_{-1} \end{cases}$$

We estimate the preference ratios $\log \frac{p_t(y|u)}{p_t(x_t|u)}$ with $s_\Theta(x_t,t,u)_y$:

$$l_{SE}(x_0, x_t, y|u) = e^{s_\Theta(x_t,t,u)_y} - e^{r_t(x_0,x_t,y)} s_\Theta(x_t,t,u)_y + e^{r_t(x_0,x_t,y)}[r_t(x_0,x_t,y) - 1].$$

For one user preference data $(u, x_0)$, with faded item $x_t$, we compute $\mathcal{L}_{SE}(x_0, x_t, y|u)$:

$$\mathcal{L}_{SE} = \sum_{y \in \mathcal{X}} \mathbf{Q}_t(x_t, y) \cdot l_{SE}(x_0, x_t, y|u)$$

$$= \mathbf{Q}_t(x_t, x_t) \cdot l_{SE}(x_0, x_t, x_t | u) + \sum_{y \in \mathcal{X} \backslash \{x_t\}} \mathbf{Q}_t(x_t, y) \cdot l_{SE}(x_0, x_t, y | u)$$

$$s_\Theta(x_t, t, u)_{x_t} = r_t(x_0, x_t, x_t) = 0 \Rightarrow l_{SE}(x_0, x_t, x_t | u) = 0$$

$$= \sum_{y \in \mathcal{X} \backslash \{x_t\}} \mathbf{Q}_t(x_t, y) \cdot l_{SE}(x_0, x_t, y | u)$$

$$\mathbf{Q}_t(x_t = x_{-1}, y \neq x_t) = \beta(t) \cdot (1 - \lambda), \mathbf{Q}_t(x_t \neq x_{-1}, y \neq x_t) = \beta(t) \cdot (\frac{\lambda}{N})$$

$$= \mathbf{Q}_t(x_t, y \neq x_t) \cdot \sum_{y \in \mathcal{X} \backslash \{x_t\}} l_{SE}(x_0, x_t, y | u)$$

$$e^{\bar{\mathbf{s}}_\Theta(x_t, t, u)} = \frac{1}{N} \cdot \sum_{y \in \mathcal{X} \backslash \{x_t\}} e^{s_\Theta(x_t, t, u)_y} = \frac{1}{N} \cdot \left[ \sum_{y \in \mathcal{X}} e^{s_\Theta(x_t, t, u)_y} - 1 \right]$$

$$\bar{\mathbf{s}}_\Theta(x_t, t, u) = \frac{1}{N} \cdot \sum_{y \in \mathcal{X} \backslash \{x_t\}} s_\Theta(x_t, t, u)_y = \frac{1}{N} \cdot \sum_{y \in \mathcal{X}} s_\Theta(x_t, t, u)_y$$

if $x_t = x_{-1}$, denote $\Delta = N \cdot \frac{\alpha_t}{1 - \alpha_t}$ and $\Lambda = \frac{\lambda \cdot \Delta}{\lambda \cdot \Delta + N}$ :

$$= \beta(t) \cdot \left[ N(1 - \lambda) \cdot e^{\bar{\mathbf{s}}_\Theta(x_t, t, u)} - \lambda \cdot \bar{\mathbf{s}}_\Theta(x_t, t, u) - \frac{\frac{1}{\Lambda} - \lambda}{N} \cdot s_\Theta(x_t, t, u)_{x_0} \right]$$

$$+ \beta(t) \cdot \left\{ \lambda \cdot \left[ 1 + \frac{1}{N} \left( \frac{1}{\Lambda} - 1 \right) \right] \cdot \left[ \log \left( \frac{1}{N} \cdot \frac{\lambda}{1 - \lambda} \right) - 1 \right] - \frac{\log \Lambda}{\Lambda} \right\}$$

if $x_t = x_0$, denote $\Delta = N \cdot \frac{\alpha_t}{1 - \alpha_t}$ and $\Lambda = \frac{\lambda \cdot \Delta}{\lambda \cdot \Delta + N}$ :

$$= \beta(t) \cdot \left[ \lambda \cdot e^{\bar{\mathbf{s}}_\Theta(x_t, t, u)} - \lambda \cdot \Lambda \cdot \bar{\mathbf{s}}_\Theta(x_t, t, u) - (1 - \lambda - \frac{\lambda}{N}) \cdot \Lambda \cdot s_\Theta(x_t, t, u)_{x_{-1}} \right]$$

$$+ \beta(t) \cdot \left\{ (1 - \frac{\lambda}{N}) \cdot \Lambda \cdot (\log \Lambda - 1) - (1 - \lambda) \cdot \Lambda \cdot (\log(\frac{1}{N} \frac{\lambda}{1 - \lambda})) \right\}$$

if $x_t \neq \{x_0, x_{-1}\}$, denote $\Delta = N \cdot \frac{\alpha_t}{1 - \alpha_t}$ and $\Lambda = \frac{\lambda \cdot \Delta}{\lambda \cdot \Delta + N}$ :

$$= \beta(t) \cdot \left[ \lambda \cdot e^{\bar{\mathbf{s}}_\Theta(x_t, t, u)} - \lambda \cdot \bar{\mathbf{s}}_\Theta(x_t, t, u) \right]$$

$$- \beta(t) \cdot \left[ (\frac{1}{N \cdot \Lambda} - \frac{\lambda}{N}) \cdot \Lambda \cdot s_\Theta(x_t, t, u)_{x_0}) + (1 - \lambda - \frac{\lambda}{N}) \cdot s_\Theta(x_t, t, u)_{x_0} \right]$$

$$+ \beta(t) \cdot \left\{ \frac{1}{N \cdot \Lambda} \cdot (\log \Lambda + \log \lambda - 1) - (1 - \lambda) \cdot (1 + \log(\frac{1}{N} \frac{\lambda}{1 - \lambda})) - (1 - \frac{2}{N}) \cdot \lambda \right\}.$$

## D.4 Adative Setting

Furthermore, we can adaptively update the non-preference state $\vec{p}_T$ initialized with above settings. Under adaptive setting, $\vec{p}_T = \vec{\mu} = \text{softmax}(\vec{\theta})$, where $\vec{\theta}$ are learnable parameters.

Let $\vec{p}_T = \vec{\mu} \in \mathbb{R}^N$ or $\mathbb{R}^{N+1}$, with $\sum_{x \in \mathcal{X}} \mu_x = 1$, we have fading matrix $\mathbf{E}$ as follows:

$$\mathbf{E} = \begin{pmatrix} \mu_1 & \cdots & \mu_1 \\ \vdots & \ddots & \vdots \\ \mu_{|\mathcal{X}|} & \cdots & \mu_{|\mathcal{X}|} \end{pmatrix}$$

By arbitrarily specifying a distribution $\vec{\mu}$ satisfying $\sum_{x \in \mathcal{X}} \mu_x = 1$, we can instantiate any desired non-preference state, which is physically associated with different negative sampling strategies. In

this case, the reference ratios $r_t(x_0, x_t \in \mathcal{X}, y \in \mathcal{X})$ are computed as follows:

$$r_t(x_0, x_t, y) = \begin{cases} r_t(x_0, x_t, x_t) = 0 & \text{if } x_t = y \\ r_t(x_0, x_0, y \neq x_0) = -\log(\frac{\alpha_t + (1-\alpha_t)\cdot\mu_{x_t}}{(1-\alpha_t)\cdot\mu_y}) & \text{if } y \neq x_0 \text{ and } x_t = x_0 \\ r_t(x_0, x_t \neq x_0, x_0) = \log(\frac{\alpha_t + (1-\alpha_t)\cdot\mu_{x_t}}{(1-\alpha_t)\cdot\mu_y}) & \text{if } y = x_0 \text{ and } x_t \neq x_0 \\ r_t(x_0, x_t \neq x_0, y \neq x_0) = \log\frac{\mu_y}{\mu_{x_t}} & \text{otherwise} \end{cases}$$

$$\mathbf{Q}_t(x, y) = \beta(t) \cdot (\mathbf{E} - \mathbf{I}) = \begin{cases} \beta(t)(\frac{1}{N} - 1) & \text{if } x = y \\ \beta(t) \cdot \frac{1}{N} & \text{otherwise} \end{cases}$$

We eastimates the preference ratios $\log\frac{p_t(y|u)}{p_t(x_t|u)}$ with $s_\Theta(x_t, t, u)_y$:

$$l_{SE}(x_0, x_t, y|u) = e^{s_\Theta(x_t, t, u)_y} - e^{r_t(x_0, x_t, y)} s_\Theta(x_t, t, u)_y + e^{r_t(x_0, x_t, y)}[r_t(x_0, x_t, y) - 1].$$

For one user preference data $(u, x_0)$, with faded item $x_t$, we compute $\mathcal{L}_{SE}(x_0, x_t, y|u)$:

$$\mathcal{L}_{SE} = \sum_{y \in \mathcal{X}} \mathbf{Q}_t(x_t, y) \cdot l_{SE}(x_0, x_t, y|u)$$

$$= \mathbf{Q}_t(x_t, x_t) \cdot l_{SE}(x_0, x_t, x_t|u) + \sum_{y \in \mathcal{X} \setminus \{x_t\}} \mathbf{Q}_t(x_t, y) \cdot l_{SE}(x_0, x_t, y|u)$$

$$s_\Theta(x_t, t, u)_{x_t} = r_t(x_0, x_t, x_t) = 0 \Rightarrow l_{SE}(x_0, x_t, x_t|u) = 0$$

$$= \sum_{y \in \mathcal{X} \setminus \{x_t\}} \mathbf{Q}_t(x_t, y) \cdot l_{SE}(x_0, x_t, y|u)$$

$$\mathbf{Q}_t(x_t, y \neq x_t) = \beta(t) \cdot \mu_{x_t}$$

$$= \beta(t) \cdot \frac{1}{N} \cdot \sum_{y \in \mathcal{X} \setminus \{x_t\}} l_{SE}(x_0, x_{-1}, x_0|u)$$

$$e^{\bar{s}_\Theta(x_t, t, u)} = \frac{1}{N} \cdot \sum_{y \in \mathcal{X} \setminus \{x_t\}} e^{s_\Theta(x_t, t, u)_y} = \frac{1}{N} \cdot \left[\sum_{y \in \mathcal{X}} e^{s_\Theta(x_t, t, u)_y} - 1\right]$$

$$\tilde{s}_\Theta(x_t, t, u) = \sum_{y \in \mathcal{X} \setminus \{x_t\}} \mu_y \cdot s_\Theta(x_t, t, u)_y = \sum_{y \in \mathcal{X}} \mu_y \cdot s_\Theta(x_t, t, u)_y$$

$$\hat{\mu} = \sum_{y \in \mathcal{X}} \mu_y \cdot \log\mu_y$$

if $x_t = x_0$, denote $\Sigma = \frac{\alpha_t}{1 - \alpha_t}$ :

$$= \beta(t) \cdot \left[\mu_{x_t} \cdot e^{\bar{s}_\Theta(x_t, t, u)} - \frac{1}{\Sigma + \mu_{x_t}} \cdot \tilde{s}_\Theta(x_t, t, u)\right]$$

$$+\beta(t) \cdot \frac{\mu_{x_t}}{\mu_{x_t} + \Sigma} \cdot [\hat{\mu} + (\mu_{x_t} - 1) \cdot (\log(\mu_{x_t} + \Sigma) - 1) - \mu_{x_t} \cdot \log\mu_{x_t}]$$

if $x_t \neq x_0$, denote $\Sigma = \frac{\alpha_t}{1 - \alpha_t}$ :

$$= \beta(t) \cdot \left[\mu_{x_t} \cdot e^{\bar{s}_\Theta(x_t, t, u)} - \tilde{s}_\Theta(x_t, t, u) - \Sigma \cdot s_\Theta(x_t, t, u)_{x_0}\right]$$

$$+\beta(t) \cdot [\hat{\mu} + \mu_{x_t} - (1 + \Sigma)(1 + \log(\mu_{x_t})) + (\Sigma + \mu_{x_0})\log(\Sigma + \mu_{x_0}) - (\mu_{x_0}) \cdot \log(\mu_{x_0})].$$

# E  Rank-$r$ Solution of Fading Matrix E

**Theorem 3** (Nonnegative Idempotent Decomposition). *Let $\mathbf{E} \in \mathbb{R}^{N \times N}$ be entrywise nonnegative and idempotent ($\mathbf{E}^2 = \mathbf{E}$) with $\mathrm{rank}(\mathbf{E}) = r$. Then the following statements are equivalent:*

(A) $\mathbf{E}$ *is nonnegative and idempotent with* $\mathrm{rank}(\mathbf{E}) = r$.

(B) *There exist $r$ nonzero, pairwise disjoint[2] nonnegative vectors $\vec{p}^{1}, \ldots, \vec{p}^{r} \in \mathbb{R}_{\geq 0}^{N}$ and the corresponding support-indicator vectors $\vec{s}^{i} := \mathbf{1}_{\mathrm{supp}(\vec{p}^{i})} \in \{0,1\}^{N}$ such that*

$$\mathbf{E} = \sum_{i=1}^{r} \mathbf{E}_i, \qquad \mathbf{E}_i := \frac{\vec{p}^{i}(\vec{s}^{i})^{\top}}{(\vec{s}^{i})^{\top}\vec{p}^{i}}, \tag{51}$$

*where $(\vec{s}^{i})^{\top}\vec{p}^{i} > 0$ for each $i$ and $\mathrm{supp}(\vec{p}^{i}) := \{\, k \in \{1, \ldots, N\} : p_k^i > 0 \,\}$. Equivalently, for any column index $j \in \{1, \ldots, N\}$,*

$$\mathbf{E}_{:j} = \begin{cases} \vec{p}^{i}, & \text{if } j \in \mathrm{supp}(\vec{p}^{i}) \text{ for some } i \in \{1, \ldots, r\}, \\ \vec{0}, & \text{if } j \notin \bigcup_{i=1}^{r} \mathrm{supp}(\vec{p}^{i}). \end{cases}$$

*Moreover, writing the support indicator as $\vec{1}$ restricted to $\mathrm{supp}(\vec{p}^{i})$, the decomposition (51) admits the compact form*

$$\mathbf{E} = \sum_{i=1}^{r} \frac{\vec{p}^{i}\,\vec{1}_{\mathrm{supp}(\vec{p}^{i})}^{\top}}{\vec{1}_{\mathrm{supp}(\vec{p}^{i})}^{\top}\vec{p}^{i}}, \qquad \vec{p}^{i} \geq 0, \quad (\vec{p}^{i})^{\top}\vec{p}^{j} = 0 \ (i \neq j).$$

**Proof (Sufficiency (B) $\Rightarrow$ (A)).** Entrywise nonnegativity is immediate from $\vec{p}^{i} \geq 0$ and $\vec{s}^{i} \geq 0$. Idempotence follows from disjoint supports: for $i \neq j$ we have $(\vec{s}^{i})^{\top}\vec{p}^{j} = 0$, hence

$$\mathbf{E}_i \mathbf{E}_j = \frac{\vec{p}^{i}(\vec{s}^{i})^{\top}}{(\vec{s}^{i})^{\top}\vec{p}^{i}} \cdot \frac{\vec{p}^{j}(\vec{s}^{j})^{\top}}{(\vec{s}^{j})^{\top}\vec{p}^{j}} = \frac{\vec{p}^{i}\,(\vec{s}^{i})^{\top}\vec{p}^{j}\,(\vec{s}^{j})^{\top}}{(\vec{s}^{i})^{\top}\vec{p}^{i}\,(\vec{s}^{j})^{\top}\vec{p}^{j}} = \mathbf{0}.$$

For $i = j$ we obtain

$$\mathbf{E}_i^2 = \frac{\vec{p}^{i}(\vec{s}^{i})^{\top}\vec{p}^{i}(\vec{s}^{i})^{\top}}{\left((\vec{s}^{i})^{\top}\vec{p}^{i}\right)^2} = \frac{\vec{p}^{i}(\vec{s}^{i})^{\top}}{(\vec{s}^{i})^{\top}\vec{p}^{i}} = \mathbf{E}_i,$$

so $\mathbf{E}^2 = (\sum_i \mathbf{E}_i)^2 = \sum_i \mathbf{E}_i^2 = \sum_i \mathbf{E}_i = \mathbf{E}$. To compute the rank, observe that the column space of $\mathbf{E}_i$ is $\mathrm{span}\{\vec{p}^{i}\}$ and $\mathrm{supp}(\vec{p}^{i})$ are disjoint, hence the $r$ one-dimensional subspaces are independent; therefore $\mathrm{rank}(\mathbf{E}) = \sum_i \mathrm{rank}(\mathbf{E}_i) = r$.

**Proof (Necessity (A) $\Rightarrow$ (B)).** Since $\mathbf{E}$ is idempotent ($\mathbf{E}^2 = \mathbf{E}$), its eigenvalues are 0 or 1, so the image $\mathrm{Im}(\mathbf{E})$ has dimension $r$. Because $\mathbf{E} \geq 0$, every column of $\mathbf{E}$ is either the zero vector or a nonnegative fixed point of $\mathbf{E}$ (indeed, for the $j$-th standard basis vector $\vec{e}_j$, we have $\mathbf{E}(\mathbf{E}\vec{e}_j) = \mathbf{E}\vec{e}_j \geq 0$). Group the nonzero columns of $\mathbf{E}$ by proportionality: put indices $j, k$ in the same class if the $j$-th and $k$-th nonzero columns are positive multiples of each other. This is an equivalence relation, and it produces exactly $r$ classes, say $S_1, \ldots, S_r$, because each class contributes one linearly independent direction in $\mathrm{Im}(\mathbf{E})$. Pick one representative nonzero column from each class and denote it by $\vec{p}^{i} \geq 0$ $(i = 1, \ldots, r)$. By construction, for every $j \in S_i$ the $j$-th column of $\mathbf{E}$ equals $\vec{p}^{i}$, and for $j \notin \bigcup_i S_i$ the column is zero. The classes are disjoint in support by definition. Let $\vec{s}^{i} := \mathbf{1}_{S_i}$ be the indicator of $S_i$. The matrix that places column $\vec{p}^{i}$ on $S_i$ and zeros elsewhere is

$$\mathbf{E}_i = \frac{\vec{p}^{i}(\vec{s}^{i})^{\top}}{(\vec{s}^{i})^{\top}\vec{p}^{i}}, \quad \text{with } (\vec{s}^{i})^{\top}\vec{p}^{i} > 0.$$

Therefore

$$\mathbf{E} = \sum_{i=1}^{r} \mathbf{E}_i,$$

which is exactly the Equation (51). Disjoint supports immediately give $(\vec{p}^{i})^{\top}\vec{p}^{j} = 0$ for $i \neq j$. $\quad\square$

**Physical intuition:**

- **Clustering.** Partition the item set $\mathcal{X}$ $(|\mathcal{X}| = N)$ into $r$ clusters $\mathcal{C}_1, \ldots, \mathcal{C}_r$, where items in the same cluster are more similar. In practice, one may obtain $\{\mathcal{C}_i\}$ via vector-based methods (e.g., $k$-means on semantic embeddings) or via $r$-way classification on the user–item bipartite/weighted sequential graph. Since these constructions require side information beyond our core formulation, a full exploration is deferred to future work.

---

[2]Disjointness means the supports do not overlap: $\mathrm{supp}(\vec{p}^{i}) \cap \mathrm{supp}(\vec{p}^{j}) = \varnothing$ for $i \neq j$.

- **Cluster-wise target distributions.** Define for each cluster a nonnegative target vector

$$\vec{p}_T^{\,i}(x) \;>\; 0 \text{ if } x \in \mathcal{C}_i, \qquad \vec{p}_T^{\,i}(x) \;=\; 0 \text{ otherwise}, \quad i = 1, \dots, r.$$

Because $\{\vec{p}_T^{\,i}\}_{i=1}^r$ have disjoint supports, the resulting fading matrix $\mathbf{E} \;=\; \sum_{i=1}^r \dfrac{\vec{p}_T^{\,i}(\vec{s}^{\,i})^\top}{(\vec{s}^{\,i})^\top \vec{p}_T^{\,i}}$ is idempotent with $\mathrm{rank}(\mathbf{E}) = r$ (Theorem 3), where $\vec{s}^{\,i} = \mathbf{1}_{\mathrm{supp}(\vec{p}_T^{\,i})}$.

- **Effect.** During fading, each preferred item is replaced only by items within the same cluster, which filters out cross-cluster candidates and better respects item heterogeneity. This rank-$r$ structure also improves efficiency. Using the decomposition, for any vector $\mathbf{x} \in \mathbb{R}^N$,

$$\mathbf{E}\mathbf{x} = \sum_{i=1}^r \vec{p}_T^{\,i} \, \frac{(\vec{s}^{\,i})^\top \mathbf{x}}{(\vec{s}^{\,i})^\top \vec{p}_T^{\,i}},$$

so computing $(\vec{s}^{\,i})^\top \mathbf{x}$ costs $\mathcal{O}(|\mathcal{C}_i|)$ and the combination step costs $\mathcal{O}(rN)$, reducing the overall complexity from the naive $\mathcal{O}(N^2)$ to $\mathcal{O}(rN)$.

- **Takeaway.** PreferGrow equipped with a *rank-$r$* fading matrix — i.e., cluster-wise replacement — combines theoretical elegance with practical realism: it confines replacements within semantically coherent groups while offering a scalable $\mathcal{O}(rN)$ implementation.

# F  Algorithms for Training and Inference

---

**Algorithm 1** Training Algorithm of PreferGrow

---

**Input:** user preference data $\mathcal{D} = \{(u, x_0)\}$, non-preference user ratio $p$, retention probability $\alpha_t = e^{-\int_0^t \beta(\tau)\mathrm{d}\tau}$ with $\int_0^t \beta(\tau)\mathrm{d}\tau = (\beta_{\min})^{1-t}(\beta_{\max})^t$ or $\int_0^t \beta(\tau)\mathrm{d}\tau = \log(1-(1-\beta_{\mathrm{scale}} \cdot t))$, and the non-preference state $\vec{p}_T$ for preference fading.
**Output:** estimated *Preference Ratios* $\mathbf{s}_\Theta(x_t, t, u)$ and the non-preference user $\phi$.
    **repeat**
        $(u, x_0) \sim \mathcal{D}$                                                    ▷ preference user-item pair.
        $t \sim \mathrm{Uniform}(\{1, \dots, T\})$                       ▷ sampling timestep $t$ uniformly.
        $u = \phi$ with probability $p$                          ▷ non-preference user modeling.
        $x_t \sim p_{t|0}(\cdot|x_0) = \alpha_t \vec{e}_{x_0} + (1-\alpha_t)\vec{p}_T, x_t \in \mathcal{X}$     ▷ retain or replace for preference fading.
        Compute the score entropy loss $\mathcal{L}_{SE} = \sum_{y \in \mathcal{X}} \mathbf{Q}_t(x_t, y) \cdot l_{SE}(x_0, x_t, y|u)$ as Appendix D.
        Take gradient descent step on $\nabla_\theta \mathcal{L}_{SE}$ and update parameters.
    **until** converged

---

**Algorithm 2** Inference Algorithm of PreferGrow

---

**Input:** user condition $u$, sampling timesteps $S_\tau = \{\tau_i\}_{i=0}^S$ with $\tau_S = T$ and $\tau_0 = 0$, personalization strength $w$, estimated *Preference Ratios* $\mathbf{s}_\theta(x_t, t, u)$ and the non-preference user $\phi$.
**Output:** grown preference scores $p(x_0|u), x_0 \in \mathcal{X}$ of user $u$.
    $x_T = x_{\tau_S} \sim \vec{p}_T, x_T \in \mathcal{X}$                             ▷ the non-preference state.
    **for** $s = S$ to 1 **do**
        $\hat{\mathbf{s}}_\Theta(x_{\tau_s}, \tau_s, u) = (1+w)\mathbf{s}_\Theta(x_{\tau_s}, \tau_s, u) - w \cdot \mathbf{s}_\Theta(x_{\tau_s}, \tau_s, \phi)$ ▷ personalization enhancement.
        $p_{\tau_{s-1}|\tau_s}(x_{\tau_{s-1}}|x_{\tau_s}, u) = p_{\tau_s|\tau_{s-1}}(x_{\tau_s}|x_{\tau_{s-1}}) \cdot \sum_{z \in \mathcal{X}} p_{\tau_s|\tau_{s-1}}^{-1}(x_{\tau_{s-1}}|z) \cdot e^{\hat{\mathbf{s}}_\Theta(x_{\tau_s}, \tau_s, u)_z}$
                ▷ $p_{\tau_s|\tau_{s-1}}(x_{\tau_s}|x_{\tau_{s-1}}) = \frac{\alpha_{\tau_s}}{\alpha_{\tau_{s-1}}}\delta_{x_{\tau_s}}(x_{\tau_{s-1}}) + (1 - \frac{\alpha_{\tau_s}}{\alpha_{\tau_{s-1}}}) \cdot \vec{p}_T(x_{\tau_s})$ as Equation (7).
                ▷ $p_{\tau_s|\tau_{s-1}}^{-1}(x_{\tau_{s-1}}|z) = \frac{\alpha_{\tau_{s-1}}}{\alpha_{\tau_s}}\delta_{x_{\tau_{s-1}}}(z) + (1 - \frac{\alpha_{\tau_{s-1}}}{\alpha_{\tau_s}}) \cdot \vec{p}_T(x_{\tau_{s-1}})$ as Equation (9).
                ▷ $\sum_{z \in \mathcal{X}} p_{\tau_s|\tau_{s-1}}^{-1}(x_{\tau_{s-1}}|z) \cdot e^{\hat{\mathbf{s}}_\Theta(x_{\tau_s}, \tau_s, u)_z} = \frac{\alpha_{\tau_{s-1}}}{\alpha_{\tau_s}} e^{\hat{\mathbf{s}}_\Theta(x_{\tau_s}, \tau_s, u)_{x_{\tau_{s-1}}}}$
                                    $+(1 - \frac{\alpha_{\tau_{s-1}}}{\alpha_{\tau_s}}) \cdot \vec{p}_T(x_{\tau_{s-1}}) \cdot \sum_{z \in \mathcal{X}} e^{\hat{\mathbf{s}}_\Theta(x_{\tau_s}, \tau_s, u)_z}$.
        $x_{\tau_{s-1}} \sim p_{\tau_{s-1}|\tau_s}(x_{\tau_{s-1}}|x_{\tau_s}, u), x_{\tau_{s-1}} \in \mathcal{X}$.             ▷ reverse preference growing.

    **end for**
    **return** $p(x_0|u) = p_{\tau_0|\tau_1}(x_{\tau_0}|x_{\tau_1}, u), x_0 \in \mathcal{X}$            ▷ grown preference scores of user $u$.

---

Table 3: Statistics of datasets after preprocessing.

| Dataset | # users | # items | # Interactions | sparsity |
|---------|---------|---------|----------------|----------|
| Movies | 6040 | 3883 | 1001456 | 04.27% |
| Steam | 39795 | 9265 | 2949605 | 00.80% |
| Beauty | 22,363 | 12,101 | 198,502 | 00.07% |
| Toys | 19,412 | 11,924 | 138,444 | 00.06% |
| Sports | 35,598 | 18,357 | 256,598 | 00.04% |

## G    Experiments Details

### G.1    Datasets

We evaluate PreferGrow on five real-world benchmark datasets:

- **MoviesLens** [67] is a commonly used movie recommendation dataset that contains user ratings, movie titles, and movie genres.

- **Steam** [58] encompasses user reviews for video games on the Steam Store.

- **Beauty** [68] contains movie details and user reviews from Jun 1996 to Sep 2023.

- **Toys** [68] includes user reviews and metadata for toys and games from Jun 1996 to Jul 2014.

- **Sports** [68] comprises reviews and metadata for sports and outdoor products from 1996 to 2014.

Following prior works [9, 27], we adopt the user-splitting strategy, which has been shown to effectively prevent information leakage in test sets [69]. Specifically, we sort all sequences chronologically for each dataset and then split the data into training, validation, and test sets with an 8:1:1 ratio, while preserving the last 10 interactions as the historical sequence. The statistical characteristics of the processed dataset are shown in Table 3. As observed from the table, the recommendation datasets face a significant challenge of severe data sparsity.

### G.2    Baselines

We compare PreferGrow with both traditional discriminative recommenders using negative sampling and diffusion-based generative recommenders, including classical recommenders (SASRec [58], Caser [60], GRURec [61]), item-level diffusion-based recommenders (DreamRec [9], PreferDiff [27]), and preference score-level diffusion-based recommenders (DiffRec [28], DDSR [35]):

- **SASRec** [58] leverages the self-attention mechanism in Transformer to model user preference scores from interaction histories, addressing data sparsity through negative sampling.

- **Caser** [60] utilizes horizontal and vertical convolutional filters to capture sequential patterns at the point-level and union-level, allowing for skip behaviors, and models user preference scores while addressing data sparsity through negative sampling.

- **GRURec** [61] adopts RNNs to model user preference scores from interaction histories, mitigating data sparsity through negative sampling.

- **DreamRec** [9] reshapes sequential recommendation as oracle item generation, addressing data sparsity by adding Gaussian noise to dense item embeddings.

- **PreferDiff** [27] introduces an optimization objective specifically designed for item-level DM-based recommenders, which can integrate multiple negative samples, addressing data sparsity by adding noise to dense item embeddings and using negative sampling both.

- **DiffRec** [28] is a preference score-level diffusion-based generative recommender assuming a Gaussian prior, addressing data sparsity by adding Gaussian noise to preference scores, without considering the constraints of the probability simplex.

- **DDSR** [35] is a preference score-level diffusion-based generative recommender assuming a categorical prior, addressing data sparsity by adding discrete noise to preference scores while respecting the constraints of the probability simplex.

### G.3 Implementation Details

**Training settings**: We implement all models using Python 3.7 and PyTorch 1.12.1 on an Nvidia GeForce RTX 3090. During training, all methods are trained with a fixed batch size of 256 using the Adam optimizer. Additionally, we apply early stopping based on the model's performance on the validation set. To ensure reproducibility, we fix all random seeds to 100 in our main experiments, a randomly chosen value. For the classic recommenders with negative sampling, we employ the binary cross-entropy (BCE) loss. PreferGrow uses SASRec as the encoding model for the user's historical sequence, and we adopt both Hybrid-Wise and Adaptive settings for the fading matrix. For all SASRec modules, we apply RoPE position encoding [70]. The search space of hyperparameters for the baselines is shown in Table 5, and the optimal parameters for our PreferGrow under both hybrid and adaptive settings are presented in Table 6. To ensure the reliability of our findings, we have conducted a comprehensive re-evaluation of our experiments under multiple random seeds $\{100, 200, 300, 400, 500\}$ and statistical significance testing ($p < 0.001$). The results shown in Table 4 consistently confirm the significant performance gains of PreferGrow over baselines.

Table 4: Re-evaluation results (NDCG@5) under multiple random seeds.

| Dataset | Beauty | Toys | Sports | Steam | MovieLens |
|---|---|---|---|---|---|
| **SASRec** | 0.0229±.0020 | 0.0258±.0023 | 0.0104±.0016 | 0.0193±.0002 | 0.0507±.0004 |
| **PreferDiff** | 0.0224±.0031 | 0.0312±.0009 | 0.0122±.0007 | 0.0104±.0004 | 0.0348±.0005 |
| **PreferGrow** | **0.0315±.0010** | **0.0326±.0007** | **0.0146±.0004** | **0.0399±.0004** | **0.0913±.0003** |

**Evaluation Protocols and Metrics**: To ensure a comprehensive evaluation and mitigate potential biases, we adopt the all-rank protocol [71–73, 9, 27], which evaluates recommendations across all items. We employ two widely used ranking-based metrics: *Normalized Discounted Cumulative Gain* (**N**@$K$) and *Mean Reciprocal Rank* (**M**@$K$), to assess the effectiveness of the models.

### G.4 Hyper-parameter Analysis

The personalization strength $w$ is locally stable within a reasonable range and highly consistent across data splits, thus posing no practical challenge for tuning. As shown in Figure 4 of our paper, PreferGrow is sensitive to large-magnitude changes in the personalization strength parameter $w$ (e.g., from 0 to 20). On datasets like *Steam*, we observe that performance remains locally stable within a reasonable range (e.g., $w \in [5, 15]$), but drops notably when $w$ is set too small (*e.g.,* $w = 1$). This highlights the need to set $w$ appropriately for each dataset. To assess sensitivity more systematically, we tested PreferGrow under $w \in \{0, 2, 5, 10\}$ and evaluated the mean absolute error between the optimal $w$ values selected on the training, validation, and test sets. These optimal values are determined by uniformly sampling performance across 40 checkpoints throughout the training process and optimality is defined by maximizing the combined metric $HR@5 + HR@10 + NDCG@5 + NDCG@10$. Our findings in Table 7 show that the optimal $w$ is highly consistent across data splits, indicating that $w$ can be tuned on a validation (or even training) set like other standard hyperparameters. This ensures that tuning $w$ does not pose a practical challenge.

### G.5 Efficiency Analysis

As summarized in Table 9, we report each model's trainable parameters, the number of training epochs, GFLOPs per model output (recall that diffusion models require multiple outputs for denoising), and the total number of outputs.

**Quantitative scalability analysis.** We also provide a quantitative analysis of PreferGrow and clarify the solution outlined in the Limitations. PreferGrow introduces only $\mathcal{O}(N)$ complexity in the network, loss, and inference paths — comparable to contemporary diffusion-based recommenders. Table 8 contrasts model, loss, and inference complexities with representative baselines, where $L$ is the history length, $N$ the item set size, $B$ the number of negatives, $d$ the hidden dimension, and $T$ the number of diffusion steps.

- **Model parameters.** We never materialize the full $N^2$ preference-ratio matrix. Equation (12) requires only $N$ scores; the additional computation cost is thus $\mathcal{O}(Nd)$.

Table 5: Hyperparameters Search Space for Baselines.

| Method | Hyperparameter Search Space |
|---|---|
| **Shared** | lr $\sim$ {1e-2, 1e-3, 1e-4, 1e-5} with decay 0, embedding size $d \sim$ {128, 256, 512} the number of negative sampling (if using) $\sim$ {64, 128, 256}, bath size = 256 |
| **DreamRec** | $w \sim$ {0, 1, 2, 5, 10}, $T \sim$ {500, 1000, 2000,3000}, $p \sim$ {0.05, 0.1, 0.15, 0.2, 0.25, 0.3} |
| **PreferDiff** | $\lambda \sim$ {0.2, 0.4, 0.6, 0.8}, $w \sim$ {0, 1, 2, 5, 10}, $T \sim$ {500, 1000, 2000,3000} |
| **DiffRec** | noise scale $\sim$ {1e-1, 1e-2, 1e-3, 1e-4, 1e-5}, $T \sim$ {2, 5, 20, 50, 100} |
| **DDSR** | $T \sim$ {500, 1000, 2000,3000} |
| **PreferGrow** | $T \sim$ {5, 10, 20, 30, 40}, $p \sim$ {0.05, 0.1, 0.15, 0.2, 0.25, 0.3} $w \sim$ {0, 1, 2, 5, 10}, $\lambda \sim$ {0.9, 0.99, 0.999, 0.9999, 0.99999} |

Table 6: Best Hyperparameters for PreferGrow on five datasets.

| Variant | Dataset | lr | $d$ | $p$ | $T$ | $w$ | $\lambda$ |
|---|---|---|---|---|---|---|---|
| **Hybrid** | **MovieLens** | 1e-4 | 256 | 0.1 | 20 | 10 | 0.9999 |
| | **Steam** | 1e-3 | 256 | 0.1 | 20 | 10 | 0.99999 |
| | **Beauty** | 1e-4 | 256 | 0.1 | 20 | 5 | 0.999 |
| | **Toys** | 1e-3 | 256 | 0.2 | 20 | 10 | 0.9999 |
| | **Sports** | 1e-3 | 256 | 0.2 | 20 | 1 | 0.9999 |

| Variant | Method | lr | $d$ | $p$ | $T$ | $w$ | $x_{-1}$ |
|---|---|---|---|---|---|---|---|
| **Adaptive** | **Dataset** | 1e-4 | 256 | 0.2 | 20 | 10 | True |
| | **Steam** | 1e-3 | 256 | 0.05 | 20 | 10 | True |
| | **Beauty** | 1e-4 | 256 | 0.1 | 20 | 2 | True |
| | **Toys** | 1e-4 | 256 | 0.2 | 20 | 5 | True |
| | **Sports** | 1e-4 | 256 | 0.2 | 20 | 5 | True |

Table 7: Mean absolute error of $w$ across data splits on five datasets.

| Dataset | MovieLens | Steam | Beauty | Sports | Toys |
|---|---|---|---|---|---|
| Mean abs. error of $w$ | 0.000 | 0.000 | 0.778 | 0.050 | 0.775 |

- **Loss computation.** With the idempotent fading matrix $\mathbf{E}$, the objective in Eq. (4) simplifies (Appendix C.1–C.4) to element-wise means, indexing, and a precomputable constant—overall $\mathcal{O}(N)$. The score-entropy loss contains three parts: (i) a positive term and (ii) a negative term (both simple mean/index operations, $\mathcal{O}(N)$), and (iii) a constant term ensuring non-negativity that depends only on $t$ and can be precomputed in $\mathcal{O}(T)$ with $T$=20.

- **Inference cost.** Idempotency yields

$$\mathbf{E}\, s_\Theta(x_t, t, u) \;=\; \left(\frac{\mathbf{p}_T^\top s_\Theta}{\mathbf{1}^\top \mathbf{p}_T}\right)\mathbf{1},$$

reducing Equation (19) from $\mathcal{O}(N^2)$ to $\mathcal{O}(N)$.

**Where $\mathcal{O}(N^2)$ comes from.** The apparent $\mathcal{O}(N^2)$ arises only from the *modeling target*: preference ratios are more expressive than conventional logits, which raises PreferGrow's performance ceiling but does not inflate the network/algorithmic path beyond $\mathcal{O}(N)$.

**Scaling to industry scale.** When $N$ is extremely large (*e.g.,* billions of items), even $\mathcal{O}(N)$ becomes impractical. As discussed in the Limitations, Semantic IDs (SIDs) offer a principled route to reduce complexity. SIDs encode each item with $m$ codebooks of size $c$, enabling up to $c^m$ distinct items while keeping computation at $\mathcal{O}(mc)$; since $c^m \gg N \gg mc$, this yields a compact yet expressive representation space and allows PreferGrow-on-SIDs to reduce cost from $\mathcal{O}(N)$ to $\mathcal{O}(mc)$. A practical blocker is that current SID pipelines (e.g., Tiger [74], ActionPiece [75], DDSR [35], OneRec [76]) are not open-sourced. Truly deploying PreferGrow at industrial scale therefore requires

Table 8: Comparison of model complexities.

| Complexity | Model Parameters | Loss Computation | Modeling Target | Inference |
|---|---|---|---|---|
| SASRec | $\mathcal{O}(Ld^2 + L^2d)$ | $\mathcal{O}(Bd)$ | $\mathcal{O}(N)$ | $\mathcal{O}(Ld^2 + L^2d + Nd)$ |
| DreamRec | $\mathcal{O}((L+3)d^2 + L^2d)$ | $\mathcal{O}(d)$ | $\mathcal{O}(Td)$ | $\mathcal{O}(T(L+3)d^2 + TL^2d + Nd)$ |
| PreferDiff | $\mathcal{O}((L+3)d^2 + L^2d)$ | $\mathcal{O}(Bd)$ | $\mathcal{O}(Td)$ | $\mathcal{O}(T(L+3)d^2 + TL^2d + Nd)$ |
| DiffRec | $\mathcal{O}(LN^2)$ | $\mathcal{O}(N)$ | $\mathcal{O}(TN)$ | $\mathcal{O}(TLN^2)$ |
| DDSR | $\mathcal{O}(Ld^2 + L^2d)$ | $\mathcal{O}(N)$ | $\mathcal{O}(TN)$ | $\mathcal{O}(TLd^2 + TL^2d + TNd)$ |
| PreferGrow | $\mathcal{O}((L+3)d^2 + L^2d)$ | $\mathcal{O}(N+T)$ | $\mathcal{O}(TN^2)$ | $\mathcal{O}(T(L+3)d^2 + TL^2d + TN(d+3))$ |

Table 9: Comparison of efficiency on Steam.

| Models | # Trainable Parameters | # training epochs | Inference GFLOPs | Inference steps |
|---|---|---|---|---|
| SASRec | 2.70M | 61 | 0.85 | 1 |
| Caser | 2.49M | 58 | 0.38 | 1 |
| GRURec | 2.77M | 55 | 4.06 | 1 |
| DreamRec | 7.25M | 52 | 0.85 | 20 |
| PreferDiff | 7.25M | 55 | 0.85 | 20 |
| DiffRec | 18.55M | 1000 | / | 20 |
| DDSR | 3.03M | 142 | 1.77 | 20 |
| PreferGrow | 3.03M | 480 | 0.93 | 20 |

addressing the open challenge of *large-scale multimodal SID pretraining*. Extending PreferGrow to operate over SIDs is our ongoing work.

