# OpenReview forum: "Fading to Grow: Growing Preference Ratios via Preference Fading Discrete Diffusion for Recommendation"
_NeurIPS.cc/2025/Conference — NeurIPS 2025 poster_

### Official Review · Reviewer_rfEc · 2025-07-03

**Clarity:** 3
**Significance:** 1
**Originality:** 3
**Rating:** 4
**Confidence:** 4

**Summary:**

This paper introduces PreferGrow, a novel recommender system designed to overcome data sparsity by using a discrete diffusion model. Unlike other diffusion-based recommenders that rely on continuous noise, PreferGrow operates directly on the discrete set of items. It perturbs data by "fading" user preferences—replacing a liked item with alternatives in a process similar to negative sampling. The model then learns to reconstruct user preferences by "growing" them from these faded states, focusing on relative preference ratios between items rather than absolute scores. This approach, which is theoretically grounded with proofs of its key properties, demonstrates superior performance over existing methods across five datasets

**Questions:**

1. Your analysis shows high sensitivity to the personalization parameter $w$. Could you provide a qualitative analysis showing how the actual recommendations change for a user as $w$ is adjusted?

2. Do you have depolyed your method into real world industrial ?

**Ethical Concerns:**

["NO or VERY MINOR ethics concerns only"]

**Limitations:**

yes

**Quality:**

3

**Strengths And Weaknesses:**

Strengths

1. The paper introduces a novel and significant approach to recommendation systems by leveraging discrete diffusion models. It diverges from the predominant trend of using continuous Gaussian noise in diffusion-based recommenders, which is often a mismatched assumption for discrete item data.

2. The authors conduct extensive experiments on five benchmark datasets, comparing PreferGrow against a variety of classic and state-of-the-art diffusion-based recommenders.

3. The paper is of high quality, grounded in solid theoretical analysis. A key contribution is the development of the  idempotent fading matrix and the formal proof that its idempotence is crucial for ensuring the Markov property and reversibility of the diffusion process.

Weakenesses
1. The most significant weakness, which the authors openly acknowledge, is the model's scalability. The pair-wise setting, which proves most effective, requires modeling preference ratios across all item pairs. This becomes computationally infeasible for recommendation scenarios with extremely large item corpora.

2. The paper notes that PreferGrow incurs longer training times than prior diffusion-based recommenders. This is attributed to the uniform sampling of timesteps during training, where modeling preference ratios at different stages of "fading" can have varying levels of difficulty.

3. The hyperparameter analysis reveals that the model's performance is highly sensitive to the personalization strength parameter $w$. Although the authors frame this positively by noting that $w$ can be tuned during inference without retraining, high sensitivity can still be a practical challenge, potentially requiring careful and costly tuning for each specific application or dataset to achieve optimal results.

---

> ### Author Rebuttal · Authors · 2025-07-31
>
> We sincerely appreciate your thoughtful summary and encouraging evaluation of our work. Your valuable comments regarding the model’s scalability, training efficiency, and sensitivity to the personalization strength parameter $w$ have motivated us to conduct a more in-depth and quantitative analysis of these aspects. We hope that our analysis effectively addresses your concerns.  If you have any further questions or suggestions, we would be glad to discuss them with you.
>
> > Comment 1: model’s scalability
>
> We conduct a **quantitative analysis of PreferGrow's scalability** to address the concerns raised in Comment 1. Additionally, we provide a more detailed explanation of the promising solution mentioned in the Limitations section.
>
> **I. PreferGrow adds only $\mathcal O(N)$ Complexity in network, loss and inference**, comparable to contemporary diffusion-based recommenders. The table below compares model, loss, and inference complexities with other baselines.  L: history length, N: #items, B: #neg samples, d: hidden dim, T: #diffusion steps
>
> |   Complexity   |               Model Parameters                |         Loss Computation          |    Modeling Target    |                       Inference                        |
> | :------------: | :-------------------------------------------: | :-------------------------------: | :-------------------: | :----------------------------------------------------: |
> |   **SASRec**   |    $\mathcal{O}\bigl(Ld^{2}+L^{2}d\bigr)$     | $\mathcal{O}\bigl((B+1)d\bigr)$ |   $\mathcal{O}(N)$    |      $\mathcal{O}\bigl(Ld^{2}+L^{2}d+Nd\bigr)$       |
> |  **DreamRec**  |  $\mathcal{O}\bigl((L+3)d^{2}+L^{2}d\bigr)$   |         $\mathcal{O}(d)$          |   $\mathcal{O}(Td)$   |   $\mathcal{O}\bigl(T(L+3)d^{2}+TL^{2}d+Nd\bigr)$    |
> | **PreferDiff** |  $\mathcal{O}\bigl((L+3)d^{2}+L^{2}d\bigr)$   | $\mathcal{O}\bigl((B+1)d\bigr)$ |   $\mathcal{O}(Td)$   |   $\mathcal{O}\bigl(T(L+3)d^{2}+TL^{2}d+Nd\bigr)$    |
> |  **DiffRec**   |             $\mathcal{O}(LN^{2})$             |         $\mathcal{O}(N)$          |   $\mathcal{O}(TN)$   |                 $\mathcal{O}(TLN^{2})$                 |
> |    **DDSR**    |   $\mathcal{O}\bigl(Ld^{2}+L^{2}d+Nd\bigr)$   |         $\mathcal{O}(N)$          |   $\mathcal{O}(TN)$   |     $\mathcal{O}\bigl(TLd^{2}+TL^{2}d+TNd\bigr)$     |
> | **PreferGrow** | $\mathcal{O}\bigl((L+3)d^{2}+L^{2}d+Nd\bigr)$ |        $\mathcal{O}(N+T)$         | $\mathcal{O}(TN^{2})$ | $\mathcal{O}\bigl(T(L+3)d^{2}+TL^{2}d+TN(d+3)\bigr)$ |
> | **PreferGrow on SIDs** | $\mathcal{O}(m(L{+}3)d^2 {+} mL^2d {+} mcd)$ | $\mathcal{O}(mc+T)$ | $\mathcal{O}(Tmc^2)$ | $\mathcal{O}(Tm(L{+}3)d^2 {+} TmL^2d {+} Tmc(d{+}3))$ |
>
> * **Model Parameters:** We never materialise the full $N^{2}$ preference‑ratio matrix; Equation (14) only requires $N$ scores, so the extra cost is $\mathcal O(Nd)$.
> * **Loss Computation:** Leveraging the idempotent fading matrix $E$, Equation (4) simplifies (see App. C.1–C.4) to element‑wise mean, `indexing`, and a pre‑computable constant—also $\mathcal O(N)$.  The seemingly complex score entropy loss consists of three parts: positive and negative terms (involving simple mean and indexing operations, $\mathcal{O}(N)$), and a constant term (ensuring non-negativity, dependent only on $t$, and pre-computable with $\mathcal{O}(T)$ complexity, $T=20$).
> * **Inference cost:**  Idempotency gives $E\cdot s_{\theta}(x_t,t,u)=\bigl(\mathbf p_T^{\top}s_{\theta}/\mathbf1^{\top}\mathbf p_T\bigr)\mathbf1$ reducing Equation (18) from $\mathcal O(N^{2})$ to $\mathcal O(N)$.
> * **Modeling Target:**  The $\mathcal{O}(N^2)$ complexity only arises from its desired modeling target. This more expressive target contributes to a higher performance ceiling for PreferGrow.
> * **Scalability to extremely large item set (billions of items):** In this case, even $\mathcal{O}(N)$ becomes impractical. As noted in Section 5, possible solutions include sparse matrix operations and reducing the item space to $m$ smaller semantic ID spaces of size $c$.
>
> To better address this concern, we will add an additional appendix section for the analytical computational cost and extend the discussion of the scalability in section 5.
>
> **II. Incorporating Semantic IDs (SIDs) offers a promising next step, further reducing complexity from $\mathcal{O}(N)$ to $\mathcal{O}(mc)$.**   As noted in the Section 5, incorporating Semantic IDs offers a promising direction for further scaling PreferGrow. Semantic IDs (SIDs) represent each item using $m$ codebooks of size $c$, thereby enabling the representation of up to $c^m$ unique items. Since $c^m \gg N \gg mc$, SIDs provide a compact yet expressive representation space. This allows PreferGrow on SIDs to reduce its computational complexity from $\mathcal{O}(N)$ to $\mathcal{O}(mc)$ :
>
> However, existing works on SIDs (Tiger [1], DDSR [2], ActionPiece [3], OneRec [4]) have not been open-sourced. To truly deploy PreferGrow in real-world industrial scenarios, one must also address the **open challenge of large-scale multimodal SIDs pre-training**. Extending PreferGrow to operate on SIDs is our ongoing future work.
>
> We acknowledge that the discussion in the Limitations section was too general. We will re-analyze the limitations quantitatively and include the corresponding details in the Appendix. We hope that this quantitative analysis provides a clearer understanding of PreferGrow's scalability and helps alleviate your concerns.
>
> > Comment 2: training efficiency
>
> **I. After our quantitative analysis, we further attribute the longer training time of PreferGrow to a more fundamental mismatch between the modeling target complexity and the model capacity**. As shown in the complexity table, PreferGrow models an all-pair preference ratio objective with a complexity of $\mathcal{O}(N^2)$, while the model itself has a network complexity of only $\mathcal{O}(d^2 + Nd)$. When $N=10{,}000$ and $d=256$, the modeling target becomes nearly `40×` more complex than the model's representational capacity. We also analyzed training dynamics and observed that the last 50% of training time contributes only `~5%` gain in NDCG@5, indicating redundancy in the training process. This is partially due to uniform timestep sampling. Designing a dynamically adaptive timestep schedule is an exciting direction for our future work.
>
> |  Dataset  |  25%   |    50%     |  60%   |  70%   |  80%   |  90%   |    100%    |
> | :-------: | :----: | :--------: | :----: | :----: | :----: | :----: | :--------: |
> | MovieLens | 0.0791 |   0.0881   | 0.0895 | 0.0905 | 0.0907 | 0.0906 | **0.0911** |
> |   Steam   | 0.0349 |   0.0389   | 0.0392 | 0.0394 | 0.0397 | 0.0400 | **0.0406** |
> |  Beauty   | 0.0301 |   0.0306   | 0.0301 | 0.0306 | 0.0312 | 0.0321 | **0.0322** |
> |   Toys    | 0.0323 | **0.0331** | 0.0316 | 0.0305 | 0.0314 | 0.0303 |   0.0311   |
> |  Sports   | 0.0124 |   0.0122   | 0.0135 | 0.0134 | 0.0133 | 0.0133 | **0.0142** |
>
> **II. Incorporating Semantic IDs (SIDs) offers a promising solution, effectively eliminating the complexity imbalance.**  PreferGrow operates on $m$ codebooks of size $c$, leading to a modeling target complexity of $\mathcal{O}(mc^2)$ and model complexity of $\mathcal{O}(md^2 + mcd)$. When $d = c = 256$, these complexities become well matched, resolving the current imbalance.
>
> > Comment 3: sensitivity to the personalization strength parameter $w$
>
> **The personalization strength $w$ is locally stable within a reasonable range and highly consistent across data splits, thus posing no practical challenge for tuning.** As shown in Figure 4 of our paper, PreferGrow is sensitive to large-magnitude changes in the personalization strength parameter $w$ (e.g., from 0 to 20). On datasets like *Steam*, we observe that performance remains locally stable within a reasonable range (e.g., $w \in [5, 15]$), but drops notably when $w$ is set too small (e.g., $w=1$). This highlights the need to set $w$ appropriately for each dataset. To assess sensitivity more systematically, we tested PreferGrow under $w \in [0, 2, 5, 10]$ and evaluated the mean absolute error between the optimal $w$ values selected on the training, validation, and test sets. These optimal values are determined by uniformly sampling performance across 40 checkpoints throughout the training process and optimality is defined by maximizing the combined metric $HR@5 + HR@10 + NDCG@5 + NDCG@10$. Our findings show that the optimal $w$ is highly consistent across data splits, indicating that $w$ can be tuned on a validation (or even training) set like other standard hyperparameters. This ensures that tuning  $w$ does not pose a practical challenge.
>
> |        Dataset         | MovieLens | Steam | Beauty | Sports | Toys  |
> | :--------------------: | :-------: | :---: | :----: | :----: | :---: |
> | Mean abs. error of opt. $w$ |   0.000   | 0.000 | 0.778  | 0.050  | 0.775 |
>
> > Comment 4: Do you have depolyed your method into real world industrial ?
>
> We are actively working on building high-quality, large-scale multimodal Semantic IDs, which is a key challenge for real-world deployment. We plan to apply **PreferGrow with Semantic IDs** in industrial settings once this foundation is established.
>
> Thank you once again for your valuable feedback. We hope that our efforts during the rebuttal period will be taken into account in your evaluation. We remain open and willing to discuss any new questions or suggestions you might have.
>
> **References**
>
> [1] Rajput, Shashank, et al. "Recommender systems with generative retrieval." NeurIPS 2023.
>
> [2] Xie, Wenjia, et al. "Breaking determinism: Fuzzy modeling of sequential recommendation using discrete state space diffusion model." NeurIPS 2024.
>
> [3] Hou, Yupeng, et al. "ActionPiece: Contextually Tokenizing Action Sequences for Generative Recommendation." ICML 2025.
>
> [4] Deng, Jiaxin, et al. "Onerec: Unifying retrieve and rank with generative recommender and iterative preference alignment." ArXiv 2025.

---

> > ### Comment · Reviewer_rfEc · 2025-08-04
> >
> > Thank you for your response. You have resolved most of my concerns. I will keep the current score.

---

> > > ### Author Response · Authors · 2025-08-04
> > > **Thanks**
> > >
> > > We are deeply grateful for your feedback. Following your insightful suggestions, we carried out additional in-depth analyses of our method, further strengthening the paper’s soundness and completeness. Thanks again for your time and valuable comments. We welcome any additional questions you may have and are more than happy to engage in further discussions.

---

> ### Author Response · Authors · 2025-08-07
> **Improving Realism with Rank-$r$ Fading Matrix (Cluster-wise Replacement)**
>
> We would like to share some recent advancements in our design of the fading matrix, which may help further address concerns regarding realism and model flexibility.
>
> We have already presented the **rank-1** idempotent fading matrix in the paper.
> $\mathbf{E}=\dfrac{\vec{p}_T \mathbf{1}^{\top}}{\mathbf{1}^{\top}\vec{p}_T}$.
>
> ## Extending to rank-$r$ idempotent fading matrix
>
> We now generalize this to a **rank-$r$** idempotent fading matrix
>
> $$
> \mathbf{E}
> =\sum\limits_{i=1}^{r}\frac{\vec{p}_T^{i}\mathbf{1}^{\top}}
>                      {\mathbf{1}^{\top}\vec{p}_T^{i}},
> \quad
> \vec{p}_T^{i}\ge 0,\quad
> (\vec{p}_T^{i})^{\top}\vec{p}_T^{j}=0
> \quad \forall i\neq j.
> $$
>
> ## Physical intuition
>
> * **Clustering:** Partition the item set $\mathcal{I}$ ($|\mathcal{I}|=N$) into $r$ clusters $\{\mathcal{C}_1,\dots,\mathcal{C}_r\}$, where items in the same cluster are more similar.
> > To obtain the item clusters, one may apply K-means or other vector-based clustering methods to semantic embeddings of items; alternatively, $r$-way classification can be performed on the user–item bipartite graph or the weighted sequential interaction graph using GCN or similar approaches. Since these methods rely on additional side information beyond our current framework, we leave a full exploration of cluster construction to future work.
>
> * **Cluster-wise target distributions:**
>   Define
> $$
> \vec{p}_T^{i}(x) > 0 \quad \text{if } x \in \mathcal{C}_i; \quad \vec{p}_T^{i}(x) = 0 \quad \text{otherwise}, \quad i = 1, \dots, r.
> $$
>
>
>   Because $\vec{p}_T^{~i},\forall i= 1,\cdots,r$ are disjoint, the resulting $\mathbf{E}$ is idempotent with rank $r$.
>
> * **Effect:** During fading, each preferred item is replaced **only by items within the same cluster**, filtering out cross-cluster candidates and better respecting item heterogeneity. Hence the rank-$r$ fading matrix offers improved realism. Furthermore, rank-$r$ idempotency also reduces the cost of  $\mathbf{E} \cdot \mathbf{x} = \left(\sum\limits_{i=1}^{r}\frac{\mathbf{x}^{\top}\vec{p}_T^{i}}{\mathbf{1}^{\top}\vec{p}_T^{i}}\right) \vec{1}$  from $\mathcal{O}(N^2)$ to $\mathcal{O}(rN)$.
>
>
> We believe that **PreferGrow equipped with a rank-$r$ fading matrix — cluster-wise replacement —combines theoretical elegance with practical realism**.  We will add this rank-$r$ extension as an extra appendix section and cite it in the main text.  We hope this addresses your concerns about realism and inspires further research and applications.

---

### Official Review · Reviewer_XkWC · 2025-07-03

**Clarity:** 3
**Significance:** 3
**Originality:** 3
**Rating:** 5
**Confidence:** 3

**Summary:**

This paper introduces PreferGrow, a discrete diffusion-based recommender system that models preference ratios by a novel process of preference fading and growing, specifically tailored for the discrete nature of user preference data. Unlike prior diffusion-based recommenders that rely on continuous Gaussian noise, PreferGrow directly operates on discrete items using a theoretically grounded matrix-based framework. The authors claim that this approach not only better captures the discrete nature of recommendation tasks but also improves empirical performance across multiple datasets.

**Questions:**

NA

**Ethical Concerns:**

["NO or VERY MINOR ethics concerns only"]

**Limitations:**

Yes

**Quality:**

3

**Strengths And Weaknesses:**

Strength

a) The paper offers solid theoretical backing, proving Markov properties, reversibility, and the validity of the fading process using an idempotent fading matrix.

b) Extensive experiments across five benchmark datasets consistently show PreferGrow outperforming classical, item-level, and score-level diffusion-based recommenders.

Weakness
The approach could face significant computational challenges when scaling to very large item sets due to the all-pair preference ratio modeling.

---

> ### Author Rebuttal · Authors · 2025-07-31
>
> We sincerely thank you for your positive and constructive feedback. We greatly appreciate your recognition of our theoretical contributions, the novelty of the preference fading–growing framework, and the strong empirical results across multiple benchmarks. In response to your concerns, we have conducted a more in-depth and quantitative analysis of the model's scalability. We hope this additional analysis effectively addresses your concerns. Should you have any further questions or suggestions, we would be more than happy to engage in further discussion.
>
> > Comment 1: significant computational challenges when scaling to very large item sets due to the all-pair preference ratio modeling.
>
> We conduct a **quantitative analysis of PreferGrow's scalability** to address your concerns raised in Comment 1. Additionally, we provide a more detailed explanation of the promising solution mentioned in the Limitations section.
>
> **I. PreferGrow adds only $\mathcal O(N)$ Complexity in network, loss and inference**, comparable to contemporary diffusion-based recommenders. The table below compares model, loss, and inference complexities with other baselines.  L: history length, N: #items, B: #neg samples, d: hidden dim, T: #diffusion steps
>
> |   Complexity   |               Model Parameters                |        Loss Computation         |    Modeling Target    |                      Inference                       |
> | :------------: | :-------------------------------------------: | :-----------------------------: | :-------------------: | :--------------------------------------------------: |
> |   **SASRec**   |    $\mathcal{O}\bigl(Ld^{2}+L^{2}d\bigr)$     | $\mathcal{O}\bigl((B+1)d\bigr)$ |   $\mathcal{O}(N)$    |      $\mathcal{O}\bigl(Ld^{2}+L^{2}d+Nd\bigr)$       |
> |  **DreamRec**  |  $\mathcal{O}\bigl((L+3)d^{2}+L^{2}d\bigr)$   |        $\mathcal{O}(d)$         |   $\mathcal{O}(Td)$   |   $\mathcal{O}\bigl(T(L+3)d^{2}+TL^{2}d+Nd\bigr)$    |
> | **PreferDiff** |  $\mathcal{O}\bigl((L+3)d^{2}+L^{2}d\bigr)$   | $\mathcal{O}\bigl((B+1)d\bigr)$ |   $\mathcal{O}(Td)$   |   $\mathcal{O}\bigl(T(L+3)d^{2}+TL^{2}d+Nd\bigr)$    |
> |  **DiffRec**   |             $\mathcal{O}(LN^{2})$             |        $\mathcal{O}(N)$         |   $\mathcal{O}(TN)$   |                $\mathcal{O}(TLN^{2})$                |
> |    **DDSR**    |   $\mathcal{O}\bigl(Ld^{2}+L^{2}d+Nd\bigr)$   |        $\mathcal{O}(N)$         |   $\mathcal{O}(TN)$   |     $\mathcal{O}\bigl(TLd^{2}+TL^{2}d+TNd\bigr)$     |
> | **PreferGrow** | $\mathcal{O}\bigl((L+3)d^{2}+L^{2}d+Nd\bigr)$ |       $\mathcal{O}(N+T)$        | $\mathcal{O}(TN^{2})$ | $\mathcal{O}\bigl(T(L+3)d^{2}+TL^{2}d+TN(d+3)\bigr)$ |
> | **PreferGrow on SIDs** | $\mathcal{O}(m(L{+}3)d^2 {+} mL^2d {+} mcd)$ | $\mathcal{O}(mc+T)$ | $\mathcal{O}(Tmc^2)$ | $\mathcal{O}(Tm(L{+}3)d^2 {+} TmL^2d {+} Tmc(d{+}3))$ |
>
>
> * **Model Parameters:** We never materialise the full $N^{2}$ preference‑ratio matrix; Equation (14) only requires $N$ scores, so the extra cost is $\mathcal O(Nd)$.
> * **Loss Computation:** Leveraging the idempotent fading matrix $E$, Equation (4) simplifies (see App. C.1–C.4) to element‑wise mean, `indexing`, and a pre‑computable constant—also $\mathcal O(N)$.  The seemingly complex score entropy loss consists of three parts: positive and negative terms (involving simple mean and indexing operations, $\mathcal{O}(N)$), and a constant term (ensuring non-negativity, dependent only on $t$, and pre-computable with $\mathcal{O}(T)$ complexity, $T=20$).
> * **Inference cost:**  Idempotency gives $E\cdot s_{\theta}(x_t,t,u)=\bigl(\mathbf p_T^{\top}s_{\theta}/\mathbf1^{\top}\mathbf p_T\bigr)\mathbf1$ reducing Equation (18) from $\mathcal O(N^{2})$ to $\mathcal O(N)$.
> * **Modeling Target:** The $\mathcal{O}(N^2)$ complexity only arises from its desired modeling target. This more expressive target contributes to a higher performance ceiling for PreferGrow.
> * **Scalability to extremely large item set (billions of items):** In this case, even $\mathcal{O}(N)$ becomes impractical. As we have mentioned in section 5, there might exists several alternatives that can mitigate this challenge including but not limited to sparse matrix operations and reducing the item ID space into $m$ smaller (size $c$) semantic ID spaces. And for the latter PreferGrow with SID alternative, we provide the analytical result in the last line in the table.
>
> To better address this concern, we will add an additional appendix section for the analytical computational cost and extend the discussion of the scalability in section 5.
>
> **II. Incorporating Semantic IDs is a promising next step for PreferGrow.**  As noted in the Section 5 Limitations, incorporating Semantic IDs offers a promising direction for further scaling PreferGrow. Semantic IDs (SIDs) represent each item using $m$ codebooks of size $c$, thereby enabling the representation of up to $c^m$ unique items. Since $c^m \gg N \gg mc$, SIDs provide a compact yet expressive representation space. This allows PreferGrow on SIDs to reduce its computational complexity from $\mathcal{O}(N)$ to $\mathcal{O}(mc)$ .
>
> However, existing works on SIDs — such as **Tiger [1], ActionPiece [2], DDSR [3] and OneRec [4]** — have not been open-sourced. To truly deploy PreferGrow in real-world industrial scenarios, one must also address the **open challenge of large-scale multimodal SIDs pre-training**. Extending PreferGrow to operate on SIDs is our ongoing future work.
>
> We acknowledge that the discussion in the Limitations section was too general. We will re-analyze the limitations quantitatively and include the corresponding details in the Appendix. We hope that this quantitative analysis provides a clearer understanding of PreferGrow's scalability and helps alleviate your concerns regarding its potential computational challenges.
>
> Thank you once again for your valuable feedback.  We remain receptive and ready to engage with any further questions or suggestions you may have.
>
> **References**
>
> [1] Rajput, Shashank, et al. "Recommender systems with generative retrieval." NeurIPS 2023.
>
> [2] Xie, Wenjia, et al. "Breaking determinism: Fuzzy modeling of sequential recommendation using discrete state space diffusion model." NeurIPS 2024.
>
> [3] Hou, Yupeng, et al. "ActionPiece: Contextually Tokenizing Action Sequences for Generative Recommendation." ICML 2025.
>
> [4] Deng, Jiaxin, et al. "Onerec: Unifying retrieve and rank with generative recommender and iterative preference alignment." ArXiv 2025.

---

> ### Author Response · Authors · 2025-08-07
> **Reminder for the End of Rebuttal Period**
>
> We would like to express our gratitude once more for your positive and constructive feedback. This message
> serves as a friendly reminder as we approach the conclusion of the discussion phase. We sincerely hope that the
> improvements during rebuttal will be taken into consideration.
>
> We also would like to share some recent advancements in our design of the fading matrix, which may help further address concerns regarding realism and model flexibility.
>
> ## **Improving Realism with Rank-$r$ Fading Matrix (Cluster-wise Replacement)**
>
> We have already presented the **rank-1** idempotent fading matrix in the paper.
> $\mathbf{E}=\dfrac{\vec{p}_T \mathbf{1}^{\top}}{\mathbf{1}^{\top}\vec{p}_T}$.
>
> ## Extending to rank-$r$ idempotent fading matrix
>
> We now generalize this to a **rank-$r$** idempotent fading matrix
>
> $$
> \mathbf{E}
> =\sum\limits_{i=1}^{r}\frac{\vec{p}_T^{i}\mathbf{1}^{\top}}
>                      {\mathbf{1}^{\top}\vec{p}_T^{i}},
> \quad
> \vec{p}_T^{i}\ge 0,\quad
> (\vec{p}_T^{i})^{\top}\vec{p}_T^{j}=0
> \quad \forall i\neq j.
> $$
>
> ## Physical intuition
>
> * **Clustering:** Partition the item set $\mathcal{I}$ ($|\mathcal{I}|=N$) into $r$ clusters $\{\mathcal{C}_1,\dots,\mathcal{C}_r\}$, where items in the same cluster are more similar.
> > To obtain the item clusters, one may apply K-means or other vector-based clustering methods to semantic embeddings of items; alternatively, $r$-way classification can be performed on the user–item bipartite graph or the weighted sequential interaction graph using GCN or similar approaches. Since these methods rely on additional side information beyond our current framework, we leave a full exploration of cluster construction to future work.
>
> * **Cluster-wise target distributions:**
>   Define
> $$
> \vec{p}_T^{i}(x) > 0 \quad \text{if } x \in \mathcal{C}_i; \quad \vec{p}_T^{i}(x) = 0 \quad \text{otherwise}, \quad i = 1, \dots, r.
> $$
>
>
>   Because $\vec{p}_T^{~i},\forall i= 1,\cdots,r$ are disjoint, the resulting $\mathbf{E}$ is idempotent with rank $r$.
>
> * **Effect:** During fading, each preferred item is replaced **only by items within the same cluster**, filtering out cross-cluster candidates and better respecting item heterogeneity. Hence the rank-$r$ fading matrix offers improved realism. Furthermore, rank-$r$ idempotency also reduces the cost of  $\mathbf{E} \cdot \mathbf{x} = \left(\sum\limits_{i=1}^{r}\frac{\mathbf{x}^{\top}\vec{p}_T^{i}}{\mathbf{1}^{\top}\vec{p}_T^{i}}\right) \vec{1}$  from $\mathcal{O}(N^2)$ to $\mathcal{O}(rN)$.
>
>
> We believe that **PreferGrow equipped with a rank-$r$ fading matrix — cluster-wise replacement —combines theoretical elegance with practical realism**.  We will add this rank-$r$ extension as an extra appendix section and cite it in the main text.  We hope this addresses your concerns about realism and inspires further research and applications.

---

### Official Review · Reviewer_8EAw · 2025-07-08

**Clarity:** 1
**Significance:** 2
**Originality:** 2
**Rating:** 3
**Confidence:** 3

**Summary:**

This paper proposes PreferGrow, a recommender systems to learn item representations through discrete diffusion models. Existing works based on diffusion models in recommendation systems suffer from lack of negative sampling as well as misalignment in the continuous probability modeling space vs. the discrete nature of user interaction. To tackle these two problems, PreferGrow introduces a preference fading discrete diffusion model tailored for recommendation. Specifically, during the forward pass of diffusion process, the target item embedding corruption is guided by fading matrices; and during the backward pass, user embedding generated by SASRec, as well as the corrupted item embedding are concatenated to predict the preference ratio.

**Questions:**

please refer to the weakness.

furthermore, I think the authors should comparatively discuss efficiency of the proposed framework.

**Ethical Concerns:**

["NO or VERY MINOR ethics concerns only"]

**Limitations:**

limitations are properly discussed

**Paper Formatting Concerns:**

no formatting concerns

**Quality:**

2

**Strengths And Weaknesses:**

Strength:
1. this paper is technically well-motivated -- it identifies misalignment between the continuous probability modeling space in diffusion models vs. the discrete nature of user interaction.

Weakness:
1. the clarity and writing needs to be significantly improved. the technical part of this paper is extremely difficult to understand.
2. i'm confused by the incorporation of user embedding, which other diffusion-based baseline does not explore. The incorporation of user embedding seems agnostic of the idea proposed in this paper.
3. while the authors try to provide theoretical proofs, the provided proofs are mostly general diffusion derivations and the added value is very marginal.

---

> ### Author Rebuttal · Authors · 2025-07-31
>
> We sincerely thank you for your time and valuable comments. We appreciate your recognition of the technical soundness of our approach, especially our effort to address the mismatch between continuous diffusion modeling and the inherently discrete nature of user interactions in recommendation systems.
>
> > Summary: ... to learn item representations ... the target item embedding corruption ...  the corrupted item embedding ...
>
> From your summary, it appears there is a key misunderstanding of our method. Specifically, **in PreferGrow the variables $x_0$ and $x_t$ denote discrete item IDs** — that is, $x_0, x_t \in [1, 2, \dots, N]$ — rather than continuous embeddings. In our notation, only $\mathbf{x}_0, \mathbf{x}_t$ represents the continuous embedding. We will clarify this distinction more explicitly in the revised manuscript. Following the pipeline illustrated in **Figure 2**, PreferGrow consists of the following key components:
>
> 1. **Forward Perturbation Process**:  Given a preferred item $x_0$ (an integer item ID), we retain it as $x_t = x_0$ with probability $\alpha_t$, and replace it with another item $x_t \ne x_0$ with probability $1 - \alpha_t$. The replacement strategy is governed by a pre-defined fading matrix.
>
> 2. **Modeling Target**:  The model learns **preference ratios** $s_{\theta}(x_t, t, u)$, which quantify how user $u$ prefers other items over the current item $x_t$. A positive value indicates user $u$ prefers other items more than $x_t$; a negative value suggests the opposite.
>
> 3. **Backward Generation Process**:  From the estimated preference ratios $s_{\theta}(x_t, t, u)$, we compute the user’s preference distribution over all items:  $p(i \mid u), i \in [1, 2, ..., N]$.  The user embedding thus conditions the distribution $p(i \mid u)$, allowing PreferGrow to generate personalized item distributions tailored to each user—i.e., recommend different items to different users.
>
> We hope that our detailed explanation helps you gain a clear understanding of PreferGrow.
>
> > Comment 1: the clarity and writing
>
> We devoted considerable effort to carefully writing and organizing the technical parts of our paper, as well as visualizing the core pipeline in Figure 2. In fact, other reviewers (1sA3, 3oh1, XkWC, rfEc) all rated our Clarity as *Good* or *Excellent*. We believe that your concerns may stem primarily from a misunderstanding of our notation, specifically regarding the meaning of $x_0$ and $x_t$. In our response to your summary, we have clarified that $x_0$ and $x_t$ denote discrete item IDs, not continuous embeddings. We also re-explained the PreferGrow pipeline in detail. We hope that this clarification helps you better understand our method.
>
> >  Comment 2: incorporation of user embedding
>
> **PreferGrow incorporates user embeddings through conditional modeling to support personalized recommendations.** We clarify the role of the user embedding in PreferGrow, as introduced in the Preliminaries section. PreferGrow generates a discrete probability distribution over the item set, i.e., $p(i),\ i \in [1, 2, \dots, N]$. To support personalized recommendations, the model estimates a user-conditioned distribution $p(i \mid u)$, where $u$ denotes the user embedding. In our framework, $u$ is derived from the user's interaction history using the SASRec module; specifically, we use its final hidden representation as the conditional input to generate $p(i \mid u)$. Additionally, to compute the unconditional distribution $p(i)$ (i.e., without user information), we introduce a learnable non-preference embedding $\phi$, such that $p(i \mid \phi) = p(i)$. Actually, this conditional modeling approach is widely adopted in other diffusion-based recommendation models.
>
> >  Comment 3: theoretical contributions
>
> **We respectfully disagree that the theoretical contributions are marginal, as they provide a necessary foundation for future research and model development.** While it is true that our theoretical proofs focus on establishing the general correctness of the PreferGrow framework, we believe their value is far from marginal. First, a well-founded theoretical basis is crucial for enabling future developments—just as the success of DDPMs [1] and DPO [2] was grounded in solid theoretical foundations that paved the way for a broad range of follow-up work. Second, by providing clear and rigorous general proofs, we offer a reusable theoretical foundation that future studies can build upon—especially when deriving more specialized results tailored to recommendation tasks.
>
> > Comment 4: efficiency comparison.
>
> Thank you for your helpful suggestion. To address your concern, we provide a qualitative comparison of computational complexity between PreferGrow and other baselines, covering model parameters, loss computation, modeling targets, and inference cost,as summarized in the table below. (L: history length, N: number of items, B: number of neg samples, d: hidden dim, T: steps of diffusion inference)
>
> |   Complexity   |                Model Parameters                 |         Loss Computation          |    Modeling Target    |                       Inference                        |
> | :------------: | :---------------------------------------------: | :-------------------------------: | :-------------------: | :----------------------------------------------------: |
> |   **SASRec**   |    $\mathcal{O}\bigl(Ld^{2}+L^{2}d\bigr)$     | $\mathcal{O}\!\bigl((B+1)d\bigr)$ |   $\mathcal{O}(N)$    |      $\mathcal{O}\bigl(Ld^{2}+L^{2}d+Nd\bigr)$       |
> |  **DreamRec**  |  $\mathcal{O}\bigl((L+3)d^{2}+L^{2}d\bigr)$   |         $\mathcal{O}(d)$          |   $\mathcal{O}(Td)$   |   $\mathcal{O}\bigl(T(L+3)d^{2}+TL^{2}d+Nd\bigr)$    |
> | **PreferDiff** |  $\mathcal{O}\bigl((L+3)d^{2}+L^{2}d\bigr)$   | $\mathcal{O}\bigl((B+1)d\bigr)$ |   $\mathcal{O}(Td)$   |   $\mathcal{O}\bigl(T(L+3)d^{2}+TL^{2}d+Nd\bigr)$    |
> |  **DiffRec**   |              $\mathcal{O}(LN^{2})$              |         $\mathcal{O}(N)$          |   $\mathcal{O}(TN)$   |                 $\mathcal{O}(TLN^{2})$                 |
> |    **DDSR**    |   $\mathcal{O}\bigl(Ld^{2}+L^{2}d+Nd\bigr)$   |         $\mathcal{O}(N)$          |   $\mathcal{O}(TN)$   |     $\mathcal{O}\bigl(TLd^{2}+TL^{2}d+TNd\bigr)$     |
> | **PreferGrow** | $\mathcal{O}\bigl((L+3)d^{2}+L^{2}d+Nd\bigr)$ |        $\mathcal{O}(N+T)$         | $\mathcal{O}(TN^{2})$ | $\mathcal{O}\bigl(T(L+3)d^{2}+TL^{2}d+TN(d+3)\bigr)$ |
> | **PreferGrow on SIDs** | $\mathcal{O}(m(L{+}3)d^2 {+} mL^2d {+} mcd)$ | $\mathcal{O}(mc+T)$ | $\mathcal{O}(Tmc^2)$ | $\mathcal{O}(Tm(L{+}3)d^2 {+} TmL^2d {+} Tmc(d{+}3))$ |
>
> * **Model Parameters:** We never materialise the full $N^{2}$ preference‑ratio matrix; Equation (14) only requires $N$ scores, so the extra cost is $\mathcal O(Nd)$.
> * **Loss Computation:** Leveraging the idempotent fading matrix $E$, Equation (4) simplifies (see App. C.1–C.4) to element‑wise mean, `indexing`, and a pre‑computable constant—also $\mathcal O(N)$.  The seemingly complex score entropy loss consists of three parts: positive and negative terms (involving simple mean and indexing operations, $\mathcal{O}(N)$), and a constant term (ensuring non-negativity, dependent only on $t$, and pre-computable with $\mathcal{O}(T)$ complexity, $T=20$).
> * **Inference cost:**  Idempotency gives $E\cdot s_{\theta}(x_t,t,u)=\bigl(\mathbf p_T^{\top}s_{\theta}/\mathbf1^{\top}\mathbf p_T\bigr)\mathbf1$ reducing Equation (18) from $\mathcal O(N^{2})$ to $\mathcal O(N)$.
> * **Modeling Target:**  The $\mathcal{O}(N^2)$ complexity only arises from its desired modeling target. This more expressive target contributes to a higher performance ceiling for PreferGrow.
> * **Scalability to extremely large item set (billions of items):** In this case, even $\mathcal{O}(N)$ becomes impractical. As noted in Section 5, possible solutions include sparse matrix operations and reducing the item space to $m$ smaller semantic ID spaces of size $c$.
>
> To better address this concern, we will add an additional appendix section for the analytical computational cost and extend the discussion of the scalability in section 5.
>
> We also supplement this with empirical training and inference statistics, demonstrating that PreferGrow achieves superior performance without introducing additional overhead. We provide the number of training batches of size 256 ( Train B) and inference GFLOPs as $|X|$ increases, compared with DDSR.
>
> |    #Items    | 3883  | 9265  | 12101 | 11924 | 18357 |
> | :----------: | :---: | :---: | :---: | :---: | :---: |
> | Our Train B  | 1.90M | 1.85M | 0.50M | 0.50M | 0.50M |
> |  Our GFLOPs  | 18.90 | 19.31 | 19.53 | 19.52 | 20.00 |
> | DDSR Train B | 0.50M | 0.35M | 0.10M | 0.10M | 0.10M |
> | DDSR GFLOPs  | 18.30 | 18.44 | 18.51 | 18.51 | 18.67 |
>
> Overall, PreferGrow exhibits comparable algorithmic complexity to existing methods, while featuring a more expressive modeling objective, which contributes to its higher performance ceiling. **Under similar complexity, PreferGrow achieves significant performance gains without adding inference overhead by learning more expressive preference ratios.**
>
> We sincerely hope that our response has effectively **addressed your concerns**. If so, we would be grateful for your consideration in **raising the score**. Should you have any remaining questions or suggestions, **please feel free to share them**—we remain committed to **actively engaging with your feedback** and **continuously improving our work**.
>
> **References:**
>
> [1] Ho, Jonathan, Ajay Jain, and Pieter Abbeel. "Denoising diffusion probabilistic models." NeurIPS 2020.
>
> [2] Rafailov, Rafael, et al. "Direct preference optimization: Your language model is secretly a reward model." NeurIPS 2023.

---

> ### Author Response · Authors · 2025-08-07
> **Reminder for the End of Rebuttal Period**
>
> We would like to express our gratitude once more for the valuable feedback provided by the reviewer. This message
> serves as a friendly reminder as we approach the conclusion of the discussion phase. We sincerely hope that the
> improvements during rebuttal will be taken into consideration.
>
> We also would like to share some recent advancements in our design of the fading matrix, which may help further address concerns regarding realism and model flexibility.
>
> ## **Improving Realism with Rank-$r$ Fading Matrix (Cluster-wise Replacement)**
>
> We have already presented the **rank-1** idempotent fading matrix in the paper.
> $\mathbf{E}=\dfrac{\vec{p}_T \mathbf{1}^{\top}}{\mathbf{1}^{\top}\vec{p}_T}$.
>
> ## Extending to rank-$r$ idempotent fading matrix
>
> We now generalize this to a **rank-$r$** idempotent fading matrix
>
> $$
> \mathbf{E}
> =\sum\limits_{i=1}^{r}\frac{\vec{p}_T^{i}\mathbf{1}^{\top}}
>                      {\mathbf{1}^{\top}\vec{p}_T^{i}},
> \quad
> \vec{p}_T^{i}\ge 0,\quad
> (\vec{p}_T^{i})^{\top}\vec{p}_T^{j}=0
> \quad \forall i\neq j.
> $$
>
> ## Physical intuition
>
> * **Clustering:** Partition the item set $\mathcal{I}$ ($|\mathcal{I}|=N$) into $r$ clusters $\{\mathcal{C}_1,\dots,\mathcal{C}_r\}$, where items in the same cluster are more similar.
> > To obtain the item clusters, one may apply K-means or other vector-based clustering methods to semantic embeddings of items; alternatively, $r$-way classification can be performed on the user–item bipartite graph or the weighted sequential interaction graph using GCN or similar approaches. Since these methods rely on additional side information beyond our current framework, we leave a full exploration of cluster construction to future work.
>
> * **Cluster-wise target distributions:**
>   Define
> $$
> \vec{p}_T^{i}(x) > 0 \quad \text{if } x \in \mathcal{C}_i; \quad \vec{p}_T^{i}(x) = 0 \quad \text{otherwise}, \quad i = 1, \dots, r.
> $$
>
>
>   Because $\vec{p}_T^{~i},\forall i= 1,\cdots,r$ are disjoint, the resulting $\mathbf{E}$ is idempotent with rank $r$.
>
> * **Effect:** During fading, each preferred item is replaced **only by items within the same cluster**, filtering out cross-cluster candidates and better respecting item heterogeneity. Hence the rank-$r$ fading matrix offers improved realism. Furthermore, rank-$r$ idempotency also reduces the cost of  $\mathbf{E} \cdot \mathbf{x} = \left(\sum\limits_{i=1}^{r}\frac{\mathbf{x}^{\top}\vec{p}_T^{i}}{\mathbf{1}^{\top}\vec{p}_T^{i}}\right) \vec{1}$  from $\mathcal{O}(N^2)$ to $\mathcal{O}(rN)$.
>
>
> We believe that **PreferGrow equipped with a rank-$r$ fading matrix — cluster-wise replacement —combines theoretical elegance with practical realism**.  We will add this rank-$r$ extension as an extra appendix section and cite it in the main text.  We hope this addresses your concerns about realism and inspires further research and applications.

---

### Official Review · Reviewer_1sA3 · 2025-07-13

**Clarity:** 4
**Significance:** 2
**Originality:** 3
**Rating:** 4
**Confidence:** 4

**Summary:**

This paper introduces a novel recommendation system model named PreferGrow, which is based on a discrete diffusion process to address the core challenge of user preference data sparsity. Unlike previous diffusion recommendation models that primarily rely on continuous Gaussian noise, PreferGrow operates directly on the discrete item space. Its core contributions are threefold:

1.  **Discrete Preference Modeling**: The model directly models "preference ratios" between pairs of items, which is more aligned with the ranking-oriented nature of recommendation tasks.

2.  **Preference Fading**: In the forward diffusion process, instead of injecting noise, the model employs a "preference fading" mechanism (analogous to negative sampling), where a user's liked item is replaced with other items with a certain probability. This avoids assumptions about a prior noise distribution.

3.  **Preference Growing**: In the reverse generation process, the model iteratively "grows" and reconstructs user preferences from the estimated preference ratios.

The paper provides a solid theoretical foundation for PreferGrow, including a mathematical formulation based on an idempotent fading matrix, and theoretically proves the Markovian nature and reversibility of the process. Experimental results on five benchmark datasets show that PreferGrow consistently outperforms existing state-of-the-art recommendation algorithms, including other diffusion-based models.

**Questions:**

1. Please answer other questions in the weaknesses section.

2.  **Further Discussion on Scalability**: The paper candidly admits that scalability is a major limitation. Could the authors provide specific empirical data on how training/inference time varies with the number of items `|X|` on the datasets used in the experiments? Additionally, regarding the "sparse fading matrix" mentioned in future work, how can the critical "idempotency" property be guaranteed when designing such a matrix? This seems to be a non-trivial challenge.

3.  **The Deeper Connection Between "Preference Fading" and Negative Sampling**: The paper analogizes "preference fading" to negative sampling. Traditional negative sampling strategies (e.g., sampling based on item popularity) have been proven very effective. How do the authors view the relationship between the uniform replacement strategy in the "Pair-Wise" setting and these mature negative sampling techniques? Could more complex sampling strategies (like popularity bias) be incorporated into the design of the non-preferred state to further enhance performance?

4.  **Sensitivity Issue of Personalization Strength `w`**: According to the hyperparameter analysis in the paper, model performance is highly sensitive to the personalization strength `w`, which is selected during the inference phase. Does this imply that for a new deployment scenario, a validation set must be reserved specifically to tune `w`? This would introduce additional overhead and could lead to `w` overfitting to the validation set. Have the authors considered enabling the model to adaptively learn `w`, or are there more robust strategies for selecting `w`?

5.  **Explanation of $x_{-1}$ in the Point-Wise Setting**: In the paper's Point-Wise setting, the authors introduce a "universal hard negative item." How is this item actually represented? Is it a learnable embedding vector? What is the essential difference between this and simply selecting a fixed item from the item pool as a global negative?

**Ethical Concerns:**

["NO or VERY MINOR ethics concerns only"]

**Final Justification:**

The authors have resolved most of my concerns. I decided to raise my score to 4.

**Limitations:**

Yes, the authors have adequately discussed the limitations of their work. They pointed out two key issues: long training times and poor scalability on very large item sets. The authors not only identified these problems but also proposed constructive potential solutions as future research directions.

**Quality:**

3

**Strengths And Weaknesses:**

**Strengths:**

1.  **Originality:** The core idea of this paper is highly original. Applying a discrete diffusion model to recommendation systems and proposing "preference fading" as the forward process is a significant breakthrough from the existing paradigm that relies on continuous Gaussian noise. The modeling objective of "preference ratios" is also insightful, drawing inspiration from the success of ideas like DPO (Direct Preference Optimization) in language models, making it naturally suited for ranking tasks.

2.  **Quality:** The paper's theoretical foundation is very solid. The authors present a clear, matrix-based mathematical framework, particularly introducing the key concept of an "idempotent fading matrix" and providing complete proofs of the model's Markovian nature and reversibility around it. This provides strong theoretical support for the model's effectiveness.

3.  **Clarity:** The paper is well-organized and clearly written. The introduction clearly articulates the research motivation and effectively distinguishes this work from existing methods. Figure 1 intuitively contrasts different diffusion recommendation paradigms, and Figure 2 clearly illustrates the PreferGrow pipeline, greatly enhancing the paper's comprehensibility.

4.  **Significance:** This work is of significant importance to the subfield of diffusion-based recommendation systems. The experimental section comprehensively compares PreferGrow with a variety of baselines (including classic, item-level, and score-level diffusion models) and demonstrates its consistent and significant performance advantages across multiple datasets. If these results can be widely replicated, this model could become an important benchmark in the field. Furthermore, the flexibility of the model design (supporting Point-Wise, Pair-Wise, Hybrid modes) also increases its potential application value.

**Weaknesses:**

1.  **Fundamental Scalability Flaw:** This is the most critical weakness of the paper. The authors acknowledge this issue in Section 5, but its severity is somewhat underestimated. The model's core is to model the "preference ratios" of all item pairs, which leads to a computational/memory complexity of at least the square of the number of items, $|X|^2$. This is completely infeasible in real-world recommendation scenarios with millions of items. Compared to contemporary work, this is a step backward. For example, models based on negative sampling (like its encoder base, SASRec) and embedding diffusion models (like PreferDiff) have much better scalability. If the core algorithm of a recommendation system paper cannot be applied to large-scale datasets, its practical significance and impact will be considerably diminished. The proposed method feels more like a "toy" model validated on small datasets, lacking consideration for real-world application challenges.

2.  **Theoretical "Elegance" at the Cost of Realism:** The paper's theoretical derivations heavily rely on a core assumption: the fading matrix $E$ is idempotent ($E^2=E$). To satisfy this condition, the paper further derives that all column vectors of $E$ must be identical ( $ E=\frac{\vec{p} _{T}\cdot\vec{1}^{\top}}{\vec{1}^{\top}\vec{p} _{T}}$ ). This means that during the forward "fading" process, regardless of the original preferred item $x_0$, it will be replaced according to the **same** target distribution $\vec{p}_T$. This is a strong simplifying assumption that somewhat ignores the heterogeneity among items. For example, a reasonable substitute for a "faded" shirt item should be other clothing, not a refrigerator. The model cannot capture such fine-grained relationships, which calls its physical intuition and modeling capability into question. In contrast, although models like DDSR operate on simple categorical distributions, they at least do not introduce such a strong and counter-intuitive global structural assumption.

3.  **Questionable Reliability of Experimental Evaluation:** The paper reports performance surpassing all baselines on five datasets. However, the reliability of these results is questionable. The authors admit that due to computational resource constraints, all experiments were run once with a fixed random seed, without reporting error bars or conducting statistical significance tests. In machine learning, especially in recommendation systems, model performance is highly sensitive to random factors like data splits and parameter initialization. Without statistical tests, we cannot determine whether the reported performance gains are genuinely effective or merely due to random chance or specific hyperparameter tuning. For a paper seeking to advance the state-of-the-art, this is unacceptable.

4.  **Marginal Contribution vs. Disproportionate Complexity:** The core idea of this paper can be seen as "grafting" recent work in discrete diffusion and alignment (e.g., DPO) onto recommendation systems. While the combination is innovative, the resulting model complexity (e.g., the extremely tedious loss function derivation in Appendix) may be disproportionate to the actual, statistically unverified performance gains it brings. Compared to contemporary works, for instance, DDSR, which directly performs diffusion on a discrete state space, is conceptually more straightforward; PreferDiff, although a continuous model, has a more intuitive optimization objective designed for negative sampling. It is debatable whether the series of complex concepts introduced by PreferGrow, such as "preference ratios" and "idempotent fading matrix," is the right path to solving the problem.

---

> ### Author Rebuttal · Authors · 2025-07-31
>
> Thank you for your thoughtful and constructive comments. We appreciate your recognition of our framework’s originality, theory, clarity, and empirical impact. Your insights — especially on scalability and realism — guided valuable analyses and improvements outlined below.
>
> > Comment 1: **Fundamental Scalability Flaw:**
>
> **I. PreferGrow adds only $\mathcal O(N)$ Complexity in network, loss and inference.** The table below compares model, loss, and inference complexities with other baselines.  L: history length, N: #items, B: #neg samples, d: hidden dim, T: #diffusion steps
>
> |   Complexity   |          Model Parameters           |   Loss Computation    |    Modeling Target    |                 Inference                  |
> | :------------: | :---------------------------------: | :-------------------: | :-------------------: | :----------------------------------------: |
> | **PreferDiff** |  $\mathcal{O}((L+3)d^{2}+L^{2}d)$   | $\mathcal{O}((B+1)d)$ |   $\mathcal{O}(Td)$   |   $\mathcal{O}(T(L+3)d^{2}+TL^{2}d+Nd)$    |
> |    **DDSR**    |   $\mathcal{O}(Ld^{2}+L^{2}d+Nd)$   |   $\mathcal{O}(N)$    |   $\mathcal{O}(TN)$   |     $\mathcal{O}(TLd^{2}+TL^{2}d+TNd)$     |
> | **PreferGrow** | $\mathcal{O}((L+3)d^{2}+L^{2}d+Nd)$ |  $\mathcal{O}(N+T)$   | $\mathcal{O}(TN^{2})$ | $\mathcal{O}(T(L+3)d^{2}+TL^{2}d+TN(d+3))$ |
> | **PreferGrow on SIDs** | $\mathcal{O}(m(L{+}3)d^2 {+} mL^2d {+} mcd)$ | $\mathcal{O}(mc+T)$ | $\mathcal{O}(Tmc^2)$ | $\mathcal{O}(Tm(L{+}3)d^2 {+} TmL^2d {+} Tmc(d{+}3))$ |
>
> * **Model Parameters:** We never materialise the full $N^{2}$ preference‑ratio matrix; Equation (14) only requires $N$ scores, so the extra cost is only $\mathcal O(Nd)$.
> * **Loss Computation and Inference:** Thanks to idempotency, $\mathbf{E} \cdot \mathbf{x} = \left(\vec{p}_T^{\top} \mathbf{x} / \vec{1}^{\top} \vec{p}_T\right) \vec{1}$ reduces the cost from $\mathcal{O}(N^2)$ to $\mathcal{O}(N)$. The seemingly complex score entropy loss consists of three parts: positive and negative terms (involving simple mean and indexing operations, $\mathcal{O}(N)$), and a constant term (ensuring non-negativity, dependent only on $t$, and pre-computable with $\mathcal{O}(T)$ complexity).
> * **Modeling Target:** The $\mathcal{O}(N^2)$ complexity only arises from its desired modeling target. This more expressive target contributes to a higher performance ceiling for PreferGrow.
> * **Scalability to extremely large item set (billions of items):** In this case, even $\mathcal{O}(N)$ becomes impractical. As noted in Section 5, possible solutions include sparse matrix operations and reducing the item space to $m$ smaller semantic ID spaces of size $c$.
>
> To better address this concern, we will add an additional appendix section for the analytical computational cost and extend the discussion of the scalability in Section 5.
>
> **II. Incorporating SIDs is a promising next step to further reduce complexity from $\mathcal{O}(N)$ to $\mathcal{O}(mc)$.** SIDs encode each item via $m$ codebooks of size $c$, allowing up to $c^m$ items with only $\mathcal{O}(mc)$ cost.  Extending PreferGrow to operate on SIDs is our ongoing future work.
>
> > Comment 2: **Theoretical "Elegance" at the Cost of Realism:**
>
> **I. The assumption of a shared target distribution $\vec{p}_T$ is widely adopted in mainstream generative models**, such as VAE, DDPM, and RecFlow, which typically assume a common Gaussian target. Although this is a strong simplifying assumption, it has nonetheless demonstrated impressive generative power across diverse domains like text, images, and videos. In our case, the shared $\vec{p}_T$ also ensures robust cold-start generation, whereas DDSR relies on the user’s historical interaction distribution, making it difficult to generalize to cold-start users.
>
> **II. PreferGrow can also personalize the target distribution to reflect individual user preferences by setting $\vec{p}_T = \text{MLP}(\text{SASRec}_u)$.** More importantly, PreferGrow explicitly models the non-preference user state vector $\vec{\phi}$, enabling cold-start generation via $\vec{p}_T = \text{MLP}(\vec{\phi})$, even in the absence of historical interactions.
>
> **III. Incorporating semantic IDs into PreferGrow opens the door to modeling item heterogeneity more effectively.** For instance, if the semantic ID of a shirt is `[32, 32, 63]` and that of a refrigerator is `[13, 24, 17]`, then all clothing items may share the prefix `[32, 32, *]`. This allows fine-grained control in the fading process — for example, fading a shirt into other clothing can be achieved by modifying only the last ID. Similarly, fading earlier IDs enables broader transitions in item space, supporting a more flexible and semantically aware perturbation process.
>
> > Comment 3: **Questionable Reliability of Experimental Evaluation:**
>
> **To ensure the reliability of our findings, we have conducted a comprehensive re-evaluation of all experiments under multiple random seeds $[100, 200, 300, 400, 500]$ and statistical significance testing (p<0.001).** The new results consistently confirm the significant performance gains of PreferGrow over baselines under varying seeds. All updates will be included in the revised manuscript. We sincerely thank you for highlighting this important issue. Some of the results (NDCG@5) are shown below.
>
> |  Dataset   |      Beauty      |       Toys       |      Sports      |      Steam       |    MovieLens     |
> | :--------: | :--------------: | :--------------: | :--------------: | :--------------: | :--------------: |
> |   SASRec   |   0.0229±.0020   |   0.0258±.0023   |   0.0104±.0016   |   0.0193±.0002   |   0.0507±.0004   |
> | PreferDiff |   0.0224±.0031   |   0.0312±.0009   |   0.0122±.0007   |   0.0104±.0004   |   0.0348±.0005   |
> | PreferGrow | **0.0315±.0010** | **0.0326±.0007** | **0.0146±.0004** | **0.0399±.0004** | **0.0913±.0003** |
>
> > Comment 4: **Marginal Contribution vs. Disproportionate Complexity:**
>
> **I. PreferGrow’s overall complexity is $\mathcal{O}(N)$ and its performance gains have been statistically verified.** Please refer to the earlier discussion.
>
> **II. PreferGrow’s another contribution lies in its unified modeling objective that serves both generative modeling and ranking.**
>
> |   Model    | Generative Obj. | Ranking Obj. |      Loss      |
> | :--------: | :-------------: | :----------: | :------------: |
> |    DDSR    |  ✗ (discarded)  |    ✓ (CE)    | Merely Ranking |
> | PreferDiff |     ✓ (MSE)     |   ✓ (BPR)    |  Weighted sum  |
> | PreferGrow |   ✓ (ratios)    |  ✓ (ratios)  |  Unified one   |
>
> Specifically, the **preference ratio** $\log\frac{p_t(y)}{p_t(x_t)}$ is both **ranking-oriented** and a discrete approximation of the **score function** $\nabla \log p_t(x_t)$ in generative diffusion models:
> $$
> \nabla \log p_t(x_t) = \frac{p_t(y\in\mathbb{R}^d) - p_t(x_t)}{p_t(x_t)} \sim \frac{p_t(y\in[1,\cdots,N]) - p_t(x_t)}{p_t(x_t)} = \frac{p_t(y)}{p_t(x_t)} - 1.
> $$
> Optimizing preference ratios thus promotes correct rankings and improves log-likelihood estimation. The use of *preference ratios* and *idempotent fading* is not added complexity, but a principled design that offers both **theoretical grounding** and **practical gains**.
>
> > Comment 5: **Further Discussion on Scalability**:
>
> **Under similar complexity, PreferGrow achieves significant performance gains without adding inference overhead by learning more expressive preference ratios.** We provide the number of training batches of size 256 ( Train B) and inference GFLOPs as $|X|$ increases, compared with DDSR.
>
> |    #Items    | 3883  | 9265  | 12101 | 11924 | 18357 |
> | :----------: | :---: | :---: | :---: | :---: | :---: |
> | Our Train B  | 1.90M | 1.85M | 0.50M | 0.50M | 0.50M |
> |  Our GFLOPs  | 18.90 | 19.31 | 19.53 | 19.52 | 20.00 |
> | DDSR Train B | 0.50M | 0.35M | 0.10M | 0.10M | 0.10M |
> | DDSR GFLOPs  | 18.30 | 18.44 | 18.51 | 18.51 | 18.67 |
>
> Only a sparse non-preference vector $\vec{p}_T$ is needed to construct a **sparse, idempotent** fading matrix. However, based on our analysis, incorporating **semantic IDs** is a more promising direction. PreferGrow with SIDs reduces model complexity to $\mathcal{O}(mc)$ and target modeling to $\mathcal{O}(mc^2)$, further enhancing scalability and efficiency.
>
> > Comment 6: **The Deeper Connection Between "Preference Fading" and Negative Sampling**:
>
> **Even a simple uniform replacement yields strong performance, further demonstrating the effectiveness and robustness of the PreferGrow framework.** We appreciate your insightful question. As discussed at the end of **Section 3.1.2**, prior negative sampling techniques can indeed inspire the design of the fading matrix.  We agree that incorporating more mature or complex negative sampling strategies has strong potential to enhance performance. We have already explored this by introducing a learnable vector as the non-preferred state, which also shows promising gains (Table 1).
>
> > Comment 7: **Sensitivity Issue of Personalization Strength `w`**
>
> **The personalization weight $w$ is locally stable and consistent across data splits, making it tunable like a standard hyperparameter**. As shown in **Figure 4**, PreferGrow is sensitive to extreme values (e.g., $w = 0$ or $20$) but **stable within a practical range** ( $w \in [5, 15]$). We further evaluated $w \in [ 0, 2, 5, 10 ]$ and found minimal deviation in the optimal $w$ across training, validation, and test sets, confirming its robustness.
>
> |           Dataset           | MovieLens | Steam | Beauty | Sports | Toys  |
> | :-------------------------: | :-------: | :---: | :----: | :----: | :---: |
> | Mean abs. error of opt. $w$ |     0     |   0   | 0.778  |  0.05  | 0.775 |
>
> > Comment 8: **Explanation of $x\_{-1}$ in the Point-Wise Setting**:
>
> Yes, $x_{-1}$ is a learnable embedding. Using a real item as a global negative may conflict with some users' preferences, while a learnable embedding offers a consistent and conflict-free negative reference.
>
> Thanks again for insightful feedbacks.

---

> > ### Comment · Reviewer_1sA3 · 2025-08-09
> >
> > Thank you for your detailed response, which has resolved most of my doubts.

---

> > > ### Author Response · Authors · 2025-08-09
> > >
> > > We are deeply grateful for your thoughtful and constructive feedback. Guided by your insightful suggestions, we conducted additional in-depth analyses of our method, which have further enhanced the soundness and completeness of the paper. Thank you again for your time and valuable comments. We would be glad to address any further questions and are more than happy to continue the discussion.

---

> ### Author Response · Authors · 2025-08-07
> **Reminder for the End of Rebuttal Period**
>
> We would like to express our gratitude once more for the valuable feedback provided by the reviewer. This message
> serves as a friendly reminder as we approach the conclusion of the discussion phase. We sincerely hope that the
> improvements during rebuttal will be taken into consideration.
>
> We also would like to share some recent advancements in our design of the fading matrix, which may help further address concerns regarding realism and model flexibility.
>
> ## **Improving Realism with Rank-$r$ Fading Matrix (Cluster-wise Replacement)**
>
> We have already presented the **rank-1** idempotent fading matrix in the paper.
> $\mathbf{E}=\dfrac{\vec{p}_T \mathbf{1}^{\top}}{\mathbf{1}^{\top}\vec{p}_T}$.
>
> ## Extending to rank-$r$ idempotent fading matrix
>
> We now generalize this to a **rank-$r$** idempotent fading matrix
>
> $$
> \mathbf{E}
> =\sum\limits_{i=1}^{r}\frac{\vec{p}_T^{i}\mathbf{1}^{\top}}
>                      {\mathbf{1}^{\top}\vec{p}_T^{i}},
> \quad
> \vec{p}_T^{i}\ge 0,\quad
> (\vec{p}_T^{i})^{\top}\vec{p}_T^{j}=0
> \quad \forall i\neq j.
> $$
>
> ## Physical intuition
>
> * **Clustering:** Partition the item set $\mathcal{I}$ ($|\mathcal{I}|=N$) into $r$ clusters $\{\mathcal{C}_1,\dots,\mathcal{C}_r\}$, where items in the same cluster are more similar.
> > To obtain the item clusters, one may apply K-means or other vector-based clustering methods to semantic embeddings of items; alternatively, $r$-way classification can be performed on the user–item bipartite graph or the weighted sequential interaction graph using GCN or similar approaches. Since these methods rely on additional side information beyond our current framework, we leave a full exploration of cluster construction to future work.
>
> * **Cluster-wise target distributions:**
>   Define
> $$
> \vec{p}_T^{i}(x) > 0 \quad \text{if } x \in \mathcal{C}_i; \quad \vec{p}_T^{i}(x) = 0 \quad \text{otherwise}, \quad i = 1, \dots, r.
> $$
>
>
>   Because $\vec{p}_T^{~i},\forall i= 1,\cdots,r$ are disjoint, the resulting $\mathbf{E}$ is idempotent with rank $r$.
>
> * **Effect:** During fading, each preferred item is replaced **only by items within the same cluster**, filtering out cross-cluster candidates and better respecting item heterogeneity. Hence the rank-$r$ fading matrix offers improved realism. Furthermore, rank-$r$ idempotency also reduces the cost of  $\mathbf{E} \cdot \mathbf{x} = \left(\sum\limits_{i=1}^{r}\frac{\mathbf{x}^{\top}\vec{p}_T^{i}}{\mathbf{1}^{\top}\vec{p}_T^{i}}\right) \vec{1}$  from $\mathcal{O}(N^2)$ to $\mathcal{O}(rN)$.
>
>
> We believe that **PreferGrow equipped with a rank-$r$ fading matrix — cluster-wise replacement —combines theoretical elegance with practical realism**.  We will add this rank-$r$ extension as an extra appendix section and cite it in the main text.  We hope this addresses your concerns about realism and inspires further research and applications.

---

> ### Author Response · Authors · 2025-08-09
>
> Since the concerns have been resolved, we hope that our improvements will be taken into consideration, and we would greatly appreciate it if you could increase the score.

---

### Official Review · Reviewer_3oh1 · 2025-07-19

**Clarity:** 3
**Significance:** 3
**Originality:** 4
**Rating:** 4
**Confidence:** 3

**Summary:**

This paper proposes a discrete diffusion-based recommendation method that models relative preference ratios by fading and growing user preferences over the discrete item corpus, directly aligning with the ranking-oriented, discrete nature of recommendation tasks. It replaces the preferred item with alternatives, physically akin to negative sampling. During backward generation, it iteratively grows the preference signals from the estimated ratios.

**Questions:**

Please see Weakness. If there is any misunderstanding in my interpretation, please feel free to clarify it in the rebuttal. I would be happy to revise my score if my concerns are adequately addressed.

**Ethical Concerns:**

["NO or VERY MINOR ethics concerns only"]

**Final Justification:**

The authors' response addresses some of my concerns. I'd like to raise my score.

**Limitations:**

Yes.

**Quality:**

3

**Strengths And Weaknesses:**

Advantage:
1. The discrete modeling of preference ratios that aligns with the ranking-oriented nature of recommendation tasks. Noise is added by replacing the preferred item with alternatives, enabling explicit negative sampling and removing prior noise assumptions.
2. Idempotent fading matrix with closed-form solution whose idempotent property is critical for ensuring both the Markov property and reversibility. Flexible architecture supports point-wise, pair-wise, and hybrid preference modeling.
3. Extensive experiments on five datasets showing consistent outperformance of existing diffusion-based recommendation method.

My concern:
In this paper, the author adopts an idempotent matrix E as the fading matrix, which provides desirable properties for theoretical analysis (e.g., the Markov property and invertibility). However, at the implementation level, the proposed method uses a unified setting for the fading matrix, as shown in Appendices C.1, C.2, and C.3. In this setting, forward denoising either results in “staying at the current item” or “randomly replacing the current item with others uniformly.” I am concerned that such a design might degrade the model into a pure weighted negative sampling method, thereby eliminating the advantages of employing a diffusion model. Let me elaborate further. In item-level and preference score-level diffusion methods, the representations of items and preference scores are typically correlated during the forward process. However, in this paper, due to the uniform design of the fading matrix, the reference items at each step appear to be mostly independent of one another (e.g., “staying at the current item” or “randomly replacing the current item with others uniformly”  between two steps). As a result, the proposed method may degrade into a simple negative sampling scheme, as illustrated in Appendices C.1–C.3, thus undermining the core motivation and benefits of the diffusion model. In my opinion, the idempotent matrix E should not be designed in such a uniform way. At the very least, some degree of semantic or structural correlation should be preserved in the matrix—for example, assigning a higher probability to replacing the current item with a similar one.

---

> ### Author Rebuttal · Authors · 2025-07-31
>
> We sincerely thank you for your time and thoughtful comments. We appreciate your recognition of our discrete diffusion framework, the principled design of the idempotent fading matrix, and the alignment of preference ratio modeling with ranking objectives.
>
> > Comment 1: the reference items at each step appear to be mostly independent of one another
>
> **The reference items at each step are in fact not independent in PreferGrow.** As explicitly shown in **Equation (7)** of our paper, we define a probabilistic transition matrix from step $s$ to step $t$, which models how reference items evolve over time and the progress involves item correlations. The perception of independence may stem from a misunderstanding. For example, in item-level diffusion models, the item embedding and the Gaussian noise are independent. However, the inter-step correlation is introduced by the gradual increase of noise during the forward process, which perturbs the item embeddings progressively [1] [2]. Similarly, in PreferGrow, inter-step correlation arises from the changing probability of replacing the current item. As the diffusion progresses, the probability of remaining at the current item decreases, while the probability of transitioning to other items increases. This smooth shift reflects a temporal dependency, analogous to how noise accumulation introduces correlation in continuous diffusion models. We will add a brief note in the revision to clarify this confusing point.
>
>
> > Comment 2: PreferGrow degenerates into weighted negative sampling under idempotent fading, losing the key strengths of diffusion.
>
> **PreferGrow introduces correlated reference items, enabling indirect optimization of $\log p\_{\text{data}}(x\_0)$ without sacrificing the core advantages of diffusion models.** It is true that during preference fading, when the current item is replaced by others, PreferGrow may appear similar in mechanism to weighted negative sampling. However, this does not mean it loses the core advantages of diffusion models. The core advantages of diffusion models lies in their ability to introduce a sequence of correlated latent variables $\{x\_t\}\_{t=1}^{T}$ across timesteps through noise injection. These latent variables form a correlated trajectory that enables the indirect optimization of the log-likelihood of the original data distribution $\log p_{data}(x\_0) $, which is intractable to optimize directly but becomes easier to optimize in each diffusion-recover with intermediate latent variables. The inter-step correlation of latent variables has already been addressed in our response to Comment 1. Here, we emphasize that PreferGrow retains the log-likelihood optimization property. Specifically, the modeling target in PreferGrow is the preference ratios, proved to be a discretized version of the score function $\nabla \log p\_{t}(x\_t) $ [3].
>
> $$
> \nabla \log p_t(x_t) = \frac{p_t(y\in\mathbb{R}^d) - p_t(x_t)}{p_t(x_t)} \sim \frac{p_t(y\in\{1,\cdots,N\}) - p_t(x_t)}{p_t(x_t)} = \frac{p_t(y)}{p_t(x_t)} - 1.
> $$
>
> As a result, optimizing the preference ratios under score entropy loss is equivalent to optimizing the **log-likelihood of the latent variables**, which effectively serves as a surrogate for the original data log-likelihood ( proved in SEDD [3]). In summary, PreferGrow preserves the core advantages of diffusion modeling —— correlated latent variables across timesteps and their role in indirectly optimizing the log‑likelihood of the original data distribution. Moreover, its resemblance to a weighted negative‑sampling mechanism in the forward process further enhances its suitability for recommendation tasks, given the well‑established effectiveness of negative sampling in recommendation.
>
> > Comment 3: Instead of using the unified fading‑matrix settings described in Appendices C.1–C.3, the fading matrix should retain the underlying semantic or structural correlations.
>
> In addition to the unified fading‑matrix settings, we introduce an **adaptive setting** with a **learnable fading matrix** $\vec{p}_T=\text{Softmax}(\vec{\theta})$ that can adaptively adjust the replacement probability for other items. The empirical results for this adaptive variant are reported in **Table 1**, and the theoretical details are provided in **Appendix C.4**. Furthermore, PreferGrow can also personalize the target distribution to reflect individual user preferences by setting $\vec{p}_T = \text{MLP}(\text{SASRec}_u)$. More importantly, PreferGrow explicitly models the non-preference user state vector $\vec{\phi}$, enabling cold-start generation via $\vec{p}_T = \text{MLP}(\vec{\phi})$, even in the absence of historical interactions.
>
> Thank you once again for your valuable feedback. We hope that our efforts during the rebuttal period will be taken into account in your evaluation. We remain open and willing to discuss any new questions or suggestions you might have.
>
> > Comment 4: The idempotent matrix E should not be designed in such a uniform way. At the very least, some degree of semantic or structural correlation should be preserved in the matrix
>
> **I. Our use of a uniform fading matrix was intentional—to validate PreferGrow’s effectiveness under the simplest setting.**  Even a simple uniform fading matrix $E$ yields strong performance, further demonstrating the robustness and effectiveness of the PreferGrow framework. We also appreciate your insightful suggestions.
>
> **II. To capture structural correlations, the fading distribution $\vec{p}_T$ can be designed using advanced negative sampling techniques.**  As discussed in **Section 3.1.2**, prior negative sampling strategies can inspire the design of $\vec{p}_T$. We agree that incorporating more sophisticated methods has strong potential to enhance performance. Furthermore, adaptive designs such as $\vec{p}_T = \text{Softmax}(\vec{\theta})$ and $\vec{p}_T = \text{MLP}(\text{SASRec}_u)$ (mentioned in our earlier response) can also capture meaningful structural patterns.
>
> **III. To model semantic correlations, integrating Semantic IDs offers a promising direction we are actively exploring.**  For example, if the semantic ID of a shirt is `[32, 32, 63]` and that of a refrigerator is `[13, 24, 17]`, then all clothing items may share the prefix `[32, 32, *]`.  This enables fine-grained fading—e.g., fading a shirt into other clothing by modifying only the last ID. Similarly, modifying earlier IDs allows broader transitions in item space, enabling a more flexible and semantically-aware perturbation process.
>
> **References:**
>
> [1] Ho, Jonathan, Ajay Jain, and Pieter Abbeel. "Denoising diffusion probabilistic models." NeurIPS 2020.
>
> [2] Song, Yang, et al. "Score-based generative modeling through stochastic differential equations." ICLR 2021.
>
> [3] Lou, Aaron, Chenlin Meng, and Stefano Ermon. "Discrete diffusion modeling by estimating the ratios of the data distribution." ICML 2024.

---

> > ### Comment · Reviewer_3oh1 · 2025-08-04
> >
> > Thanks for your response. Comments 1~3 have been addressed. I also agree with "Theoretical Elegance at the Cost of Realism" expressed by other reviewers. I decided to raise my score to 4. Thanks!

---

> > > ### Author Response · Authors · 2025-08-04
> > >
> > > We sincerely appreciate your thoughtful feedback and deeply thank you for the score increase. In our responses to other reviewers, we also shared our perspective on the theme of "Theoretical Elegance at the Cost of Realism", and we hope our work can inspire future research that strives for elegance in both theory and practical realism.

---

> > > ### Author Response · Authors · 2025-08-07
> > > **Improving Realism with Rank-$r$ Fading Matrix (Cluster-wise Replacement)**
> > >
> > > We would like to share some recent advancements in our design of the fading matrix, which may help further address concerns regarding realism and model flexibility.
> > >
> > > We have already presented the **rank-1** idempotent fading matrix in the paper.
> > > $\mathbf{E}=\dfrac{\vec{p}_T \mathbf{1}^{\top}}{\mathbf{1}^{\top}\vec{p}_T}$.
> > >
> > > ## Extending to rank-$r$ idempotent fading matrix
> > >
> > > We now generalize this to a **rank-$r$** idempotent fading matrix
> > >
> > > $$
> > > \mathbf{E}
> > > =\sum\limits_{i=1}^{r}\frac{\vec{p}_T^{i}\mathbf{1}^{\top}}
> > >                      {\mathbf{1}^{\top}\vec{p}_T^{i}},
> > > \quad
> > > \vec{p}_T^{i}\ge 0,\quad
> > > (\vec{p}_T^{i})^{\top}\vec{p}_T^{j}=0
> > > \quad \forall i\neq j.
> > > $$
> > >
> > > ## Physical intuition
> > >
> > > * **Clustering:** Partition the item set $\mathcal{I}$ ($|\mathcal{I}|=N$) into $r$ clusters $\{\mathcal{C}_1,\dots,\mathcal{C}_r\}$, where items in the same cluster are more similar.
> > > > To obtain the item clusters, one may apply K-means or other vector-based clustering methods to semantic embeddings of items; alternatively, $r$-way classification can be performed on the user–item bipartite graph or the weighted sequential interaction graph using GCN or similar approaches. Since these methods rely on additional side information beyond our current framework, we leave a full exploration of cluster construction to future work.
> > >
> > > * **Cluster-wise target distributions:**
> > >   Define
> > > $$
> > > \vec{p}_T^{i}(x) > 0 \quad \text{if } x \in \mathcal{C}_i; \quad \vec{p}_T^{i}(x) = 0 \quad \text{otherwise}, \quad i = 1, \dots, r.
> > > $$
> > >
> > >
> > >   Because $\vec{p}_T^{~i},\forall i= 1,\cdots,r$ are disjoint, the resulting $\mathbf{E}$ is idempotent with rank $r$.
> > >
> > > * **Effect:** During fading, each preferred item is replaced **only by items within the same cluster**, filtering out cross-cluster candidates and better respecting item heterogeneity. Hence the rank-$r$ fading matrix offers improved realism. Furthermore, rank-$r$ idempotency also reduces the cost of  $\mathbf{E} \cdot \mathbf{x} = \left(\sum\limits_{i=1}^{r}\frac{\mathbf{x}^{\top}\vec{p}_T^{i}}{\mathbf{1}^{\top}\vec{p}_T^{i}}\right) \vec{1}$  from $\mathcal{O}(N^2)$ to $\mathcal{O}(rN)$.
> > >
> > >
> > > We believe that **PreferGrow equipped with a rank-$r$ fading matrix — cluster-wise replacement —combines theoretical elegance with practical realism**.  We will add this rank-$r$ extension as an extra appendix section and cite it in the main text.  We hope this addresses your concerns about realism and inspires further research and applications.

---

### Note · Authors · 2025-08-12

Dear Reviewers, ACs, SACs, and PCs,

We sincerely thank the reviewers for their positive evaluation, highlighting the following strengths:
1. **Novelty**: novel framework ($\color{red}{\text{1sA3}}$, $\color{blue}{\text{XkWC}}$, $\color{orange}{\text{rfEc}}$); highly original core idea ($\color{red}{\text{1sA3}}$);
2. **Significance**: significant approach to recommendation systems (**$\color{orange}{\text{rfEc}}$**); notable importance to the subfield of diffusion-based recommendation systems (**$\color{red}{\text{1sA3}}$**).
3. **Quality**: solid theoretical foundation (**$\color{orange}{\text{rfEc}}$, $\color{blue}{\text{XkWC}}$, $\color{red}{\text{1sA3}}$**); extensive experiments (**$\color{orange}{\text{rfEc}}$, $\color{blue}{\text{XkWC}}$, $\color{red}{\text{1sA3}}$, $\color{green}{\text{3oh1}}$**); well-organized and clearly written (**$\color{red}{\text{1sA3}}$**).

The **major concerns** raised by the reviewers and our responses are as follows:
1. **High computational complexity** (**$\color{orange}{\text{rfEc}}$, $\color{blue}{\text{XkWC}}$, $\color{red}{\text{1sA3}}$, $\color{green}{\text{3oh1}}$, $\color{purple}{\text{8EAw}}$**): We **conducted a quantitative analysis** comparing our model’s complexity (parameters, loss computation, and inference cost) with baselines, showing it remains comparable $\mathcal{O}(N)$.
2. **Unrealistic uniform rank-1 fading matrix** (**$\color{red}{\text{1sA3}}$, $\color{green}{\text{3oh1}}$**): We **already explored adaptive rank-1 fading matrix** for user personalization **in the paper** and **extend the theoretical proof to a rank-$r$ fading matrix** respecting item heterogeneity.
3. **Sensitivity to parameter $w$** (**$\color{red}{\text{1sA3}}$, $\color{orange}{\text{rfEc}}$**): We **conducted experiments** demonstrating that $w$ is locally stable, consistent across data splits, and tunable like any standard hyperparameter.

All reviewers who replied (**$\color{orange}{\text{rfEc}}$, $\color{green}{\text{3oh1}}$, $\color{red}{\text{1sA3}}$**) acknowledged that most concerns were resolved. We hope these improvements will be taken into consideration.

**Post-rebuttal commitments**:
1. Revise the *Limitations* section with quantitative complexity and scalability analyses (with details in Appendix).
2. Add an appendix section with the theoretical proof and physical interpretation of the extended rank-$r$ solution.
3. Improve Section 4.3 with a detailed analysis of parameter $w$.

Best regards,

Authors

---

### Decision · Program_Chairs · 2025-09-17

**Decision:**

Accept (poster)

**Comment:**

This paper proposes PreferGrow, a discrete diffusion-based recommender that models preference ratios by fading and growing user preferences over the discrete items. PreferGrow possesses three distinct aspects compared to existing diffusion-based recommenders: 1) Discrete modeling of preference ratios, 2) Perturbing via preference fading, and 3) Preference reconstruction via growing. The authors showed the performance gains of PreferGrow compared to SOTA diffusion-based recommenders across five benchmark datasets.

Strengths:
+ Discrete modeling of preference ratios aligns well with the ranking-oriented nature of recommender systems.
+ A flexible architecture that supports  point-wise, pair-wise, and hybrid preference modeling
+ Thorough experiments on five datasets showing consistent performance compared to SOTA diffusion-based recommenders

Weaknesses:
- The authors addressed major concerns from the reviewers during rebuttal
- Minor limitations in terms of longer training time, as mentioned by the authors.

Overall, PreferGrow is a novel framework with an original core idea, solid theoretical analysis, and thorough experiments. I support this paper to be accepted.